# Intrinsic Self-Supervision for Data Quality Audits

**Fabian Gröger**[* 1,2]**, Simone Lionetti**[2]**, Philippe Gottfrois**[1]**, Alvaro Gonzalez-Jimenez**[1]**,**
**Ludovic Amruthalingam**[2]**, Labelling Consortium**[† 3]**, Matthew Groh**[4]**,**
**Alexander A. Navarini**[‡ 1,3]**, Marc Pouly**[‡ 2]

[1]University of Basel   [2]Lucerne University of Applied Sciences and Arts
[3]University Hospital of Basel   [4]Northwestern University

## Abstract

Benchmark datasets in computer vision often contain off-topic images, near duplicates, and label errors, leading to inaccurate estimates of model performance. In this paper, we revisit the task of data cleaning and formalize it as either a ranking problem, which significantly reduces human inspection effort, or a scoring problem, which allows for automated decisions based on score distributions. We find that a specific combination of context-aware self-supervised representation learning and distance-based indicators is effective in finding issues without annotation biases. This methodology, which we call SELFCLEAN, surpasses state-of-the-art performance in detecting off-topic images, near duplicates, and label errors within widely-used image datasets, such as ImageNet-1k, Food-101N, and STL-10, both for synthetic issues and real contamination. We apply the detailed method to multiple image benchmarks, identify up to 16% of issues, and confirm an improvement in evaluation reliability upon cleaning. The official implementation can be found at: `https://github.com/Digital-Dermatology/SelfClean`.

## 1   Introduction

In traditional machine learning (ML), data cleaning is essential since minor contamination in the dataset can significantly impact model performance and robustness [1]. However, with the rise of deep learning (DL) and large-scale datasets, data cleaning has become less crucial as large models have shown to work relatively well even when training data has low quality [2]. Validating and cleaning large datasets is challenging, especially for high-dimensional data, because thorough manual verification is often not feasible. Thus, a lot of research has been focusing on learning from noisy data [3] rather than fixing quality issues, as the overwhelming benefits of large-scale datasets are believed to exceed the drawback of diminished control. On the other hand, for many domains, the size of available datasets is still one of the main limiting factors for the progress of artificial intelligence (AI). In these low-data regimes, the importance of clean data is more pronounced since even fractional amounts of poor-quality samples can substantially hamper performance and possibly lead to wrong conclusions [4]. This is especially relevant in high-stakes settings such as the medical domain, where high-quality data is needed to train robust models and validate their performance. However, also in these domains, many practitioners rather focus on data quantity as a key performance driver and implicitly assume a high-quality collection process [5]. Thus, even medical datasets are known to contain varying noise levels, which can substantially undermine the progress of ML [6].

The necessity to report comparable results has led DL practitioners to heavily rely on benchmark datasets despite them being known for containing data quality issues. For example, an evaluation of

---

[*]Correspondence: `fabian.groeger@unibas.ch`

[†]Valerie Amann, Elisabeth Gössinger, Hazem Juratli, Beda Mühleisen, Alina Müller, and Veronika Schmidt

[‡]Joint last authorship

38th Conference on Neural Information Processing Systems (NeurIPS 2024) Track on Datasets and Benchmarks.

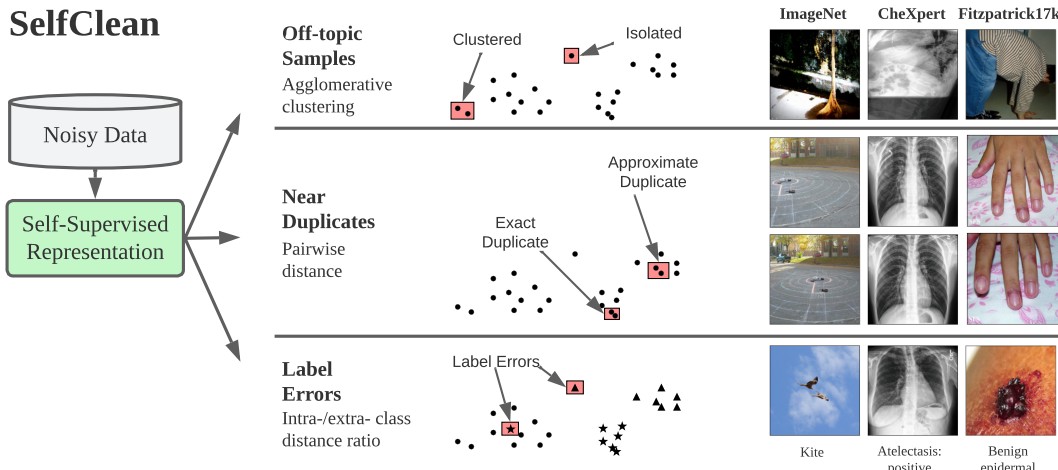

Figure 1: SELFCLEAN first trains a self-supervised encoder on noisy data to obtain latent representations for dataset samples. It then detects off-topic samples with agglomerative clustering, near duplicates based on pairwise distances, and label errors using the intra-/extra- class distance ratio.

ten of the most used benchmark datasets found them to have an average label error rate of 3.4% in the evaluation set [7]. Such issues in benchmark sets, especially when used for evaluation, undermine the framework by which scientific progress is measured. Specifically, contamination in evaluation sets corrupts scores, making it unclear which methods successfully handle edge cases and obscuring their proximity to optimal performance. This is particularly relevant since many popular benchmarks are saturating, i.e., only saw minor relative changes in performance over the last few years [8]. Data quality issues in training sets, instead, may hinder optimization and produce suboptimal models. Importantly, despite the need for correct evaluation data, cleaning evaluation sets can be problematic, as it may optimistically bias performance estimates. Ignoring known data quality issues during evaluation is, however, also incorrect, so an appropriate compromise is necessary.

In this paper, we address three types of data quality issues that illustrate these mechanisms well. *Off-topic samples*, i.e., inputs included in a dataset by mistake, add noise to evaluation metrics while slowing down and confusing training. *Near duplicates*, i.e., different views of the same object, produce arbitrary re-weighting in the evaluation set, reduce variability in the training set, and most importantly, often introduce leaks between training and evaluation sets that can lead to over-optimistic results. *Label errors*, i.e., wrongly annotated samples, result in incorrect evaluation and poison the training process. We focus on these three data quality issues because we empirically found them to be frequent in existing image benchmark datasets and challenging to detect. There are of course other types of data quality issues, including many that can be detected using ad-hoc rules, such as odd brightness, aspect ratio, resolution, sharpness, and entropy in the case of images.

In this paper, we formulate dataset cleaning as a set of ranking problems, which greatly reduce the effort for manual inspection, or alternatively as a set of scoring problems, which can be used for fully automatic decisions based on score distributions. We then find that a combination of self-supervised, dataset-specific representation learning and distance-based indicators can effectively identify multiple issues in image collections. We apply this approach to well-known benchmark datasets in computer vision and medical imaging, and discuss implications for reliability of results across these domains. The outlined method enables practitioners to audit data collections, increase evaluation reliability, and amend the training set to improve results. This work contributes to data-centric ML [9] and aims to bolster confidence in both existing and newly collected datasets. In summary, the main contributions are: 1) A novel data cleaning procedure called SELFCLEAN, which can be used to find off-topic samples, near duplicates, and label errors, and relies exclusively on the dataset itself, illustrated in figure 1. 2) A detailed comparison between this cleaning method and competing approaches on synthetic and natural contamination, including validation against human experts. 3) The application of SELFCLEAN to well-known benchmarks in computer vision and medical imaging and the identification of their issues. 4) A practical recommendation to clean training and evaluation splits of benchmark datasets as a reasonable trade-off between correctness and bias for more accurate performance estimates.

## 2 Related work

Data cleaning is a core component of data analytics and a topic of interest in the data management community [10]. Recently, the data-centric AI initiative [9] brought it back to the attention of ML researchers, resulting in the development of data cleaning tools. For instance, Vailoppilly et al. [11] proposed an all-in-one "data cleansing" tool based on dimensionality reduction, a DL noise classifier, and a denoising model. Tools for data cleaning also started to appear, including CleanLab [12] and CleanVision [13], Lightly [14], and FastDup [15]. Most data cleaning approaches require dimensionality reduction to work with high-dimensional data such as images. This includes traditional approaches such as PCA [16] or t-SNE [17], and feature extraction with deep encoders, which are usually trained on natural image databases such as ImageNet [18]. In the last few years, self-supervised learning (SSL) [19] was shown to learn more representative latent spaces compared to supervised training [20–22]. Furthermore, Cao and Wu [23] demonstrated that SSL can learn meaningful latent spaces even with small datasets, low resolution, and small architectures. Inspired by these results and unlike previous works, we rely on SSL as a basis to detect three important types of data quality issues encountered in practice: off-topic samples, near duplicates, and label errors [10]. Since these sub-problems are typically addressed separately in the literature, we briefly review them in turn.

The problem of identifying off-topic samples is closely related to generalized out-of-distribution detection [24] and is akin to outlier detection, which involves both normal and anomalous samples [25]. Outlier detection can be addressed with supervised, unsupervised, and semi-supervised learning and was initially developed to fit data more smoothly [26]. In the realm of data cleaning, where the nature of off-topic samples is generally unknown, it is most similar to the unsupervised setting. Outliers in low-density regions can be found using reconstruction errors [27, 28], classification [29], or probabilistic approaches [30]. For a detailed review of these methods, see [25].

Near-duplicate detection is traditionally based on representation matching [31, 32]. Most DL approaches follow a similar strategy, where feature vectors are extracted by a deep network and used for content-based matching [33]. Another option is to learn a similarity metric between samples with Siamese neural networks [34]. A recent approach for copy detection (i.e., near-duplicate detection) uses a contrastive self-supervised objective with entropy regularization to ensure consistent separation of image descriptions [35]. However, it requires a manually adapted threshold for each dataset [36].

The identification of label errors is generally focused on prediction-label agreement via confusion matrices and proceeds by removing samples with low recognition rate [37] or parts of the minority classes [38]. There are exceptions, such as recent approaches based on supervised contrastive learning for label error correction [39, 40]. Another prominent method is confident learning, which identifies label errors based on noisy data pruning, using probabilistic thresholds to estimate noise and ranking examples to train with confidence [41].

## 3 Methodology

Let $\mathcal{X} = \{(\boldsymbol{x}_i, l_i) \mid i \in \mathcal{I}\}$ be an image classification dataset to be cleaned, where $\mathcal{I} = \{1, \ldots, N\}$ is the index set, $\boldsymbol{x}_i$ is the $i$-th sample, and $l_i \in \{1, \ldots, L\}$ is the $i$-th label. For each issue type, we construct a scoring function $s$ that assigns values in $[0, 1]$ to samples or pairs thereof, such that elements with a lower score are more likely to be problematic. Sorting samples by the value obtained from the scoring function $s$ induces a ranking $R$ where more likely issues appear earlier.

### 3.1 Representation learning

As a first step, we train a deep feature extractor $f$ with parameters $\theta$ on the dataset $\mathcal{X}$ using self-supervised learning (SSL), which learns representations by solving auxiliary tasks. Let $\boldsymbol{e}_i = f(\boldsymbol{x}_i; \theta) \in \mathbb{R}^D$ be the representation of sample $\boldsymbol{x}_i$ obtained with $f$, where $D$ denotes the latent dimension. Note that SSL is performed on the entire dataset including data quality issues. Any SSL method can be used, as investigated in appendix F.5. Here, we consider SimCLR [42] and DINO [43], which were shown to produce meaningful latent spaces [20, 21]. SimCLR is a contrastive approach that compares different views of the same image against other randomly sampled ones. DINO is a self-distillation method which trains a student network to match a teacher network on different views of the same image. For both strategies, we rely on vision transformer (ViT) encoders, as detailed in appendix C and ablated in F.6.

As feature normalization is often built into the SSL training objective, it is natural to compare points in its latent space using cosine similarity, $\text{sim}(\boldsymbol{e}_i, \boldsymbol{e}_j) = \boldsymbol{e}_i^\top \boldsymbol{e}_j / (\|\boldsymbol{e}_i\|_2 \|\boldsymbol{e}_j\|_2)$, and the associated distance scaled to $[0, 1]$, $\text{dist}(\boldsymbol{e}_i, \boldsymbol{e}_j) = (1 - \text{sim}(\boldsymbol{e}_i, \boldsymbol{e}_j))/2$. We explicitly include $L_2$-normalization during training and inference for strategies without normalization (e.g., DINO), such that their latent space is a unit hypersphere of dimension $D - 1$. In appendix F.1, we present an ablation study of this normalization and investigate the influence of different distance functions.

## 3.2  Distance-based indicators

Dataset-specific representations based on inductive bias can be coupled with separate distance-based indicators to identify candidate issues. Below we introduce each issue type and the corresponding indicator function used to detect them.

**Off-topic samples.** We define samples as off-topic when they are included in the dataset by mistake. Images from extraneous modalities, affected by device malfunctions, or without any object of interest are some examples. Atypical samples, due e.g. to the phenomenon of hidden stratification [44], that are included intentionally, are not off-topic, and although they may be revealed in the same search, they require different treatment. We achieve off-topic sample ranking by agglomerative clustering with single linkage [45] in representation space. The idea is that the later a cluster is merged with a larger one, the more it can be considered an outlier [46]. The ranking is obtained by sorting the clustering dendrogram such that, at each merge, the elements of the cluster with fewer leaves appear first. We also associate each sample with a numerical score, which takes small values for abnormal instances and is compatible with the described ranking. In appendix J, we construct such a score $s_{\text{OT}}(\boldsymbol{e}_i)$ starting from the idea that merges, which happen at very different distances or between clusters of very different sizes, should produce large numerical variations.

**Near duplicates.** We define near duplicates as pairs of images that contain different views of the same object. In this sense, exact duplicates are a special case of near duplicates. We rank potential near duplicates by sorting each pair of distinct samples $(i, j), i < j$ in ascending order according to the distance between their representations in the latent space, $s_{\text{ND}}(\boldsymbol{e}_i, \boldsymbol{e}_j) = \text{dist}(\boldsymbol{e}_i, \boldsymbol{e}_j)$.

**Label errors.** We define label errors as samples annotated with a wrong class label. We rank potential label errors by sorting samples in ascending order according to their intra-/extra- class distance ratio [47]. For an anchor point $\boldsymbol{e}_i$, this ratio compares the distances to the nearest representation of a different label $m_{\neq}(\boldsymbol{e}_i)$ and the distance to the nearest representation of the same label $m_{=}(\boldsymbol{e}_i)$:

$$
\begin{aligned}
m_{=}(\boldsymbol{e}_i) &= \min_{j \in \mathcal{I},\, l_j = l_i} \big[ \text{dist}(\boldsymbol{e}_i, \boldsymbol{e}_j) \big], \\
m_{\neq}(\boldsymbol{e}_i) &= \min_{j \in \mathcal{I},\, l_j \neq l_i} \big[ \text{dist}(\boldsymbol{e}_i, \boldsymbol{e}_j) \big],
\end{aligned}
\qquad
s_{\text{LE}}(\boldsymbol{e}_i) = \frac{m_{\neq}^2(\boldsymbol{e}_i)}{m_{=}^2(\boldsymbol{e}_i) + m_{\neq}^2(\boldsymbol{e}_i)}. \tag{1}
$$

In all three cases, SELFCLEAN leverages the local structure of the embedding space: Cluster distances are computed only using the closest samples during agglomeration for off-topic samples, near duplicates are identified among sample pairs with the smallest distances, and label errors are found using only the nearest examples of the same and a different class.

## 3.3  Operation modes

The criteria above rank and score candidate issues, but do not specify which ones are inferred to be actual issues. This can be achieved with two operating modes: Human-in-the-loop or fully automatic.

**Human-in-the-loop.** This mode leverages candidate issue rankings to facilitate human confirmation which is often infeasible exhaustively, especially when considering pairwise relationships such as near duplicates. A human curator inspects a data sequence where issues tend to appear earlier, either confirming and correcting problems or looking for a specific rank threshold that gives the desired balance between precision and recall. In appendix H, we estimate that for a typical dataset SELFCLEAN reduces this inspection effort by a factor between 5 and 50 depending on issue type and baseline.

**Fully-automatic.** To perform automatic cleaning, specifying a fraction of data quality issues *a priori* is suboptimal, as contamination is not easy to estimate. The scores of section 3.2 empirically produce a smooth distribution for clean samples and relegate contaminated ones to significantly lower values. Depending on the contaminated data distribution, it may then be possible to isolate problematic samples with statistical arguments based on two robust hyperparameters, the contamination rate guess $\alpha$

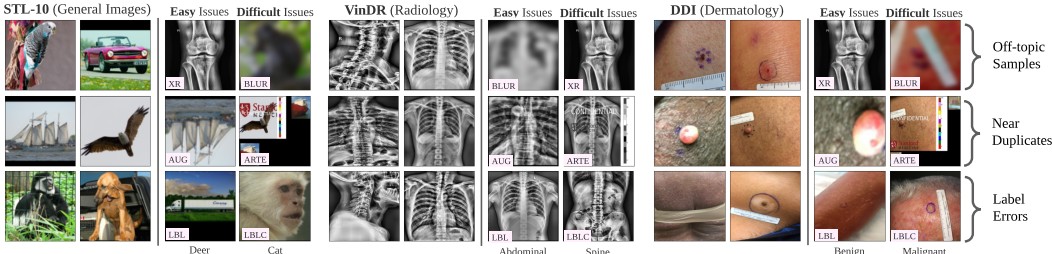

Figure 2: Illustration of synthetic data quality issues of all three types in STL-10, VinDR, and DDI.

and the significance level $q$, as detailed in appendix K. In short, we first use a logit transformation to induce a gap between scores of normal and problematic samples. We then set an upper bound for the left tail of the score distribution using a logistic functional form, and estimate its parameters using quantiles. Afterward, we identify issues based on their violation of the upper probability bound.

## 4    Experimental setup

**Datasets.** We experiment on a total of twelve datasets described in appendix D. These are four large-scale vision benchmarks: ImageNet [18], STL-10 [48], CelebA [49], and Food-101N [50], three general medical datasets of X-rays and histopathological images: CheXpert [51], VinDr-BodyPartXR [52], and PatchCamelyon [53], and five dermatology datasets: HAM10000 [54], ISIC-2019 [54], Fitzpatrick17k [55], DDI [56], and PAD-UFES-20 [57].

**Evaluation metrics.** The evaluation in this work relies on ranking metrics, as ranking constitutes the core of SELFCLEAN independently of the operation mode. All approaches are therefore evaluated in terms of the area under the receiver operating characteristic curve (AUROC) and average precision (AP) following standard practice [58]. AUROC measures the likelihood that a random relevant sample is ranked higher than a random irrelevant sample. AP measures precision across all values of recall, and is therefore sensitive to the proportion of positive and negative samples.

**Synthetic experiment setup.** To compare SELFCLEAN against other methods, we create synthetic datasets by altering benchmarks of different modalities (i.e., STL-10, VinDr-BodyPartXR, and DDI), as illustrated in figure 2. These synthetic contaminations are inspired by typical issues present in the respective dataset domains. We consider 5% and 10% contamination to mimic real-world noise prevalence estimates [59]. For each issue type, we compare against other unsupervised methods that have performed well on the given task. A detailed description of these competing approaches can be found in appendix E. Since SELFCLEAN learns representations on the contaminated dataset, we train a separate encoder for every issue type, contamination level, and synthetic contamination strategy.

The first synthetic contamination strategy for off-topic samples, *XR*, adds images from the "other" category of VinDr-BodyPartXR [52], which shows scans of lower limbs and device malfunctions. The second strategy for off-topic samples, *BLUR*, corrupts images with strong Gaussian blurring to simulate badly out-of-focus pictures. The first contamination strategy for near duplicates, *AUG*, adds samples from the original dataset after augmenting them with rotation, flipping, resizing, padding, and blurring. The second approach for near duplicates, *ARTE*, adds samples from the original dataset after including artifacts such as watermarks, color bars, and rulers, followed by scaling and composition with other images to create a collage. For label errors, the first contamination strategy, *LBL*, randomly changes a fraction of the labels choosing uniformly from incorrect ones. The second strategy to evaluate label errors, *LBLC*, randomly changes a fraction of the labels choosing incorrect ones proportionally to class prevalence in the original dataset. Depending on which dataset these strategies are applied to, they produce either easy or difficult problematic samples.

Different contamination strategies can be applied sequentially to create a dataset with a more realistic constellation of artificial data quality issues, resulting in a mixed-contamination strategy. In order to consider all interactions, we start by adding off-topic samples, proceed by creating near duplicates, and finally introduce label errors. To preserve the overall contamination rate $C$, each contamination in the sequence is added with prevalence $C_S$ such that $(1 + C_S)^S = (1 + C)$, where $S$ is the number of contamination steps.

Table 1: Performance in detecting synthetic data quality issues. Evaluation is performed for each of the three considered issue types across three benchmark datasets, augmented with two strategies for 5% synthetic contamination each, as illustrated in figure 2. Consult section 4 for more details on the contamination, and appendix E for details on competing approaches. Results are given in percentages (%).

| | Method | Rep. | STL + XR | | STL + BLUR | | VDR + BLUR | | VDR + XR | | DDI + XR | | DDI + BLUR | |
|---|---|---|---|---|---|---|---|---|---|---|---|---|---|---|
| | | | AUROC | AP | AUROC | AP | AUROC | AP | AUROC | AP | AUROC | AP | AUROC | AP |
| Off-topic Samples | HBOS [60] | INet | 66.9 | 6.6 | 1.9 | 2.6 | 95.7 | 36.6 | 82.3 | 24.4 | 93.0 | 68.0 | 19.0 | 3.0 |
| | ECOD [61] | INet | 68.4 | 7.0 | 2.2 | 2.6 | 95.0 | 34.1 | 81.4 | 25.7 | 92.8 | 68.0 | 23.6 | 3.1 |
| | SELFCLEAN | INet | 11.4 | 2.7 | 67.7 | 7.3 | 99.9 | 91.2 | 77.1 | 32.8 | 98.9 | 84.2 | 86.5 | 18.2 |
| | SELFCLEAN | SimCLR | 40.6 | 3.9 | 77.4 | 19.0 | **100.0** | **98.7** | 86.0 | 35.5 | 99.0 | 68.9 | 70.0 | 21.9 |
| | SELFCLEAN | DINO | **98.4** | **55.1** | **100.0** | **97.9** | **100.0** | **100.0** | **95.6** | **53.3** | **100.0** | **100.0** | **86.8** | **32.6** |

| | Method | Rep. | STL + AUG | | STL + ARTE | | VDR + AUG | | VDR + ARTE | | DDI + AUG | | DDI + ARTE | |
|---|---|---|---|---|---|---|---|---|---|---|---|---|---|---|
| | | | AUROC | AP | AUROC | AP | AUROC | AP | AUROC | AP | AUROC | AP | AUROC | AP |
| Near Duplicates | pHashing [62] | | 57.8 | $< 0.1$ | 73.1 | 20.1 | 47.5 | $< 0.1$ | 57.5 | 18.2 | 59.4 | 0.1 | 66.2 | 15.1 |
| | SSIM [63]. | | 62.5 | 0.2 | 83.6 | 19.9 | 46.3 | $< 0.1$ | 48.4 | **22.5** | 57.6 | 0.2 | 83.0 | 19.4 |
| | SELFCLEAN | INet | 96.6 | 7.6 | 96.5 | 15.2 | 79.7 | $< 0.1$ | 53.7 | 11.1 | 97.6 | 4.1 | 81.1 | 34.4 |
| | SELFCLEAN | SimCLR | 86.1 | 0.1 | 93.8 | 13.9 | 76.1 | $< 0.1$ | 78.9 | 12.6 | 89.8 | 0.7 | 87.2 | 0.7 |
| | SELFCLEAN | DINO | **100.0** | **43.7** | **99.9** | **48.0** | **98.5** | **0.4** | **91.6** | 16.8 | **99.7** | **50.8** | **98.2** | **48.2** |

| | Method | Rep. | STL + LBL | | STL + LBLC | | VDR + LBL | | VDR + LBLC | | DDI + LBL | | DDI + LBLC | |
|---|---|---|---|---|---|---|---|---|---|---|---|---|---|---|
| | | | AUROC | AP | AUROC | AP | AUROC | AP | AUROC | AP | AUROC | AP | AUROC | AP |
| Label Errors | CLearning [41] | INet | 86.2 | 41.6 | 83.2 | 36.8 | 96.7 | 79.0 | 96.8 | 74.9 | 67.9 | 11.0 | 75.0 | 12.9 |
| | FastDup [15] | INet | 87.5 | 20.5 | 87.0 | 19.8 | 95.0 | 38.9 | 94.1 | 37.8 | 69.0 | 8.6 | 69.9 | 11.6 |
| | SELFCLEAN | INet | **97.7** | **77.6** | **97.9** | **76.4** | 98.5 | 84.6 | 98.5 | 84.8 | 67.8 | 11.6 | **79.8** | 18.3 |
| | SELFCLEAN | SimCLR | 79.1 | 27.4 | 77.4 | 26.5 | 95.0 | 62.2 | 95.4 | 64.4 | 64.8 | 8.3 | 69.0 | 11.1 |
| | SELFCLEAN | DINO | 90.7 | 54.2 | 91.1 | 48.3 | **99.2** | **88.1** | **99.0** | **85.6** | **71.4** | **13.5** | 71.7 | **21.4** |

**Natural experiment setup.** We also evaluate cleaning on data quality issues naturally found in benchmark datasets. To this end, we devise two different experiments. In the first experiment, we measure how well the ranking matches available metadata, e.g., if two images show the same person or if the label was already identified as incorrect by prior work. In a second experiment, we use SELFCLEAN to propose a ranking for some datasets and evaluate it against human confirmation of issues with the statistical procedure outlined in appendix I.

## 5 Results

### 5.1 Synthetic contamination

**Comparison on data quality issues.** Table 1 displays the results of SELFCLEAN using either supervised ImageNet (INet), SimCLR, or DINO pre-training, and the two best competing methods per issue type. Performance is reported for 18 synthetic datasets based on general vision (STL), radiology (VDR), and dermatology (DDI) benchmarks described in section 4 with a contamination rate of 5%. Table 10 in appendix G.1 includes results for all competing approaches for both 5% and 10% contamination. SELFCLEAN with DINO pre-training outperforms all competing methods for off-topic-sample, near-duplicate, and label-error detection. Notably, some competing approaches for off-topic-sample detection show varying performance depending on the considered outlier type. In contrast, SELF-CLEAN does not show the same behavior, mainly because the dataset-specific pre-training captures the context of the task itself. SimCLR and supervised ImageNet features achieve mixed performance depending on the specific dataset and issue type. Lower performance of SimCLR is presumably caused by the small dataset size, as the batch size cannot be large enough, which is crucial for the contrastive approaches. AP for VDR with AUG is very low, likely because these synthetic issues are difficult in highly standardized settings and the dataset is not particularly clean, as further investigated in G.5.

**Influence of contamination.** Figure 3 illustrates the influence of the contamination on SELFCLEAN and the best two competing models. For approaches operating on features, we compare performance using both supervised INet and self-supervised, dataset-specific DINO training. Central value and error bars are obtained from three random initializations resulting in different synthetic datasets. This experiment is run on mixed-contamination datasets. SELFCLEAN outperforms competing approaches across contamination rates. The exception is off-topic detection for VDR with high contamination, where other indicator functions on dataset-specific SSL features perform marginally better. In general, dataset-specific image representations tend to outperform general-purpose ones across tasks. For

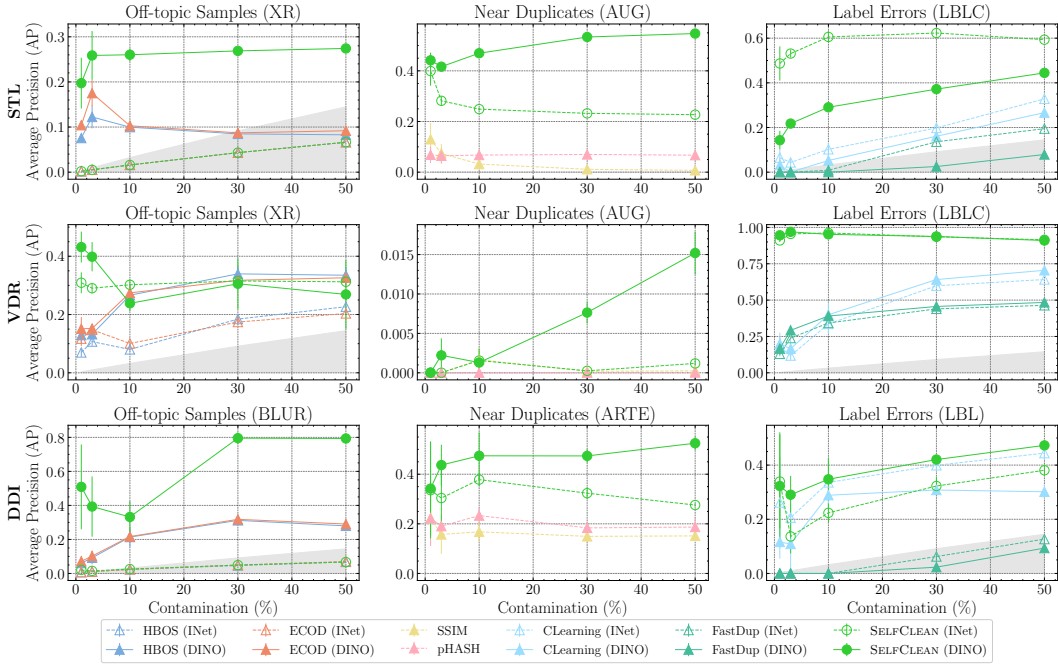

Figure 3: Performance of the best two approaches for each issue type to SELFCLEAN across different representations for a mixed-contamination strategy at varying contamination rates. Gray regions indicate random performance with an AP equal to the respective contamination $C_S$.

label error detection on STL, SELFCLEAN performs significantly better with INet features than with DINO features, presumably because INet features are trained with supervision on data and labels similar to STL.

## 5.2 Natural contamination

**Comparison with metadata.** We validate the label error ranking in a more realistic setting using annotations from the literature, such as 5,440 verified samples of ImageNet's validation set [7] and 57,608 of Food-101N [64]. SELFCLEAN achieves almost double the performance in AP for both datasets compared to other approaches, with 8.4% vs. 4.3% AP for ImageNet and 47.8% vs. 30.7% for Food-101N. We evaluate near-duplicate detection against CelebA labels that indicate images of the same celebrity. SELFCLEAN achieves 30.9% AP, demonstrating it effectively learned facial recognition without supervision. For medical datasets, we first check how well SELFCLEAN can find pairs of images showing the same skin lesion. We obtain good correspondence for HAM10000 and ISIC-2019, with an AP of 28.4% and 26.6%, respectively. On the other hand, for PAD-UFES-20 AP is only 10.0%, which we further investigate in appendix G.2 and is likely caused by inaccurate metadata. We also attempt to identify X-rays from the same patient within CheXpert and find only minor agreement with 7.5% AP, suggesting again that a case-by-case investigation should be performed. Overall, this shows that the rankings produced by SELFCLEAN align with existing metadata and considerably outperform competitors. A table with detailed results can be found in appendix G.2.

**Comparison with human annotators.** We evaluate SELFCLEAN rankings against human verification across two common vision and two medical benchmarks as described in appendix I. Human experts confirmed significantly more data quality issues in the top 50 images ranked by SELFCLEAN compared to 50 randomly sampled images, with 95% significance in nine out of twelve tasks (table 15). We repeat the comparison for images ranked 1-25 against images ranked 26-50 and observe significance for six out of ten evaluations. Two cases in the second comparison are excluded as only containing positive samples (i.e., data quality issues) results in undefined metrics. These results indicate that SELFCLEAN rankings align well with human assessment for these three issue types.

## 5.3 Influence of representation learning

Table 2 examines the influence of SSL objective, dataset, and augmentation on SELFCLEAN by measuring performance on STL. In the upper panel, we observe that dataset-specific representations (DINO STL) yield the best results for both off-topic and near duplicate detection, showcasing the strength of learning the dataset context. This is remarkable considering that STL has only 5,000 samples compared to the 1 million available for ImageNet. Label error detection seems instead to benefit from the larger data volume of ImageNet. However, this amount of data is not always available with so little domain shift, and dataset-specific representations strike a good trade-off. The lower panel investigates the influence of augmentation during pre-training. For DINO, removing color and size or multi-crop augmentations, the model loses its ability to reliably detect some issue types, in particular off-topic samples. For SimCLR, adding multi-crop substantially improves data cleaning performance. Interestingly, adding color and size augmentations alongside multi-crop seems to have a negative influence on near-duplicate detection, while isolating off-topic samples and label errors well.

Table 2: Ablation of pre-training choices in SELF-CLEAN. The upper part investigates SSL objective and dataset, and the lower the influence of SSL augmentations. For the different variants (lower part), we highlight the differences from the default setting. We use a 10% mixed-contamination dataset starting from STL and creating off-topic samples (OT) using XR, near duplicates (ND) using AUG, and label errors (LE) using LBLC. Performance is reported in average precision (AP).

| Pre-training strategy | OT (%) | ND (%) | LE (%) |
|---|---|---|---|
| SELFCLEAN (Sup. INet) | 1.6 | 24.6 | 63.0 |
| SELFCLEAN (DINO INet) | 13.7 | 6.1 | **69.5** |
| SELFCLEAN (DINO STL) | **27.4** | **47.1** | 24.8 |

| Color+Size | Multi-Crop | OT (%) | ND (%) | LE (%) |
|---|---|---|---|---|
| SELFCLEAN (DINO) | | | | |
| ✓ | ✓ | **27.4** | 47.1 | 24.8 |
| ✓ | ✗ | 2.8 | 17.5 | **39.6** |
| ✗ | ✓ | 4.2 | **67.2** | 12.6 |
| SELFCLEAN (SimCLR) | | | | |
| ✓ | ✓ | **39.1** | 12.8 | **18.1** |
| ✓ | ✗ | 26.1 | 12.1 | 15.8 |
| ✗ | ✓ | 3.9 | **21.9** | 11.7 |

In appendix F, we further demonstrate that it is important to pre-train for sufficient epochs and to either normalize embeddings or use the cosine distance. We also find that DINO works best among four SSL objectives and investigate the effectiveness of different backbones. Finally, we show that label-error detection deteriorates with label granularity, but SELFCLEAN stays on par with other methods.

## 6 Discussion

**Application to benchmark datasets.** We apply the fully automatic mode of SELFCLEAN to well-known image benchmark datasets and estimate the prevalence of data quality issues. For the estimation, we used conservative guesses of a contamination rate of $\alpha = 0.10$ and a significance level of $q = 0.05$. Detailed results can be found in appendix L.1. For highly curated datasets with extensive manual verification, such as DDI, PAD-UFES-20, HAM10000, CheXpert, and ImageNet-1k, we find noise levels below 1%. However, for ISIC-2019 and PatchCamelyon, we estimate 5.4% and 3.9% of near duplicates that are not accounted for in the metadata. When considering datasets with less manual curation, such as Fitzpatrick17k, CelebA, and Food-101N, we find less than 1% of off-topic samples and label errors, and approximately $14.8\%$, $0.4\%$, and $1.4\%$ near duplicates, respectively. The abundance of near duplicates in these benchmarks can often be traced back to crawling data of different pages using the same illustration or thumbnail images. When data splits with near-duplicate data leaks are used, performance estimates on these datasets are optimistically biased.

**Influence of dataset cleaning.** In table 3 we examine the impact of cleaning data quality issues to better understand their relevance. We train linear and $k$NN classifiers based on dataset-specific SSL representations for multiple classification benchmarks and measure the performance difference in F1 score when removing the problematic samples found above, first from the evaluation set and then also from the training set. For most benchmark datasets, cleaning the evaluation set significantly alters scores. Variations are either positive or negative depending on whether wrong samples were misclassified, and larger for datasets with significant data leaks. Cleaning the training set has a significant positive impact for many benchmarks, indicating that issues in the training set hindered optimization.

Table 3: Influence of removing samples detected in the automatic cleaning mode with $\alpha = 0.10$ and $q = 0.05$ on downstream tasks. We report macro-averaged F1 scores for linear and $k$NN classifiers on DINO features over 100 random training/evaluation splits with 80% and 20% fractions, respectively. We compute paired performance differences before and after cleaning the evaluation set, and before and after cleaning also the training set. We report the median and the intervals to the 5% (subscript) and 95% (superscript) percentiles. Additionally, we indicate significance of a paired permutation test on the difference sign with $^*p < 0.05$, $^{**}p < 0.01$, and $^{***}p < 0.001$.

| | $\Delta$ $k$NN Classifier (%pt.) | | $\Delta$ Linear Classifier (%pt.) | |
| Dataset | Clean Eval | Clean Train | Clean Eval | Clean Train |
|---|---|---|---|---|
| DDI | $+1.2^{+1.9}_{-1.2}$ *** | $+0.0^{+1.7}_{-1.4}$ *** | $+1.0^{+11.1}_{-11.2}$ | $-0.7^{+7.7}_{-10.8}$ |
| HAM10000 | $+0.2^{+0.5}_{-0.4}$ *** | $+0.2^{+1.3}_{-0.8}$ ** | $+0.1^{+3.2}_{-3.5}$ | $-0.1^{+3.9}_{-3.6}$ |
| Fitzpatrick17k | $-4.1^{+1.2}_{-1.3}$ *** | $+0.1^{+2.0}_{-1.7}$ | $-0.6^{+2.9}_{-3.6}$ ** | $+0.2^{+3.3}_{-3.9}$ * |
| ImageNet-1k | $-0.4^{+0.1}_{-0.2}$ *** | $+0.4^{+0.3}_{-0.4}$ *** | $-0.4^{+0.6}_{-0.6}$ *** | $-0.0^{+0.9}_{-0.5}$ |
| Food-101N | $+0.1^{+0.1}_{-0.1}$ *** | $+0.1^{+0.2}_{-0.2}$ *** | $+0.2^{+0.6}_{-0.5}$ *** | $+0.1^{+0.6}_{-0.5}$ ** |

The importance of each individual data quality issue type depends on the dataset and task, and identifying trends by domain and modality requires further investigation. For the limited number of cases in Table 3, and taking into account Table 16, data leaks caused by near duplicates across splits seem to have the highest impact, followed by label errors. However, we argue that information on off-topic samples and near duplicates within the same data split is always valuable, even if it only serves the purpose of restoring trust.

**Recommended use.** SELFCLEAN determines context based on the dataset rather than a specific task, so the candidates it provides for correction may represent desired features (e.g., rare diseases or longitudinal data). The identification of a data quality issue should not be automatically considered a suggestion to remove it. Instead, discovering relationships among samples is always an advantage, as it can inform proper action. While undesirable behavior may occur with the automatic mode, this is similar to other cleaning methods applied without checks, and such biases can be mitigated with the human-in-the-loop approach.

The tension between correcting data quality issues and the veto against the examination of evaluation data, mentioned in the introduction, has no easy resolution. We suggest the following compromise as an improvement to the current practice. A benchmark dataset should be refined using an SSL model developed on the training set. SELFCLEAN can be used to clean both training and evaluation sets, but for the latter the human-in-the-loop mode is required, and labels should not be altered. The number of problems found for each set separately and across them for near duplicates should be reported. Even with human confirmation and refraining from correcting label errors, the cleaning procedure introduces some degree of bias due to the sampling of the candidate issues to be confirmed. We believe that in many practical cases, the benefit of data cleaning outweighs this bias.

## 7    Conclusion and outlook

We found a data-cleaning strategy called SELFCLEAN, based on dataset-specific self-supervised learning and local, distance-based indicator functions, to be effective for detecting off-topic samples, near duplicates, and label errors. We demonstrated this by comparing to state-of-the-art methods across multiple general vision and medical image benchmarks both with synthetic issues and with natural contamination. SELFCLEAN outperformed competing approaches for synthetic data quality issues, and demonstrated superior correspondence to metadata and expert verification in natural settings. Notably, the detailed methodology surpassed the state-of-the-art in label-error detection, achieving a twofold increase in AP over existing approaches on known ImageNet-1k and Food-101N issues. Moreover, applying the cleaning strategy to highly curated medical datasets and general vision benchmarks revealed multiple data quality issues with significant impact on model scores. By correcting these data collections, confidence can be regained in reported benchmark performances. In the future, we plan to incorporate SELFCLEAN during annotation to collect higher quality datasets and during inference to enhance model robustness.

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

# Appendix

## Table of Contents

## A   Broader impact

SELFCLEAN is a new data-cleaning procedure that can be applied to any visual data collection. The procedure relies on SSL and, therefore, does not inherit annotation bias. Practitioners can choose if they want the cleaning process to happen fully automatically or with human intervention. Gaining insights into data collections of unknown quality can lead to the curation of more reliable benchmarks, which in turn result in performance measurements that are more accurate generalization estimates. Moreover, reported benchmark results can be questioned if they contain substantial contamination. SELFCLEAN thus significantly contributes to clarifying which methods are the most valuable and to steering future research directions both in academia and applied innovation.

The near-duplicate detection component of SELFCLEAN could be used for person re-identification and data de-anonymization, even if it was not designed for this purpose. Although new in peer-reviewed publications for data cleaning to the best of our knowledge, this method can already be found in at least one publicly available tool [14]. We believe that the benefits of increased awareness outweigh the increased chances of malignant use.

## B   Limitations

SELFCLEAN hinges on the considered dataset and inherits biases from its intrinsic composition. For example, given an image collection with a minority group that can be easily distinguished from the rest, the minority samples may be suggested to be off-topic. This risk is studied in section G.3, where we find no evidence for this behavior for multiple datasets such as DDI, Fitzpatrick17k, and CheXpert.

From a computational perspective, the current formulation of near-duplicate detection does not scale well with dataset size. This could be improved with approximation methods or by relying on an iterative analysis of nearest-neighbor distances. Also, the detailed methodology requires SSL pre-training on the dataset in order to clean it, which requires sufficient computational power and might be a limiting factor to some. However, training on the considered dataset is required by other methods such as confident learning [41], although this training is supervised and requires labeled data. In contrast, SELFCLEAN does not require any annotations during training.

Currently, there is no standard protocol for evaluating data cleaning frameworks. To address this, we designed synthetic experiments that simulate data quality issues. The datasets used for evaluation are, however, already contaminated (see section L), which means that performance measures are capped. However, since all approaches are subject to the same conditions, we expect only minor interference in their comparison.

While several mechanisms that produce data quality issues were considered (such as longitudinal studies, watermarks, blurring, and different resolutions for near duplicates), exhaustively exploring all possibilities is unfeasible. It is likely that in some scenarios SELFCLEAN can fail. A hint of this behavior can be seen in section G.5.

Finally, we acknowledge that certain data quality issue types such as ambiguous labels were not investigated in this work. Likewise, limited investigation was carried out on how to remedy identified issues, as this is expected to strongly depend on dataset and task.

## C   Training details

We use a randomly initialized ViT-tiny [65] with a patch size of $16 \times 16$ as encoder unless otherwise specified. The latent representation is given by the class token output from the encoder's last layer, which has dimension 192 for a ViT-tiny.

Table 4 lists the hyperparameters used for pre-training with DINO [43] and SimCLR [42]. Parameter values are similar to the introductory papers of these approaches [42, 43] with the exception that for DINO the global crop scale is larger and we sample more local crops, which we have found to be beneficial for smaller datasets ($<$20,000) while we observed no benefit or harm for larger datasets. All SSL models were pre-trained for 500 epochs with only minor manual hyperparameter tuning to ensure proper convergence. We resize images to $224 \times 224$ pixels and normalize them using the mean and standard deviation of ImageNet [18].

For the synthetic experiment setup, table 5 lists the hyperparameters for producing near-duplicate images. The configuration was chosen to mimic the natural contamination of near duplicates in benchmark datasets.

Table 4: Hyperparameters used for pre-training using SimCLR and DINO on the dataset to clean. Here "-" indicates that the respective parameter is not used for the corresponding pre-training strategy. Parameters not given in this list follow the introductory paper. More detailed information about the hyperparameters can be found in the open-sourced codebase.

| Hyperparameter | SimCLR [42] | DINO [43] |
|---|---|---|
| Batch size | 90 | 64 |
| Epochs | 500 | 500 |
| Optimizer | Adam | AdamW |
| Learning rate | 0.001 | 0.0005 |
| Min. learning rate | 1e-6 | 1e-6 |
| Weight decay | 0.04 | 0.04 |
| Weight decay end | 0.4 | 0.4 |
| Warmup epochs | 10 | 10 |
| Momentum teacher | - | 0.996 |
| Clip grad. | 3.0 | 3.0 |
| Base model | ViT-tiny | ViT-tiny |
| Model embedding dim. | 192 | 192 |
| Model output dim. | 128 | 4096 |
| Model patch size | 16 | 16 |
| Model drop path rate | 0.1 | 0.1 |
| Norm last layer | - | True |
| Global crops scale | - | (0.7, 1.) |
| Local crops scale | - | (0.05, 0.4) |
| Global crops number | - | 2 |
| Local crops number | - | 12 |
| Warmup teacher temp. | - | 0.04 |
| Teacher temp. | - | 0.04 |
| Warmup teacher temp. epochs | - | 30 |
| Contrastive temp. | 0.5 | - |

Table 5: Configuration of the synthetic near duplicate strategies AUG (5a) and ARTE (5b).

(a) AUG

| Hyperparameter | AUG |
|---|---|
| Rotation probability | 0.5 |
| Padding probability | 0.5 |
| Blur probability | 0.5 |
| Rotation degree range | (0, 180) |
| Scale range | (0.5, 0.9) |
| Padding | 3 |
| Gaussian kernel size | 5 |

(b) ARTE

| Hyperparameter | ARTE |
|---|---|
| Watermark probability | 0.5 |
| Colorbar probability | 0.5 |
| Collage probability | 0.5 |
| Watermark max. scale | 0.5 |
| Collage max. scale | 0.5 |
| Reference size | 512 |

The implementation of SELFCLEAN and the code used for evaluation are based on PyTorch 1.9 [66] and can be found at

https://github.com/Digital-Dermatology/SelfClean, and
https://github.com/Digital-Dermatology/SelfClean-Evaluation.

Experiments were performed on an Nvidia DGX station, which features eight V100 GPUs, each with 32 GB of memory, 512 GB of system memory, and 40 CPU cores, for a total of 10,800 GPU hours which roughly correspond to 1,200 kg $CO_2$.

# D   Datasets

In this study, we selected twelve well-known, open-source image datasets comprising four general-purpose vision benchmarks and eight medical datasets. These datasets contain different modalities, such as smartphone, X-ray, histopathology, dermatoscopy, and clinical images. The diversity of the datasets and domains should illustrate SELFCLEAN's versatility. Furthermore, some datasets were chosen because of their high-quality standards, as their curation involved extensive manual correction, including validation by multiple domain experts.

## D.1    Datasets from the general image domain

**ImageNet-1k (INet)** is a well-known image benchmark with 1,000 classes [18]. Images were scraped by querying words from WordNet's "synonym sets" (synsets) on several image search engines. The images were labeled by Amazon Mechanical Turk workers, who were asked whether each image contained objects of a given synset. License: Custom (research, non-commercial).

**STL-10 (STL)** is a benchmark consisting of 10 classes, each with 500 training images, 800 test images, and an additional 100,000 unlabeled images for unsupervised learning [48]. It focuses on higher resolution images (96x96 pixels) compared to other similar collections like CIFAR-10. The images in STL-10 are sourced from labeled examples in ImageNet and are chosen to represent a broad range of object categories and real-world scenarios. License: Custom (attribution + ImageNet license).

**Food-101N** is an image dataset that contains 310,009 images of food divided into 101 classes [64]. Both Food-101N and the Food-101 [50] dataset share the same 101 classes. However, Food-101N has a significantly larger number of images and contains more noise. The pictures were scraped from Google, Bing, Yelp, and TripAdvisor. 60,000 of them were manually verified and used for evaluation. The evaluation set includes information for each sample on whether or not it features a label problem. License: CC BY 4.0.

**CelebFaces Attributes Dataset (CelebA)** is a large-scale dataset with 202,599 celebrity face images, each with 40 attribute annotations [49]. The images in this dataset cover 10,177 identities, large pose variations and mixed backgrounds. The CelebA dataset contains images of public figures, and while it is widely used in research, it is important to consider privacy, consent, and potential biases [67]. We have ensured that our usage complies with the dataset's terms and conditions, and we advise caution to avoid perpetuating any biases inherent in the dataset. Our work does not involve any manipulation or generation of images that could misrepresent individuals. License: Custom (research, non-commercial).

## D.2    Datasets from the medical domain

**CheXpert** is a large public dataset for chest radiograph interpretation, consisting of 224,316 X-ray scans from 65,240 patients [51]. The authors retrospectively collected chest radiographic examinations from Stanford Hospital, performed between October 2002 and July 2017 in both inpatient and outpatient centers, along with their associated radiology reports. Labels were extracted from the free-text radiology reports with an automated rule-based system. The dataset further contains radiologist-labeled reference evaluation sets. License: Stanford University School of Medicine's Research Use Agreement.

**VinDr-BodyPartXR (VDR)** consists of 16,093 X-ray images that were manually annotated for body part classification [52]. The authors differentiate between five groups, including abdominal, adult chest, pediatric chest, spine, and other X-rays. The "other" category contains X-rays of any other body part, device malfunctions, and scans of clinical tools. License: CC BY-NC 4.0.

**PatchCamelyon** consists of 327,680 color image patches extracted from histopathologic scans of lymph node sections [68] from the Camelyon16 dataset [53]. Each patch is annotated with a binary label indicating the presence of metastatic tissue. Camelyon16 contains 399 whole-slide images and corresponding glass slides of sentinel axillary lymph nodes, which were retrospectively sampled from 399 patients who underwent breast cancer surgery at two hospitals in the Netherlands. All metastases in the slides were annotated under the supervision of multiple expert pathologists. License: CC0.

**Diverse Dermatology Images (DDI)** is a public, deeply-curated, and pathologically-confirmed image dataset with diverse skin tones [56]. It contains 656 clinical images of 570 unique patients with 78 common and uncommon diseases originating from pathology reports of the Stanford Clinics. License: Stanford University School of Medicine's Research Use Agreement.

**PAD-UFES-20** is a public benchmark dataset composed of clinical images collected from smartphone devices including patient clinical data [57]. The dataset comprises 1,373 patients, 1,641 skin lesions, and 2,298 images for six different diagnoses: three skin diseases and three skin cancers. License: CC BY 4.0.

**HAM10000** is a public benchmark dataset consisting of 10,015 dermatoscopic images collected from different populations and institutions [54]. The collected cases include a representative sample of seven categories of pigmented lesions. License: CC BY-NC.

**Fitzpatrick17k (FST)** is a public benchmark dataset containing 16,577 clinical images with skin condition annotations and skin type labels based on the Fitzpatrick scoring system [55]. The images originate from two online dermatology atlases and thus are known to contain issues [6]. In this study, we used the middle granularity level, which partitions the labels into nine disease categories. License: CC BY-NC-SA 3.0.

**High-Quality Fitzpatrick17k (HQ-FST)** is a subset of the Fitzpatrick17k dataset used in the paper [55] as a data quality check. It was obtained by randomly selecting 3% of the images (504 samples) and gathering

annotations by two board-certified dermatologists to evaluate diagnostic accuracy. This subset is assumed to be of much higher quality than its original, larger counterpart. License: CC BY-NC-SA 3.0.

**ISIC-2019** is a public benchmark dataset of 25,331 dermoscopic images with metadata split into eight diagnostic categories. Additionally, the test set contains an additional outlier class not represented in the training data. The images originate from the HAM10000 [54] and the BCN_20000 [69] datasets. License: CC BY-NC-SA 3.0.

# E    Competing approaches

We selected different competing approaches to detect each of the three data quality issue categories, i.e., off-topic samples, near duplicates, and label errors. Some of these methods require to encode images in a low-dimensional latent space. For this projection, we used a ViT-tiny, the same architecture used for SELFCLEAN, pre-trained with supervision on ImageNet or with DINO self-supervision on each dataset. We refer to these encoders with "(INet)" and "(DINO)" respectively, after the name of each detection approach. In this section, we briefly summarize each competing approach used in this work.

## E.1    Approaches for off-topic samples

**Isolation Forest (IForest)** isolates observations by randomly selecting a feature and splitting the value between the minimum and maximum of the selected feature. The number of splits required to isolate a sample corresponds to the path length from the root node to the leaf node in a tree [70]. This path length averaged over a forest of random trees is a measure of normality, where noticeably shorter paths are produced for anomalies.

**Histogram-based outlier detection (HBOS)** is an efficient unsupervised method that creates a histogram of the feature vector for each dimension and then calculates a score based on how likely a particular data point is to fall within the histogram bins for each dimension [60]. The higher the score, the more likely the data point is an outlier, i.e., a feature vector coming from an anomaly will occupy unlikely bins in one or several of its dimensions and thus produce a higher anomaly score.

**Empirical Cumulative Distribution Functions (ECOD)** is a parameter-free, highly-interpretable unsupervised outlier detection algorithm [61]. It estimates an empirical cumulative distribution function (ECDF) for each variable in the data separately. To generate an outlier score for an observation, it computes the tail probability for each variable using the univariate ECDFs and multiplies them together. This calculation is done in log space, accounting for each dimension's left and right tails.

## E.2    Approaches for near duplicates

**Perceptual Hash (pHashing)** is a type of locality-sensitive hash, which is similar if features of the sample are similar [62]. It relies on the discrete cosine transform (DCT) for dimensionality reduction and produces hash bits depending on whether each DCT value is above or below the average value. In this paper, we use pHash with a hash size of 8.

**Structural Similarity Index Measure (SSIM)** is a type of similarity measure to compare two images with each other based on three features, namely luminance, contrast, and structure [63]. Instead of applying SSIM globally, i.e., all over the image at once, one usually applies the metrics regionally, i.e., in small sections of the image, and takes the mean overall. This variant of SSIM is often called "Mean Structural Similarity Index". In this paper, we apply SSIM locally to 8x8 windows but still refer to the method as SSIM for simplicity.

## E.3    Approaches for label errors

**Confident Learning (CLearning)** is a data-centric approach that focuses on label quality by characterizing and identifying label errors in datasets based on the principles of pruning noisy data, counting with probabilistic thresholds to estimate noise, and ranking examples to train with confidence [41]. It builds upon the assumption of a class-conditional noise process to directly estimate the joint distribution between noisy (given) and uncorrupted (unknown) labels, resulting in a generalized learning process that is provably consistent and experimentally performant. In this study, we use AdaBoost [71] as a classifier on top of pre-trained representations to estimate probabilities. We did not observe any significant performance difference when using different classifiers similarly to Northcutt et al. [41].

**NoiseRank (Noise)** is a method for unsupervised label noise detection using Markov Random Fields [72]. It constructs a dependence model to estimate the posterior probability of an instance being incorrectly labeled, given the dataset, and then ranks instances based on this probability.

### E.4 Approaches for multiple issue types

**FastDup** is an open-source, non-peer-reviewed tool designed to rapidly extract valuable insights from image and video datasets, aiming to increase the dataset quality and reduce data operations costs at an unparalleled scale [15]. It detects outliers, duplicate, and near-duplicate images and videos, and wrongly labeled samples.

# F Further ablation studies

This section presents additional ablation studies that investigate different components of SELFCLEAN. Note that we cannot consistently use the same dataset for these ablation studies, as each ablation is most meaningful for a dataset with a specific domain and degree of cleanliness, also in relationship with the considered issue type and amount of required compute.

## F.1 Influence of $L_2$-normalization and distance functions

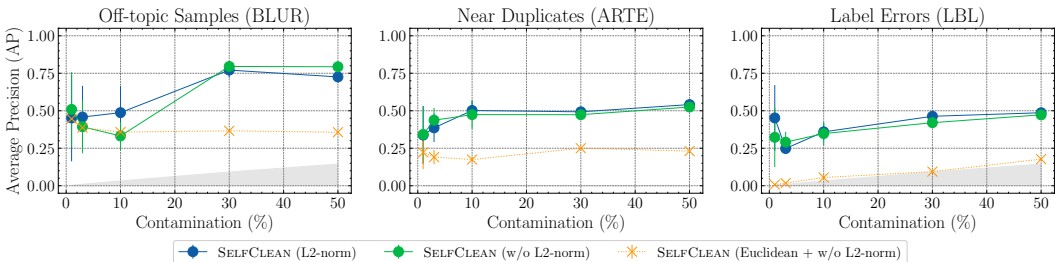

Figure 4: Performance of SELFCLEAN when changing the distance function and removing the $L_2$-normalization. The performance is measured in terms of average precision (AP) for a mixed-contamination strategy when varying the contamination rate. The artificial dataset is created from DDI by adding off-topic samples (BLUR), then injecting augmented duplicates (ARTE), and finally changing labels at random (LBL). Shaded regions indicate random performance.

For SSL strategies without explicit normalization, we included $L_2$-normalization in the latent space during both training and inference (e.g., DINO). A similar explicit $L_2$-normalization for representation layers is also used in theoretical works on SSL [73], where it was inherited from the neural collapse literature [74]. We investigate the influence of this $L_2$-normalization on the detection performance for the different dataset quality issues. Figure 4 shows the performance of SELFCLEAN with and without normalization. The experiment is run on a 10% mixed-contamination dataset, starting from DDI and creating off-topic samples using BLUR, near duplicates using ARTE, and label errors using LBL. The results show that $L_2$-normalization has a mild, slightly positive effect on the performance. One possible explanation for the improved performance is that limiting the latent space to the unit hypersphere enforces a more direct relation between the training objective and the relative distances of encoded samples.

Additionally, we examined the influence of the choice of the distance function between cosine and Euclidean distance. Since the Euclidean and cosine distance on a $L_2$-normalized space always produce the same ranking, we only show the results of different distance functions for the non-normalized latent space. Figure 4 shows that performance is strongly influenced by the choice of distance function. Specifically, using Euclidean distance leads to significantly lower performance.

## F.2 Influence of the number of pre-training epochs

We evaluate the learned representations after a different number of pre-training epochs to investigate the influence of the pre-training length. The experiment is run on a 10% mixed-contamination dataset, starting from DDI and creating off-topic samples using BLUR, near duplicates using ARTE, and label errors using LBL. The performance of the representations is evaluated every 50 epochs for both representations with and without $L_2$-normalization.

Figure 5 shows that performance for off-topic sample and near duplicate detection increases with longer pre-training with $L_2$-normalization. Without normalization, the performance for off-topic detection has no clear trend. For label error detection, both methods first degrade slightly and later stabilize. Overall, at least with $L_2$-normalization, longer pre-training leads to stronger performance.

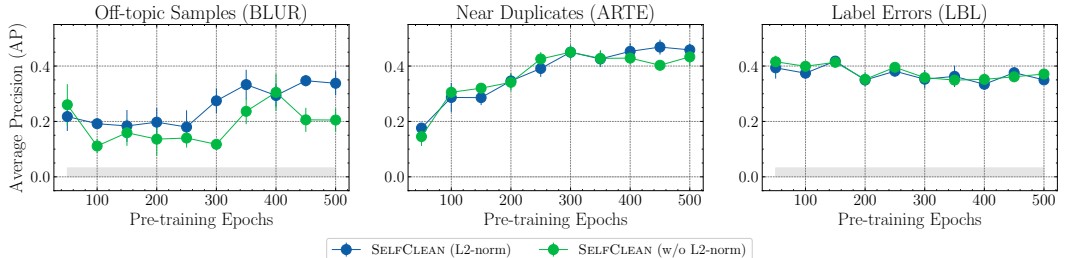

Figure 5: Performance of SELFCLEAN during pre-training. The performance is measured in terms of average precision (AP) for a 10% mixed-contamination strategy. The artificial dataset is created from DDI by adding off-topic samples (BLUR), then injecting augmented duplicates (ARTE), and finally changing labels at random (LBL). Shaded regions indicate random performance.

## F.3 Influence of label granularity

We investigate the performance of label error detection on different label granularities using the high-quality Fitzpatrick17k dataset. This dataset features three hierarchy levels with 3, 9, and 104 classes respectively, and it has only around 500 samples, which makes the task difficult. In table 6 we report results for synthetic label issues (i.e., LBL and LBLC) for 10% contamination. Overall, it is harder to detect label errors as granularity increases, in agreement with intuition. We observe that SELFCLEAN excels at coarse granularity, and performs similarly to other approaches for fine-grained classification.

Table 6: Performance of models on the detection of label errors. Evaluation is performed for each of the two synthetic label error strategies across HQ-FST with three different label partitions. All scores are reported in percentages (%).

| | Method | Rep. | 3-Partition | | | | 9-Partition | | | | 104-Partition | | | |
|---|---|---|---|---|---|---|---|---|---|---|---|---|---|---|
| | | | **LBL** | | **LBLC** | | **LBL** | | **LBLC** | | **LBL** | | **LBLC** | |
| | | | AUROC | AP | AUROC | AP | AUROC | AP | AUROC | AP | AUROC | AP | AUROC | AP |
| Label Errors | CLearning [41] | INet | 79.7 | 30.7 | 80.2 | 27.5 | **84.6** | **40.4** | 69.0 | 26.2 | 46.2 | 11.5 | 51.7 | 23.2 |
| | NoiseRank [72] | INet | 50.8 | 10.4 | 57.0 | 12.1 | 53.4 | 17.7 | 60.4 | 23.8 | 65.1 | **34.9** | 58.6 | 35.6 |
| | FastDup [15] | INet | 64.8 | 14.6 | 74.1 | 21.4 | 70.2 | 18.6 | 68.3 | 21.1 | 52.6 | 12.4 | 55.8 | 20.2 |
| | SELFCLEAN | INet | 76.8 | 28.7 | 79.2 | 29.9 | 81.2 | 30.3 | 78.3 | 31.9 | 67.1 | 20.8 | **73.4** | **44.3** |
| | SELFCLEAN | DINO | 80.9 | 35.4 | 85.3 | 40.0 | 80.3 | 28.3 | **78.7** | 32.8 | 68.8 | 19.2 | 70.5 | 39.5 |

## F.4 Influence of the type of features

We investigate the influence of different types of features, i.e., general, domain-specific, and dataset-specific features. The experiment is run on a 10% mixed-contamination dataset, starting from VDR and DDI. Table 7 shows that for VDR, domain-specific features have the strongest overall performance followed by dataset-specific features. General supervised and self-supervised features both fail at near duplicates even if they show strong performance on label errors. For DDI, dataset-specific features yield the best performance, followed by domain-specific, self-supervised general, and supervised general. For both datasets, these results show the importance of learning representations that successfully capture the task's context in order to achieve good detection performance.

## F.5 Influence of the self-supervised learning objective

We investigate further SSL objectives for detecting data quality issues. In addition to SimCLR and DINO, which are used throughout the paper, we include BYOL [75] and MAE [76]. The experiment is run on a 10% mixed-contamination dataset, starting from STL and creating off-topic samples using XR, near duplicates using AUG, and label errors using LBLC. Table 8 shows that DINO has the strongest overall performance. Some SSL objectives only obtain strong results for specific issue types. This is the case for BYOL, which separates off-topic samples well but fails on near duplicates and label errors. Other methods, such as MAE and SimCLR, achieve similar performance across issue types, although significantly lower than DINO.

## F.6 Influence of the encoder architecture

We investigate further encoder architectures in table 9. In addition to the ViT-tiny with a patch size of $16 \times 16$ used throughout the paper, we include larger and different types of architectures, i.e., ViTs [65] and ResNets [77].

Table 7: Ablation of the feature types, i.e., general, domain-specific, and dataset-specific features. We use a different 10% mixed contaminated dataset starting from VDR and DDI. Scores are average precision (AP) percentages, and aggregated across the three tasks using the harmonic mean.

| Dataset | Pre-training | OT (%) | ND (%) | LE (%) | H. Mean |
|---|---|---|---|---|---|
| *VDR (XR+AUG+LBLC)* | | | | | |
| INet | Supervised | 28.5 | < 0.1 | **95.8** | < 0.1 |
| INet | DINO | 28.7 | < 0.1 | 95.5 | < 0.1 |
| CheXpert | DINO | **31.3** | **0.3** | 94.8 | **0.9** |
| VDR | DINO | 23.3 | 0.1 | 95.0 | 0.3 |
| *DDI (BLUR+ARTE+LBL)* | | | | | |
| INet | Supervised | 2.6 | 37.8 | 22.4 | 6.6 |
| INet | DINO | 18.6 | 28.8 | **36.7** | 25.9 |
| HAM10000 | DINO | 29.1 | 27.1 | 31.2 | 29.0 |
| DDI | DINO | **33.2** | **47.4** | 34.8 | **37.5** |

Table 8: Ablation of the pre-training strategy. We use a 10% mixed contaminated dataset starting from STL and creating off-topic samples (OT) using XR, near duplicates (ND) using AUG, and label errors (LE) using LBLC. Scores are average precision (AP) percentages, and aggregated across the three tasks using the harmonic mean.

| Pre-training | Dataset | OT (%) | ND (%) | LE (%) | H. Mean |
|---|---|---|---|---|---|
| SimCLR [42] | STL | 26.1 | 12.1 | 15.8 | 16.3 |
| BYOL [75] | STL | **29.7** | < 0.1 | 3.5 | < 0.1 |
| MAE [76] | STL | 8.3 | 18.1 | 17.7 | 12.9 |
| DINO [43] | STL | 27.4 | **47.1** | **24.8** | **30.6** |

The experiment is run on a 10% mixed-contamination dataset, starting from STL and creating off-topic samples using XR, near duplicates using AUG, and label errors using LBLC. Results indicate that the smaller models (i.e., ViT-tiny and ResNet-18) produce stable results with DINO pre-training, although ViTs show overall superior performance, similar as found in [43]. Label error detection works best with supervised training as already observed in section 5.3, presumably because ImageNet and STL have very similar contexts. Furthermore, for label errors, performance with supervised pre-training increases with model size. Larger models (i.e., ViT-small and ResNet-50) show mixed results, likely because of the small pre-training dataset of 5,000 samples.

Table 9: Ablation of the encoder architecture, i.e., ViT [65] and ResNet [77]. We use a 10% mixed contaminated dataset starting from STL and creating off-topic samples (OT) using XR, near duplicates (ND) using AUG, and label errors (LE) using LBLC. Scores are average precision (AP) percentages, and aggregated across the three tasks using the harmonic mean.

| Encoder | N.o. Parameters | Pre-training | Dataset | OT (%) | ND (%) | LE (%) | H. Mean |
|---|---|---|---|---|---|---|---|
| ViT-tiny $16 \times 16$ | 5.5 Mio. | Supervised | INet | 1.6 | 24.6 | **63.0** | 4.4 |
| | | DINO | STL | **27.4** | **47.1** | 24.8 | **30.6** |
| ResNet-18 | 11.7 Mio. | Supervised | INet | 4.6 | 4.4 | **94.8** | 6.6 |
| | | DINO | STL | **14.8** | **22.9** | 30.1 | **20.8** |
| ViT-small $16 \times 16$ | 21.7 Mio. | Supervised | INet | **2.4** | **20.9** | 94.5 | **6.3** |
| | | DINO | STL | 1.8 | 20.7 | 44.0 | 4.8 |
| ResNet-50 | 25.6 Mio. | Supervised | INet | **10.1** | 1.7 | 96.5 | 4.3 |
| | | DINO | STL | 4.1 | **25.1** | 69.1 | **10.1** |

# G  Detailed dataset cleaning results

This section provides extended tables with performance results related to dataset cleaning. More precisely, section G.1 investigates synthetic contamination detection with different methods, metrics, and contamination levels, expanding on section 5.1. Section G.2 presents in tabular form the comparison of SELFCLEAN with available metadata as discussed in section 5.2. Section G.4 extends table 3 in section 6 by including information on the performances used to compute paired differences.

## G.1  Detailed comparison on synthetic data quality issues

Table 10 details results of the comparison of synthetic data quality issues. Conclusions are drawn in section 5.1.

Table 10: Performance of various models on the detection of synthetic data quality issues. Evaluation is performed for each of the three considered issue types across three benchmark datasets, STL, VDR, and DDI, augmented with different strategies for synthetic contamination (XR, BLUR, AUG, ARTE, LBL, and LBLC). All scores are reported in percentages (%).

### Contamination 5%

**Off-topic Samples**

| Method | Rep. | STL + XR AUROC | AP | STL + BLUR AUROC | AP | VDR + BLUR AUROC | AP | VDR + XR AUROC | AP | DDI + XR AUROC | AP | DDI + BLUR AUROC | AP |
|---|---|---|---|---|---|---|---|---|---|---|---|---|---|
| IForest [70] | INet | 68.2 | 7.0 | 1.6 | 2.6 | 94.0 | 31.1 | 81.2 | 22.6 | 93.3 | 59.5 | 21.5 | 3.1 |
| HBOS [60] | INet | 66.9 | 6.6 | 1.9 | 2.6 | 95.7 | 36.6 | 82.3 | 24.4 | 93.0 | 68.0 | 19.0 | 3.0 |
| ECOD [61] | INet | 68.4 | 7.0 | 2.2 | 2.6 | 95.0 | 34.1 | 81.4 | 25.7 | 92.8 | 68.0 | 23.6 | 3.1 |
| FastDup [15] | INet | 4.1 | 2.5 | 8.3 | 2.6 | 25.1 | 7.7 | 69.6 | 20.0 | 53.5 | 29.5 | 19.7 | 3.1 |
| SELFCLEAN | INet | 11.4 | 2.7 | 67.7 | 7.3 | 99.9 | 91.2 | 77.1 | 32.8 | 98.9 | 84.2 | 86.5 | 18.2 |
| SELFCLEAN | SimCLR | 40.6 | 3.9 | 77.4 | 19.0 | 100.0 | 98.7 | 86.0 | 35.5 | 99.0 | 68.9 | 70.0 | 21.9 |
| SELFCLEAN | DINO | 98.4 | 55.1 | 100.0 | 97.9 | 100.0 | 100.0 | 95.6 | 53.3 | 100.0 | 100.0 | 86.8 | 32.6 |

**Near Duplicates**

| Method | Rep. | STL + AUG AUROC | AP | STL + ARTE AUROC | AP | VDR + AUG AUROC | AP | VDR + ARTE AUROC | AP | DDI + AUG AUROC | AP | DDI + ARTE AUROC | AP |
|---|---|---|---|---|---|---|---|---|---|---|---|---|---|
| pHashing [62] | | 57.8 | < 0.1 | 73.1 | 20.1 | 47.5 | < 0.1 | 57.5 | 18.2 | 59.4 | 0.1 | 66.2 | 15.1 |
| SSIM [63]. | | 62.5 | 0.2 | 83.6 | 19.9 | 46.3 | < 0.1 | 48.4 | 22.5 | 57.6 | 0.2 | 83.0 | 19.4 |
| FastDup [15] | INet | 50.2 | 2.2 | 49.2 | 3.3 | 37.6 | < 0.1 | 40.1 | 2.9 | 56.2 | 4.8 | 44.6 | 7.1 |
| SELFCLEAN | INet | 96.6 | 7.6 | 96.5 | 15.2 | 79.7 | < 0.1 | 53.7 | 11.1 | 97.6 | 4.1 | 81.1 | 34.4 |
| SELFCLEAN | SimCLR | 86.1 | 0.1 | 93.8 | 13.9 | 76.1 | < 0.1 | 78.9 | 12.6 | 89.8 | 1.6 | 87.2 | 0.7 |
| SELFCLEAN | DINO | 100.0 | 43.7 | 99.9 | 48.0 | 98.5 | 0.4 | 91.6 | 16.8 | 99.7 | 50.8 | 98.2 | 48.2 |

**Label Errors**

| Method | Rep. | STL + LBL AUROC | AP | STL + LBLC AUROC | AP | VDR + LBL AUROC | AP | VDR + LBLC AUROC | AP | DDI + LBL AUROC | AP | DDI + LBLC AUROC | AP |
|---|---|---|---|---|---|---|---|---|---|---|---|---|---|
| CLearning [41] | INet | 86.2 | 41.6 | 83.2 | 36.8 | 96.7 | 79.0 | 96.8 | 74.9 | 67.9 | 11.0 | 75.0 | 12.9 |
| NoiseRank [72] | INet | 49.5 | 5.0 | 51.4 | 5.4 | 48.9 | 5.3 | 51.8 | 5.3 | 51.4 | 5.8 | 52.0 | 6.1 |
| FastDup [15] | INet | 87.5 | 20.5 | 87.0 | 19.8 | 95.0 | 38.9 | 94.1 | 37.8 | 69.0 | 8.6 | 69.9 | 11.6 |
| SELFCLEAN | INet | 97.7 | 77.6 | 97.9 | 76.4 | 98.5 | 84.6 | 98.5 | 84.8 | 67.8 | 11.6 | 79.8 | 18.3 |
| SELFCLEAN | SimCLR | 79.1 | 27.4 | 77.4 | 26.5 | 95.0 | 62.2 | 95.4 | 64.4 | 64.8 | 8.3 | 69.0 | 11.1 |
| SELFCLEAN | DINO | 90.7 | 54.2 | 91.1 | 48.3 | 99.2 | 88.1 | 99.0 | 85.6 | 71.4 | 13.5 | 71.7 | 21.4 |

### Contamination 10%

**Off-topic Samples**

| Method | Rep. | STL + XR AUROC | AP | STL + BLUR AUROC | AP | VDR + BLUR AUROC | AP | VDR + XR AUROC | AP | DDI + XR AUROC | AP | DDI + BLUR AUROC | AP |
|---|---|---|---|---|---|---|---|---|---|---|---|---|---|
| IForest [70] | INet | 45.5 | 7.6 | 1.6 | 5.2 | 81.4 | 21.9 | 75.3 | 30.6 | 91.6 | 61.5 | 12.3 | 5.6 |
| HBOS [60] | INet | 48.7 | 7.9 | 1.2 | 5.2 | 87.6 | 28.8 | 81.0 | 35.4 | 95.8 | 78.0 | 13.5 | 5.7 |
| ECOD [61] | INet | 52.6 | 8.6 | 1.6 | 5.2 | 89.1 | 32.5 | 79.4 | 33.6 | 95.1 | 76.7 | 19.3 | 5.9 |
| FastDup [15] | INet | 2.9 | 4.7 | 7.9 | 5.4 | 22.1 | 7.7 | 64.8 | 24.2 | 31.7 | 22.5 | 14.5 | 5.9 |
| SELFCLEAN | INet | 0.7 | 4.7 | 47.3 | 9.1 | 99.8 | 93.3 | 74.0 | 38.8 | 94.6 | 72.8 | 82.3 | 22.4 |
| SELFCLEAN | SimCLR | 36.0 | 7.1 | 75.4 | 46.3 | 99.8 | 96.4 | 86.5 | 43.5 | 96.4 | 58.6 | 80.4 | 32.2 |
| SELFCLEAN | DINO | 97.6 | 60.8 | 100.0 | 97.8 | 100.0 | 100.0 | 96.9 | 62.1 | 100.0 | 100.0 | 88.4 | 55.4 |

**Near Duplicates**

| Method | Rep. | STL + AUG AUROC | AP | STL + ARTE AUROC | AP | VDR + AUG AUROC | AP | VDR + ARTE AUROC | AP | DDI + AUG AUROC | AP | DDI + ARTE AUROC | AP |
|---|---|---|---|---|---|---|---|---|---|---|---|---|---|
| pHashing [62] | | 53.9 | 0.1 | 73.2 | 22.5 | 46.2 | < 0.1 | 56.9 | 19.3 | 56.5 | 0.5 | 72.6 | 25.5 |
| SSIM [63]. | | 62.8 | 0.2 | 83.8 | 22.5 | 49.4 | 0.1 | 50.2 | 22.3 | 57.3 | 0.9 | 80.6 | 26.3 |
| FastDup [15] | INet | 54.5 | 3.3 | 54.9 | 5.5 | 37.1 | < 0.1 | 44.5 | 3.9 | 58.8 | 3.3 | 54.7 | 4.9 |
| SELFCLEAN | INet | 96.2 | 17.9 | 96.8 | 17.9 | 80.9 | < 0.1 | 54.9 | 13.4 | 98.2 | 12.5 | 82.0 | 25.9 |
| SELFCLEAN | SimCLR | 81.7 | 0.1 | 93.5 | 10.3 | 70.4 | < 0.1 | 77.5 | 12.9 | 89.0 | 0.3 | 61.1 | < 0.1 |
| SELFCLEAN | DINO | 100.0 | 51.0 | 99.9 | 46.4 | 98.7 | 0.3 | 88.5 | 14.3 | 99.3 | 49.0 | 97.4 | 49.8 |

**Label Errors**

| Method | Rep. | STL + LBL AUROC | AP | STL + LBLC AUROC | AP | VDR + LBL AUROC | AP | VDR + LBLC AUROC | AP | DDI + LBL AUROC | AP | DDI + LBLC AUROC | AP |
|---|---|---|---|---|---|---|---|---|---|---|---|---|---|
| CLearning [41] | INet | 83.5 | 47.9 | 85.0 | 46.4 | 97.4 | 85.1 | 97.4 | 84.5 | 73.7 | 25.6 | 72.5 | 24.9 |
| NoiseRank [72] | INet | 49.4 | 10.0 | 50.0 | 10.3 | 51.5 | 10.5 | 51.5 | 10.8 | 51.7 | 11.1 | 50.3 | 10.4 |
| FastDup [15] | INet | 2.9 | 4.7 | 0.3 | 5.2 | 3.3 | 6.0 | 64.8 | 24.2 | 31.7 | 22.5 | 9.3 | 5.5 |
| SELFCLEAN | INet | 97.1 | 80.0 | 96.8 | 80.0 | 96.2 | 81.6 | 97.1 | 83.0 | 70.6 | 24.5 | 77.3 | 28.3 |
| SELFCLEAN | SimCLR | 73.3 | 27.6 | 74.7 | 31.6 | 91.8 | 61.6 | 92.3 | 65.2 | 68.3 | 24.1 | 68.5 | 22.1 |
| SELFCLEAN | DINO | 89.5 | 57.6 | 89.0 | 56.5 | 97.5 | 84.1 | 97.8 | 86.3 | 75.9 | 27.6 | 78.3 | 29.1 |

## G.2 Detailed comparison with metadata

Table 11 details the comparison of the SELFCLEAN ranking with metadata from multiple benchmark datasets, as discussed in section 5.2.

For PAD-UFES-20, we investigated SELFCLEAN's relatively low performance, as discussed in "Comparision with metadata" of section 5.2. We provided the first 200 near-duplicate candidates of PAD-UFES-20 to three practicing dermatologists and asked them to verify whether the given samples were near duplicates. The experts reached a good inter-annotator agreement with Krippendorff's alpha $> 0.6$. Of the samples they unanimously agreed to be near duplicates (56 samples), 32% had faulty metadata where the lesion ID was not correctly maintained. Thus, we find evidence that the poor alignment of SELFCLEAN and the metadata of PAD-UFES-20 is likely caused by imperfect metadata.

Table 11: Comparison of SELFCLEAN and competitor rankings with metadata from multiple benchmark datasets. We include the proportion of positive samples, which corresponds to the baseline AP. Consult section 5.2 for interpretation.

| Dataset | Metadata | Positive Samples (%) | Method | Rep. | AUROC (%) | AP (%) |
|---|---|---|---|---|---|---|
| PAD-UFES-20 | Same Lesion | 0.06 | pHashing [62] | | 56.6 | 0.2 |
| | | | SSIM [63] | | 63.7 | 0.3 |
| | | | SELFCLEAN | DINO | **71.0** | **10.0** |
| HAM10000 | Same Lesion | 0.01 | pHashing [62] | | n.a.[4] | n.a.[4] |
| | | | SSIM [63] | | n.a.[4] | n.a.[4] |
| | | | SELFCLEAN | DINO | **98.7** | **28.4** |
| ISIC-2019 | Same Lesion | 0.01 | pHashing [62] | | n.a.[4] | n.a.[4] |
| | | | SSIM [63] | | n.a.[4] | n.a.[4] |
| | | | SELFCLEAN | DINO | **98.2** | **26.6** |
| CheXpert | Same Patient | 0.01 | pHashing [62] | | n.a.[4] | n.a.[4] |
| | | | SSIM [63] | | n.a.[4] | n.a.[4] |
| | | | SELFCLEAN | DINO | **70.5** | **7.5** |
| CelebA | Same Person | 0.02 | pHashing [62] | | n.a.[4] | n.a.[4] |
| | | | SSIM [63] | | n.a.[4] | n.a.[4] |
| | | | SELFCLEAN | DINO | **78.8** | **30.9** |
| ImageNet-1k | Verified Label Errors[2] | 4.38 | CLearning [41] | INet | 46.6 | 4.3 |
| | | | FastDup [15] | INet | 42.6 | 3.6 |
| | | | SELFCLEAN | DINO | **67.7** | **8.7** |
| Food-101N | Verified Label Errors[3] | 18.51 | CLearning [41] | INet | 61.0 | 25.2 |
| | | | FastDup [15] | INet | 72.1 | 30.7 |
| | | | SELFCLEAN | DINO | **79.8** | **47.8** |
| **Subsampled results[4]** | | | | | | |
| HAM10000 | Same Lesion | 0.01 | pHashing [62] | | 71.3 [69.9, 74.2] | 2.7 [2.6, 7.6] |
| | | | SSIM [63] | | 67.3 [66.3, 72.4] | 7.7 [4.4, 7.8] |
| | | | SELFCLEAN | DINO | **98.7 [97.9, 99.0]** | **30.0 [23.9, 34.3]** |
| ISIC-2019 | Same Lesion | 0.01 | pHashing [62] | | 62.3 [58.5, 63.4] | 0.1 [$< 0.1$, 2.1] |
| | | | SSIM [63] | | 69.1 [66.3, 70.0] | 1.3 [0.3, 1.6] |
| | | | SELFCLEAN | DINO | **98.9 [97.4, 98.9]** | **28.6 [26.6, 29.2]** |
| CheXpert | Same Patient | 0.01 | pHashing [62] | | 54.7 [53.8, 57.0] | 0.2 [0.1, 0.4] |
| | | | SSIM [63] | | 65.7 [64.7, 66.1] | 0.2 [0.2, 0.3] |
| | | | SELFCLEAN | DINO | **86.5 [85.5, 88.1]** | **1.9 [0.3, 2.3]** |
| CelebA | Same Person | 0.02 | pHashing [62] | | 53.3 [52.8, 54.7] | $< 0.1$ [$< 0.1$, $< 0.1$] |
| | | | SSIM [63] | | 56.3 [55.3, 58.3] | $< 0.1$ [$< 0.1$, $< 0.1$] |
| | | | SELFCLEAN | DINO | **81.0 [80.6, 81.2]** | **0.6 [0.6, 0.6]** |

---

[2] Refers to the subset of ImageNet-1k validation set which was verified by Northcutt et al. [7].

[3] Refers to the subset of Food-101N set which was verified by Lee et al. [64].

[4] As the number of near duplicates for comparison exceeds memory limitations for the baseline methods (as indicated by "n.a." in the upper panel), they were subsampled three times with the same percentage of positive samples to 2,000 samples (i.e., 1,999,000 comparisons). We report the median and the min-max variation in brackets.

## G.3 Potential bias of off-topic ranking

There is a chance that off-topic sample detection may exacerbate data distribution biases because underrepresented samples are more likely to be proposed as candidate issues. Therefore, we investigate if some specific dataset attributes correlate with the off-topic sample ranking, assessing, for example, if pigment-rich skin is more often suggested to be off-topic. For this experiment, we focus on the demographics of CheXpert and skin types in DDI and Fitzpatrick17k. We compare the ranking of the feature attribute using AP and AUROC, similar to the comparison with metadata in appendix G.2. The results show no evidence of an increased likelihood of underrepresented groups appearing earlier in the ranking, as AUROC stays around 50% and AP is similar to the non-informed baseline, i.e., the percentage of samples belonging to the group.

Table 12: Comparison of the SELFCLEAN ranking with various demographic attributes. For reference, we include the prevalence of each group, also corresponding to the not-informed baseline performing best in terms of AP.

| Dataset | Attribute | Value | Prevalence (%) | AUROC (%) | AP (%) |
|---------|-----------|-------|----------------|-----------|--------|
| DDI | Skin Tone | Fitzpatrick Type 3&4 | 36.7 | 46.8 | 35.4 |
| | | Fitzpatrick Type 1&2 | 31.7 | 52.5 | 31.2 |
| | | Fitzpatrick Type 5&6 | 31.6 | 50.9 | 35.9 |
| Fitzpatrick17k | Skin Tone | Fitzpatrick Type 2 | 29.0 | 53.2 | 31.1 |
| | | Fitzpatrick Type 3 | 20.0 | 47.5 | 19.1 |
| | | Fitzpatrick Type 1 | 17.8 | 52.8 | 18.9 |
| | | Fitzpatrick Type 4 | 16.8 | 45.4 | 15.2 |
| | | Fitzpatrick Type 5 | 9.2 | 46.3 | 8.5 |
| | | Fitzpatrick Type 6 | 3.8 | 50.8 | 3.8 |
| | | Fitzpatrick Type Unknown | 3.4 | 57.5 | 4.3 |
| CheXpert | Ethnicity | Non-Hispanic/Non-Latino | 72.9 | 50.0 | 72.8 |
| | | Unknown | 14.2 | 53.3 | 15.4 |
| | | Hispanic/Latino | 12.1 | 46.5 | 11.1 |
| | | Patient Refused | 0.3 | 43.5 | 0.2 |
| | | Not Hispanic | < 0.1 | 35.1 | < 0.1 |
| | | Hispanic | < 0.1 | 9.1 | < 0.1 |
| CheXpert | Gender | Male | 55.2 | 43.5 | 51.4 |
| | | Female | 44.3 | 56.4 | 48.5 |
| | | Unknown | < 0.1 | 17.7 | < 0.1 |
| CheXpert | Primary Race | White | 45.5 | 47.7 | 44.0 |
| | | Other | 12.9 | 46.4 | 11.9 |
| | | White, non-Hispanic | 10.0 | 55.7 | 11.5 |
| | | Asian | 9.5 | 51.7 | 9.7 |
| | | Unknown | 6.6 | 52.5 | 7.1 |
| | | Black or African American | 4.0 | 47.4 | 3.8 |
| | | Race and Ethnicity Unknown | 3.9 | 53.8 | 4.3 |
| | | Other, Hispanic | 1.7 | 49.6 | 1.6 |
| | | Asian, non-Hispanic | 1.2 | 56.0 | 1.4 |
| | | Native Hawaiian or Other Pacific Islander | 1.2 | 44.1 | 1.0 |
| | | Black, non-Hispanic | 0.8 | 55.5 | 1.0 |
| | | White, Hispanic | 0.5 | 53.7 | 0.6 |
| | | Other, non-Hispanic | 0.3 | 56.4 | 0.4 |
| | | Patient Refused | 0.2 | 44.4 | 0.2 |
| | | American Indian or Alaska Native | 0.2 | 46.5 | 0.2 |
| | | Pacific Islander, non-Hispanic | 0.1 | 51.4 | 0.2 |
| | | Native American, non-Hispanic | < 0.1 | 60.7 | < 0.1 |
| | | Black, Hispanic | < 0.1 | 60.7 | < 0.1 |
| | | Native American, Hispanic | < 0.1 | 60.2 | < 0.1 |
| | | Asian, Hispanic | < 0.1 | 63.4 | < 0.1 |
| | | White or Caucasian | < 0.1 | 18.9 | < 0.1 |
| | | Pacific Islander, Hispanic | < 0.1 | 65.1 | 0.2 |
| | | Asian - Historical Conv | < 0.1 | 58.7 | < 0.1 |

## G.4 Detailed influence of dataset cleaning

Table 13 complements table 3 with score ranges before differences. Conclusions are drawn in section 6.

Table 13: Influence of removing samples detected in the automatic cleaning mode with $\alpha = 0.10$ and $q = 0.05$ on downstream tasks. We report macro-averaged F1 scores for linear and $k$NN classifiers on DINO features over 100 random training/evaluation splits with 80% and 20% fractions respectively. We compute paired performance differences before and after cleaning the evaluation set, and before and after cleaning also the training set. We report the median and the intervals to the 5% (subscript) and 95% (superscript) percentiles. Additionally, we indicate significance of a paired permutation test on the difference sign with $^{*}p < 0.05$, $^{**}p < 0.01$, and $^{***}p < 0.001$.

| | kNN Classifier | | | | |
| | Scores (%) | | | Differences (%pt.) | |
| | Cont + Cont | Cont + Clean | Clean + Clean | Clean Eval | Clean Train |
|---|---|---|---|---|---|
| DDI | $58.2^{+7.7}_{-8.3}$ | $59.2^{+7.5}_{-8.3}$ | $59.7^{+7.3}_{-8.8}$ | $+1.2^{+1.9}_{-1.2}$ *** | $+0.0^{+1.7}_{-1.4}$ *** |
| HAM10000 | $58.3^{+3.4}_{-4.9}$ | $58.3^{+3.7}_{-4.7}$ | $58.7^{+3.1}_{-4.6}$ | $+0.2^{+0.5}_{-0.4}$ *** | $+0.2^{+1.3}_{-0.8}$ ** |
| Fitzpatrick17k | $60.2^{+1.8}_{-1.9}$ | $56.1^{+1.9}_{-2.2}$ | $56.1^{+2.0}_{-2.3}$ | $-4.1^{+1.2}_{-1.3}$ *** | $+0.1^{+2.0}_{-1.7}$ |
| Food-101N | $40.3^{+0.8}_{-0.9}$ | $40.4^{+0.7}_{-1.1}$ | $40.5^{+0.7}_{-1.1}$ | $+0.1^{+0.1}_{-0.1}$ *** | $+0.1^{+0.2}_{-0.2}$ *** |
| ImageNet-1k | $31.2^{+0.8}_{-0.9}$ | $30.8^{+0.9}_{-0.9}$ | $31.1^{+0.8}_{-0.9}$ | $-0.4^{+0.1}_{-0.2}$ *** | $+0.4^{+0.3}_{-0.4}$ *** |

| | Linear Classifier | | | | |
| **Dataset** | Cont + Cont | Cont + Clean | Clean + Clean | Clean Eval | Clean Train |
|---|---|---|---|---|---|
| DDI | $59.2^{+9.6}_{-10.2}$ | $59.6^{+12.0}_{-11.2}$ | $58.9^{+9.0}_{-9.7}$ | $+1.0^{+11.1}_{-11.2}$ | $-0.7^{+7.7}_{-10.8}$ |
| HAM10000 | $62.6^{+4.2}_{-4.2}$ | $63.0^{+3.3}_{-4.0}$ | $62.8^{+3.2}_{-3.8}$ | $+0.1^{+3.2}_{-3.5}$ | $-0.1^{+3.9}_{-3.6}$ |
| Fitzpatrick17k | $52.8^{+2.6}_{-3.1}$ | $52.5^{+2.5}_{-4.1}$ | $52.6^{+2.9}_{-2.8}$ | $-0.6^{+2.9}_{-3.6}$ ** | $+0.2^{+3.3}_{-3.9}$ * |
| Food-101N | $50.0^{+0.9}_{-1.2}$ | $50.1^{+1.1}_{-1.0}$ | $50.4^{+0.8}_{-1.2}$ | $+0.2^{+0.6}_{-0.5}$ *** | $+0.1^{+0.6}_{-0.5}$ ** |
| ImageNet-1k | $42.4^{+0.7}_{-0.9}$ | $42.0^{+0.9}_{-0.9}$ | $42.2^{+0.6}_{-1.0}$ | $-0.4^{+0.6}_{-0.6}$ *** | $-0.0^{+0.9}_{-0.5}$ |

## G.5 Investigation of VinDr-BodyPartXR near duplicates

In the synthetic experiments (see 5.1) we observe particularly low AP for synthetic near duplicates with AUG for VinDr-BodyPartXR (VDR). Here this discrepancy is further investigated. Figure 6a illustrates the top-10 near duplicate candidates for VDR without synthetic contamination. At least some of them are natural contamination that is not accounted for in the dataset's metadata, and others have highly standardized poses which may match more easily than synthetic contamination. Figure 6b shows the score distribution of the injected duplicates in comparison to the overall distribution and illustrates that they lie in the earlier parts of the ranking.

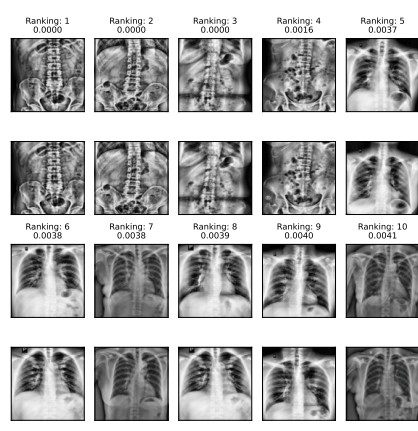

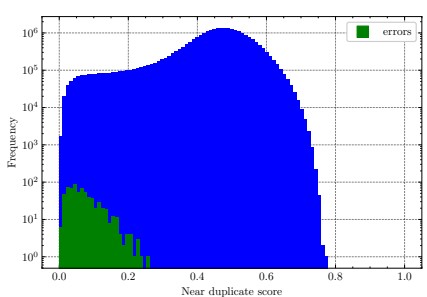

(a) Ranking produced by SELFCLEAN for near duplicates in the VinDr-BodyPartXR, of which the top-10 are shown along with the respective rank and score.

(b) Histogram of the scores for VinDr-BodyPartXR with injected near duplicates using AUG. The green distribution shows synthetic issues, and blue is the overall score distribution.

Figure 6: Investigation of VinDr-BodyPartXR near duplicates.

# H Inspection effort saved

When potential data quality issues are verified by a human, it is valuable to quantify the reduction in inspection effort achieved through the ranking. This reduction should be viewed as a function of the residual contamination that can be tolerated in the dataset, i.e., of the recall for data quality issues. We quantify effort using the number of inspections required rather than the actual time spent, as this is a good proxy more directly related to the ranking. Specifically, we calculate the fraction of effort (FE) needed to achieve a given recall by dividing the number of inspections required using the ranking by the number of inspections needed when candidate issues are sorted randomly. The comparison baseline is random sorting, which always requires confirming a number of examples equal to the target recall times the number of potential issues, due to the uniform density of actual issues in the sequence. The fraction of effort equals 1 when confirmation using the ranking is just as cumbersome as the baseline. It is $<1$ when the ranking is beneficial for cleaning, and $>1$ when it is detrimental. The best and worst cases possible are obtained by a ranking algorithm that sorts all positive samples first or last respectively. They obtain FEs equal to $\alpha_+$ and $[1-(1-R)\alpha_+]/R$, where $R$ is the recall and $\alpha_+$ is the contamination in the dataset, i.e., the number of actual data quality issues divided by the number of possible data quality issues. Note that the fraction of effort saved by a method compared to another can easily be obtained by dividing the two corresponding FEs.

To summarize the inspection effort savings in a single number, we compute the average fraction of effort (AFE) over all possible recalls, i.e., the area under the FE–R curve. To this end, we proceed as in the computation of average precision, and define

$$\text{AFE} = \sum_i (R_{i+1} - R_i)\text{FE}_i. \tag{2}$$

In table 14 we compare the best two competing approaches with SELFCLEAN on a 10% mixed-contamination dataset starting from STL in terms of AFE. In figure 7 we further plot the FE–R curves for all approaches. For competing methods which operate on extracted features, we compare performance using both supervised INet and self-supervised dataset-specific DINO features. For both off-topic samples and near duplicate detection, the AFE is significantly lower for SELFCLEAN than for its competitors, indicating a large amount of time and effort saved in using it. For label errors, SELFCLEAN with self-supervised dataset-specific representation leads to a similar AFE as competitors, which however may be aided by the similarity of ImageNet and STL in this specific case.

Table 14: Average fraction of effort (AFE) for the detection of synthetic data quality issues. Evaluation is performed on a 10% mixed-contamination dataset starting from STL and creating off-topic samples (OT) using XR, near duplicates (ND) using AUG, and label errors (LE) using LBLC.

| | Method | Rep. | ↑ AUROC (%) | ↑ AP (%) | ↓ AFE (%) |
|---|---|---|---|---|---|
| OT | HBOS [60] | INet | 0.6 | 1.6 | 508.4 |
| | ECOD [61] | INet | 0.7 | 1.6 | 518.4 |
| | SELFCLEAN | INet | 0.7 | 1.6 | 472.8 |
| | SELFCLEAN | DINO | **86.9** | **24.4** | **20.2** |
| | **Method** | **Rep.** | **↑ AUROC (%)** | **↑ AP (%)** | **↓ AFE (%)** |
| ND | pHashing [62] | | 72.1 | 6.1 | 37.0 |
| | SSIM [63] | | 74.7 | 2.0 | 32.8 |
| | SELFCLEAN | INet | 97.3 | 26.1 | 2.9 |
| | SELFCLEAN | DINO | **98.2** | **46.2** | **1.8** |
| | **Method** | **Rep.** | **↑ AUROC (%)** | **↑ AP (%)** | **↓ AFE (%)** |
| LE | CLearning [41] | INet | 76.6 | 6.9 | 33.2 |
| | FastDup [15] | INet | 88.1 | 0.4 | 24.3 |
| | SELFCLEAN | INet | **98.3** | **63.4** | **5.2** |
| | SELFCLEAN | DINO | 85.3 | 32.6 | 21.5 |

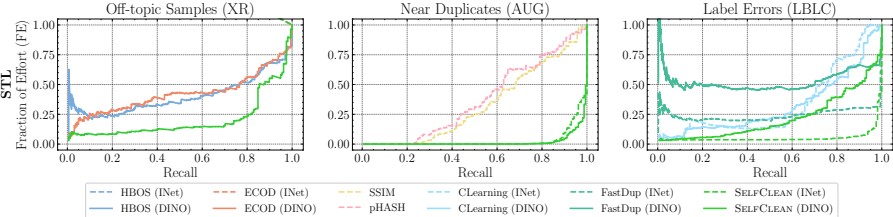

Figure 7: FE–R curves for a mixed-contamination strategy at 10% level. The artificial dataset is created from STL by adding X-ray images (XR), injecting augmented duplicates (AUG), and changing labels at random (LBLC). The closer the curves are to zero, the less effort is needed to find data quality issues.

# I Validating algorithmic rankings with humans

In this section, we describe the procedure used to confirm that, also according to human criteria, SELFCLEAN assigns low ranks to problematic samples and high ranks to normal data, as discussed in the second part of section 5.2. To this end, for each data quality issue type, we collect human verification for the first 50 images in the ranking and for 50 images randomly sampled from the dataset. Crowd workers use the respective platform's tool[5] for annotation, and medical expert annotators use a custom tool, which is shown in figure 8. The verification process starts with the selection of a dataset and data quality issue (e.g., the Fitzpatrick17k dataset and off-topic samples) and then proceeds with binary questions about single images or pairs thereof depending on the task. Section I.1 shows the task descriptions for each quality issue. Note that the samples' ranks are not displayed to avoid potential bias. Annotations were aggregated using majority voting for both crowd workers and medical experts. Medical experts agreed with an average Krippendorff's alpha of 0.52, 0.97, and 0.55 for off-topic samples, near duplicates, and label errors, respectively.

We paid crowd workers 0.03 US dollars per annotation for images from ImageNet and Food-101N, which roughly corresponds to 9 US dollars per hour. Medical experts were not compensated financially but were instead acknowledged with co-authorship in a labeling consortium.

During annotation, we solely collected answers as binary labels along with anonymized annotator identification. Thus, these annotations contain no personally identifiable information or offensive content. In discussion with experts from the institutions of the co-authors, it was concluded that this verification process does not require IRB approval because the conducted study examines publicly available datasets and does not involve human subjects beyond binary annotations.

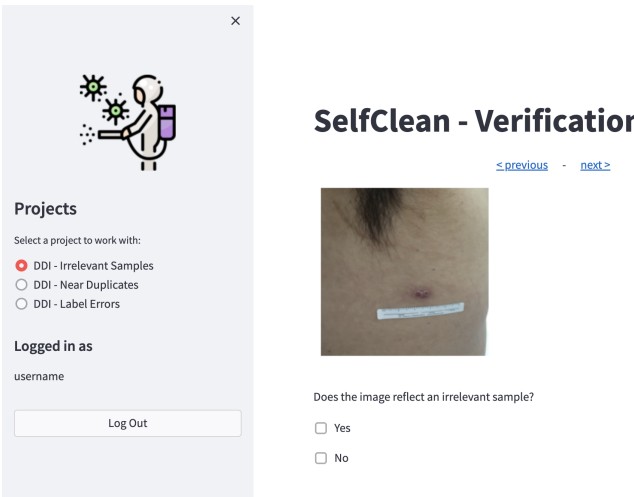

Figure 8: Screenshot of the verification tool used by medical experts to annotate data quality issues.

## I.1 Task descriptions

This section reports all task descriptions shown to the annotators:

- Off-topic samples: "Your task is to judge if the image shown is irrelevant. Select *yes* when the image is not a valid input for the task at hand."
- Near duplicates: "Your task is to judge whether the two images shown together are pictures of the same object. Note that pictures of the same object can be identical or different shots with the same object of interest."
- Label errors: "Your task is to judge whether the image's label is correct. Please select that the label is an error only if you think it is wrong and not when there is low uncertainty or ambiguity."

## I.2 Detailed results

In order to verify that problematic samples tend to appear first in the ranking provided by SELFCLEAN, for each issue type, we first consider the first 50 images in the ranking against the 50 random ones, and then the first

---

[5] `https://www.clickworker.com/`, accessed on the 28th of October.

group of 25 in the ranking against the second group of 25. We conduct one-sided Mann-Whitney $U$ statistical tests to verify that humans are more likely to identify data quality issues in samples that appear first in the SELFCLEAN ranking. In order to gain a more intuitive understanding, we also report the fraction of samples that were found to be problematic within the first 50 and the 50 random samples, and within samples ranked 1 through 25 and 26 through 50. Finally, we visualize the distribution of human-confirmed problems through the ranking by plotting the fraction of confirmed problems in a rolling window of ten ranks in figure 9.

We observe significant alignment for near-duplicate detection throughout the considered datasets. Label-error identification is significant in all cases but for DDI. The different concentration of problems is mostly observed between images with low ranking and random samples, while the difference between samples 1-25 and 26-50 is less pronounced. We observe that identifying label errors in a highly-curated dataset such as DDI is a nontrivial task which might exceed the design of the conducted experiment. Finally, the detection of off-topic samples is the task where SELFCLEAN achieves the lowest overall agreement with human annotators. Nevertheless, these results suggest a significant separation of off-topic samples within the ranking in at least half of the cases.

Table 15: Comparison of the percentage of issues found by humans in the 50 lowest-ranked samples with 50 random samples, and in samples 1 to 25 with samples 26 through 50. We report the percentage of issues in each sample and the corresponding $p$-value of a Mann–Whitney $U$ test, which represents the probability for the ranking to be unrelated to the position of problematic samples.

| | | Percentage of Human-Confirmed Problems | | | | | |
|---|---|---|---|---|---|---|---|
| Dataset | Data Quality Issue | Lowest 1-50 (%) | Random Sample (%) | $p$-value | Lowest 1-25 (%) | Lowest 26-50 (%) | $p$-value |
| DDI | Off-topic Samples | 12 | 8 | 0.25 | 20 | 4 | **0.04** |
| DDI | Near Duplicates | 12 | 0 | **0.006** | 24 | 0 | **0.005** |
| DDI | Label Errors | 22 | 32 | 0.86 | 20 | 24 | 0.63 |
| Fitzpatrick17k | Off-topic Samples | 14 | 4 | **0.04** | 12 | 16 | 0.65 |
| Fitzpatrick17k | Near Duplicates | 100 | 0 | $\mathbf{1.3 \times 10^{-23}}$ | 100 | 100 | undef |
| Fitzpatrick17k | Label Errors | 54 | 12 | $\mathbf{4.4 \times 10^{-6}}$ | 52 | 56 | 0.61 |
| ImageNet | Off-topic Samples | 62 | 48 | 0.08 | 56 | 68 | 0.80 |
| ImageNet | Near Duplicates | 92 | 0 | $\mathbf{2.1 \times 10^{-20}}$ | 100 | 84 | **0.02** |
| ImageNet | Label Errors | 36 | 0 | $\mathbf{1.6 \times 10^{-6}}$ | 48 | 24 | **0.04** |
| Food-101N | Off-topic Samples | 24 | 4 | **0.002** | 36 | 12 | **0.02** |
| Food-101N | Near Duplicates | 100 | 0 | $\mathbf{1.3 \times 10^{-23}}$ | 100 | 100 | undef |
| Food-101N | Label Errors | 72 | 34 | $\mathbf{7.6 \times 10^{-5}}$ | 80 | 64 | 0.61 |

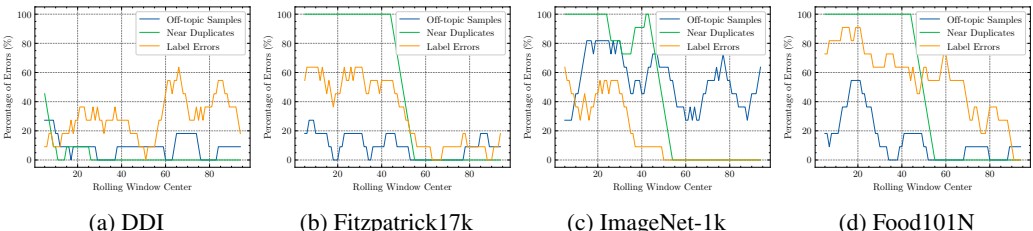

(a) DDI          (b) Fitzpatrick17k          (c) ImageNet-1k          (d) Food101N

Figure 9: Visualization of the percentage of quality issues found across the first 50 samples in the SELFCLEAN ranking and in 50 random samples, using a rolling window of size 10. Results are reported across four datasets and for each issue type.

# J Scoring for off-topic samples

This section describes how to construct a score based on hierarchical clustering, such that samples with a high probability of being off-topic have significantly lower values compared to the bulk of the data. Note that, although in practice we use single-linkage agglomerative clustering, this heuristic construction can be applied to any distance-based hierarchical clustering and is formulated accordingly.

**Notation.** Hierarchical clustering over a set of data points numbered $\{1, \ldots, N\}$ can be represented with a hierarchy of sets $\mathcal{H}_n$ which specify clusters at each level $n$. Let $n$ correspond to the number of clusters at a specific step $\mathcal{H}_n = \{\mathcal{C}_{1n}, \ldots, \mathcal{C}_{nn}\}$, where $\mathcal{C}_{in}$ represents the $i$-th cluster at level $n$ in the hierarchy. For instance, in agglomerative clustering, $n$ runs from $N$ to 1 as the algorithm proceeds and more data points are merged. Without loss of generality, it is possible to reindex clusters such that indices of merged sets are always consecutive, and the other sets in $\mathcal{H}_n$ do not change their relative order

$$\mathcal{C}_{in} = \begin{cases} \mathcal{C}_{i(n+1)} & \text{if } i < i_n, \\ \mathcal{C}_{i(n+1)} \cup \mathcal{C}_{(i+1)(n+1)} & \text{if } i = i_n, \\ \mathcal{C}_{(i+1)(n+1)} & \text{if } i > i_n, \end{cases} \quad \text{for } i = 1, \ldots, n \text{ and } n = 1, \ldots, N, \tag{3}$$

where from step $n+1$ to step $n$ clusters $i_n$ and $i_n+1$ are merged into cluster $i_n$. The hierarchy of sets $\mathcal{H}_n$ induces a dendrogram, i.e., a tree graph where each cluster is a node connected to its direct parent and children. Each element $n$ of the hierarchy (except for $\mathcal{H}_N$, where every point is in a separate cluster) can also be associated with a distance $d_n$ which is the one at which the last two clusters were merged, $d_n = \text{dist}(\mathcal{C}_{i_n(n+1)}, \mathcal{C}_{(i_n+1)(n+1)})$. To define a ranking, we sort the dendrogram such that at every step $|\mathcal{C}_{i_n(n+1)}| \leq |\mathcal{C}_{(i_n+1)(n+1)}|$, i.e., the cluster which contains the fewest leaves comes first, based on the idea that outliers are associated with merges containing fewer leaves [46]. In case of ties, the cluster created at the largest distance precedes the other.

**Scoring.** To produce a score for each node in the dendrogram, natural building blocks are the merge distance, the sizes of the merged clusters, and their interactions [78]. Accordingly, we define scores by drawing the dendrogram in a $[0, 1] \times [0, 1]$ square where the horizontal axis is one minus the (merge) distance $d$ and the vertical axis is the weight $w_{in}$ of cluster $\mathcal{C}_{in}$ which is defined recursively below. Note that the possible values for the distance range from 0 to 1, which can generally be achieved with a transformation. This guarantees that $1 - d$ spans the same range. This graphical construction is illustrated in the right panel of figure 10. The score of each leaf is determined at each merge distance $d$ by the weight $w_{jn}$ of the cluster $\mathcal{C}_{jn}$ it belongs to between merge distance $d_n$ and $d_{n-1}$. Formally, the off-topic sample score $s_{\text{OT}}(\boldsymbol{e}_i)$ is then given by the area under the curve $f_i(d)$ where

$$f_i(d) = w_{jn} \quad \text{if} \quad d_n \leq d < d_{n-1} \quad \text{and} \quad i \in \mathcal{C}_{jn}, \qquad n = 1, \ldots, N, \tag{4}$$

with $d_N = 0$ and $d_0 = 1$. For convenience, we define $p_{in} = |\mathcal{C}_{in}|/N$ to be the probability of cluster $\mathcal{C}_{in}$ and set $w_{0n} = 0$ and $w_{11} = 1$. To define the weights, we propose a rule which we call leaves and distances (LAD) and reads

$$w_{i(n+1)} = \begin{cases} w_{in} & \text{if } i < i_n, \\ w_{(i_n-1)n} + (w_{i_n n} - w_{(i_n-1)n})p_{i(n+1)}/p_{i_n n} & \text{if } i_n \leq i \leq i_n + 1, \\ w_{(i-1)n} & \text{if } i > i_n + 1. \end{cases} \tag{5}$$

Essentially, at each split, children cluster $\mathcal{C}_{i(n+1)}$ receives a weight $w_{i(n+1)}$ which is proportional to its relative size $p_{i(n+1)}/p_{i_n n}$ with respect to the parent cluster, while bound between the previous cluster weight $w_{(i_n-1)n}$ and the parent cluster weight $w_{i_n n}$.

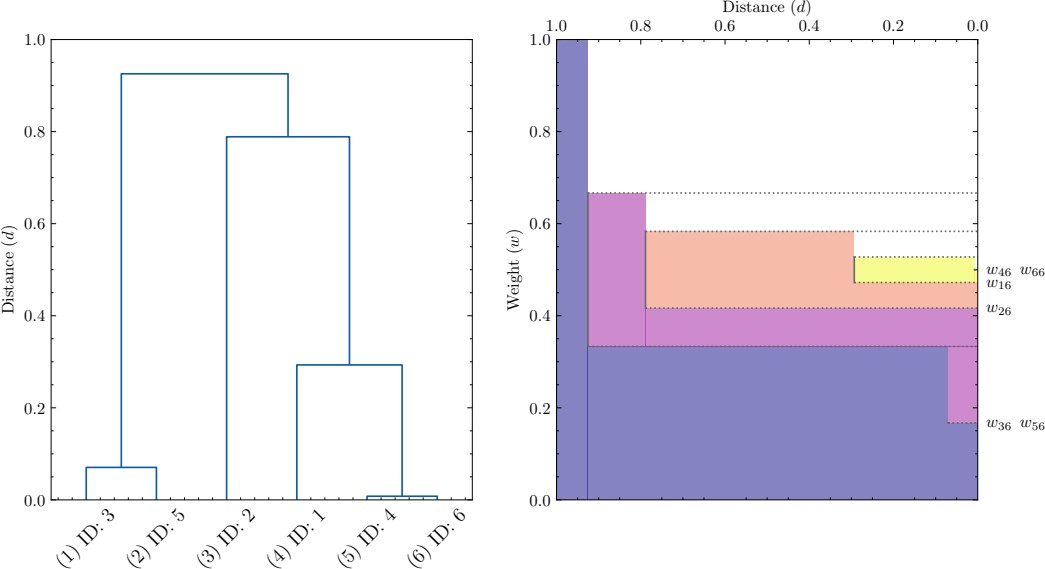

Figure 10: The left plot shows an example of a dendrogram for hierarchical clustering, and the right plot an illustration of the leaves and distances (LAD) scoring. In the left plot, the x-axis shows the ranking of the different points in brackets and the corresponding identification number. In the right plot, the right side of the y-axis shows the weights $w_{in}$ corresponding to equation 5.

# K    Automatic cleaning

In section 3.2, we defined scores that take extreme values for candidate issues. Isolating data issues using such a score is essentially a one-dimensional outlier detection problem. Here, we construct a procedure to detect outlier scores, which works well with SELFCLEAN. We then demonstrate that detected outliers are robust to the values of the two hyperparameters introduced by such procedure.

## K.1    Automatic cleaning procedure

We start with the intuition that detecting problematic samples is straightforward if the scores are smoothly distributed for normal data, but are far from the bulk for data with quality issues. However, all scores in this work range from 0 to 1, and increasingly extreme issues approach zero score without leaving large gaps on the score scale. For this reason, we expand the neighborhoods of 0 and 1 using a logit transformation, $\tilde{s} = \log[s/(1-s)]$. The transformed scores $\tilde{s}$ then range over the whole real axis enabling a better separation between normal and problematic samples.

Since the logit transformation has Jacobian $|\,\mathrm{d}\tilde{s}\,/\,\mathrm{d}s\,| = e^{-\tilde{s}}/(1 + e^{-\tilde{s}})^2$, under broad assumptions we expect the dominant behavior of the transformed score distribution to drop at least as quickly as a logistic probability density function for $\tilde{s} \to \pm\infty$. Note that this is the case even if the original score distribution is not just constant but presents an integrable power-law singularity for $s \to 0, 1$.

To identify outlier samples, we first attempt to isolate a region on the left tail of the distribution that is free of issues. To this end, we introduce a hyperparameter $\alpha$, the "contamination rate guess", which represents a generous estimate of the fraction of issues in the dataset. For data quality issues where a score is associated to each sample, we simply drop the lowest $\lfloor \alpha_1 N \rfloor$ scores with $\alpha_1 = \alpha$, while when a score is associated to a pair of samples, we discard the lowest $\lfloor \alpha_1 N(N-1)/2 \rfloor$ scores with $\alpha_1 = \alpha^2$. Indeed, when there are no interactions (e.g., only pairs of near-duplicates) we expect $\alpha N$ abnormally low near-duplicate scores, but in the worst-case interaction scenario (e.g., all views of the same sample) we await $\alpha N(\alpha N - 1)/2$ low out-of-distribution scores, which is equivalent to the above expression for $\alpha_1$ when $\alpha N \gg 1$. Besides dropping the potentially problematic samples, we also select an upper score bound for the range of interest, since we aim at reproducing only the smooth *left* tail of the distribution. Reasonable choices are values between the lower score cutoff determined by $\alpha_1$ and the median, paying attention that enough data is included for sufficient robustness to noise. For this reason, we choose the upper score cutoff to be the quantile corresponding to a fraction of data $\alpha_2$ which is the geometric mean between $\alpha_1$ and $1/2$, i.e., $\alpha_2^2 = \alpha_1/2$. We observe that the range produces robust statistical information if the number of samples is sufficiently large and $\alpha \ll 1/2$, where in practice $\alpha \lesssim 1/4$ is already stable.

Following the outlined heuristic argument, we approximate the smooth component of the left tail of the distribution using a logistic distribution with suitably chosen scale and location parameters, which has probability density function

$$\text{pdf}(\tilde{s}; \mu, \sigma) = \frac{1}{\sigma}\text{pdf}\left(\frac{\tilde{s} - \mu}{\sigma}\right), \qquad \text{pdf}(\hat{s}) = \frac{e^{-\hat{s}}}{(1 + e^{-\hat{s}})^2}. \tag{6}$$

Given the score cutoffs $\bar{s}_1$ and $\bar{s}_2$ corresponding to the quantiles $\alpha_1$ and $\alpha_2$ of the empirically observed distribution, the scale $\sigma$ and location $\mu$ can be estimated as

$$\sigma = \frac{\bar{s}_2 - \bar{s}_1}{\bar{s}(\alpha_2) - \bar{s}(\alpha_1)}, \qquad \mu = \frac{\bar{s}_1 \bar{s}(\alpha_2) - \bar{s}_2 \bar{s}(\alpha_1)}{\bar{s}(\alpha_2) - \bar{s}(\alpha_1)}, \qquad \bar{s}(\alpha_m) = \log \frac{\alpha_m}{1 - \alpha_m} \quad \text{for } m = 1, 2. \tag{7}$$

Here $\bar{s}(\alpha_m)$ indicates the percentage point function of the logistic distribution, i.e., the inverse of its cumulative distribution function. Note that the whole estimation procedure for the left tail of the distribution relies exclusively on quantiles and is, therefore, naturally robust to outliers.

With an estimate of the smooth score distribution for normal data, we can identify abnormal samples by requesting that they be unlikely generated by the same random process. This is achieved by demanding that the probability of obtaining a score below an outlier cutoff $s_{\text{cut}}$ be less than a significance level $q$ times the expected fraction of outliers, which is $2\alpha/(N-1)$ in the case of pairs of samples and $\alpha$ otherwise. We set the hyperparameter $q$ to 0.05 corresponding to a 95% one-sided confidence level and study the influence of this choice in section K.4. All samples with scores lower than the outlier cutoff will be then classified as problematic.

The strength of the aforementioned procedure lies in its ability to consistently detect outliers despite requiring multiple steps and introducing two additional hyperparameters. The number of outliers identified remains largely unaffected by reasonable choices for $\alpha$ and $q$. The remaining parts of appendix K are dedicated to showing that the procedure is intuitive and assumptions are empirically acceptable (K.2), and to demonstrating that detected outliers exhibit low sensitivity to the contamination rate guess $\alpha$ (K.3) and to the significance level $q$ (K.4).

## K.2   Automatic cleaning examples

In figure 11, we illustrate the fit to the left tails of distributions for representative datasets, together with the relevant range used to estimate scale and location and the position of the outlier cutoff to classify data quality issues. We observe that the probability density function is a qualitatively good estimate of the density-normalized histograms in the expected range, i.e., for the smooth component of the histogram's left tail, within sampling uncertainties. The fit quality is somewhat lower for off-topic samples, probably due to the score range which is all above $\tilde{s} = 0$. We also carried out experiments with a Gaussian functional form for score distribution tails and observed only minor changes, which resulted in a slightly reduced number of detected problems.

## K.3   Influence of the contamination rate guess $\alpha$

In figure 12, we analyze the sensitivity of the number of detected data quality problems with respect to the contamination rate guess $\alpha$ for all issue types and representative datasets analyzed in this paper. In these plots, the significance level $q$ is fixed to its default value of 0.05. We observe that the fraction of found problems does not depend much on $\alpha$ over several orders of magnitude, suggesting a sensitivity to this hyperparameter that is approximately vanishing or at most logarithmic. It is by virtue of this reduced dependence that we can fix $\alpha = 0.10$ throughout the paper and that fully automatic cleaning is able to produce stable results with limited prior knowledge of dataset quality.

## K.4   Influence of the significance level $q$

In figure 13 we report the fraction of detected problematic samples as a function of the significance level $q$, for all considered issue types and representative datasets. We can see that this hyperparameter essentially determines the number of outliers found, which is monotonically increasing with $q$. Moreover, the number of identified issues has, in most cases, a dependence on $q$ which is less than linear. In some cases, especially when the number of detected outliers is below percent level or $q$ approaches 1, we see more severe sensitivity to the specific value. This may be because the empirical score distribution changes more abruptly than estimated by the logistic fit, as happens for off-topic samples, or because the region immediately below the lower score cutoff $\bar{s}(\alpha_1)$ (which corresponds to $q = 1$) is densely populated almost by construction. It is however clear that $q$ regulates how extreme scores need to be for a sample to be considered problematic. A value of $q = 10^{-3}$ will only select very apparent data quality issues, $q = 1/4$ will almost certainly also include a significant fraction of valid samples, and the choice of $q = 0.05$ strikes a compromise between precision and recall.

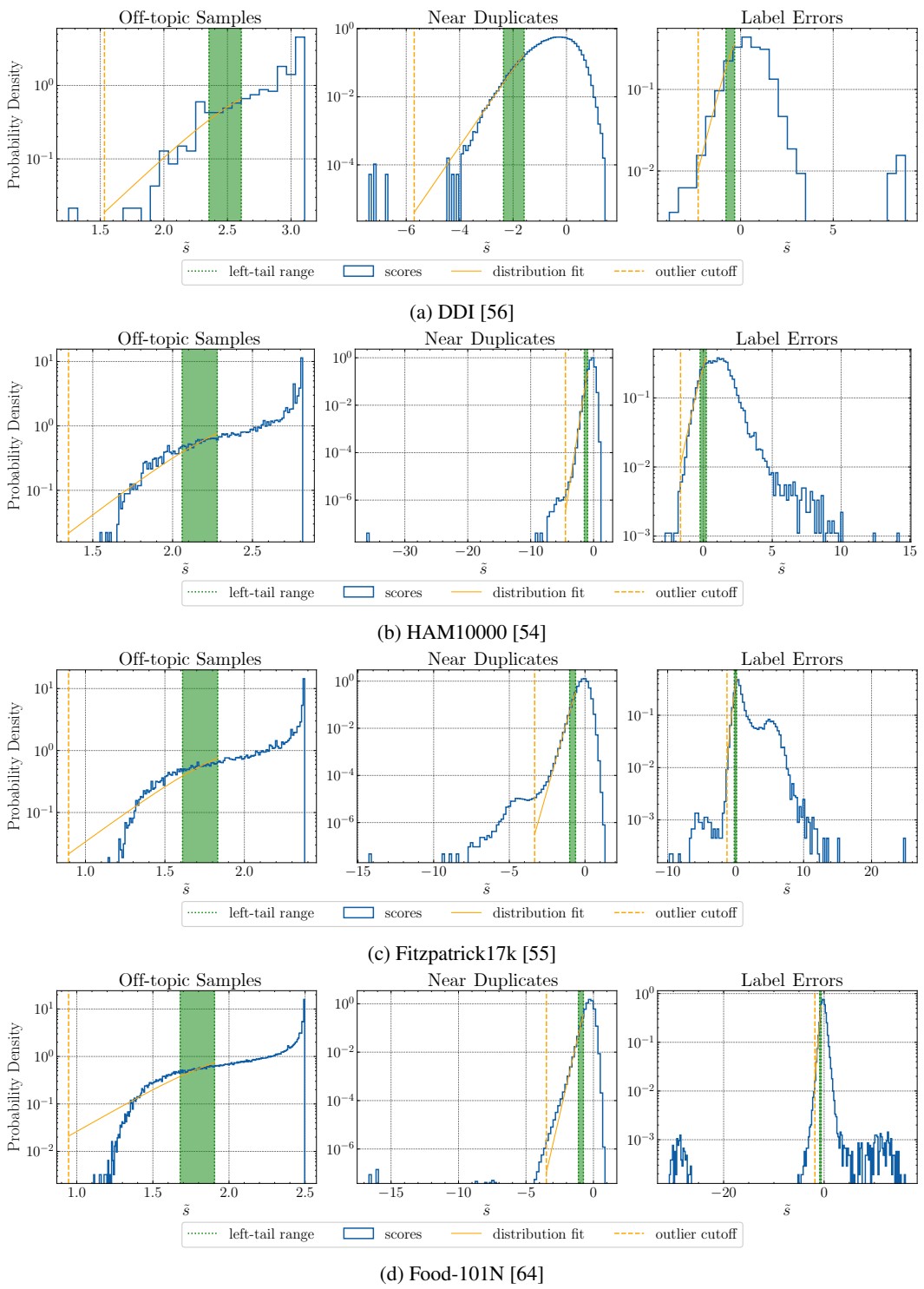

Figure 11: Score histogram (blue) and associated left-tail distribution fit (solid orange) with outlier cutoff (dashed orange) for all considered issue types and representative datasets. The green shaded area represents the range $[\bar{s}_1, \bar{s}_2]$ which is used to determine location and scale of the associated logistic distribution. The values $\alpha = 0.10$ and $q = 0.05$ are used throughout.

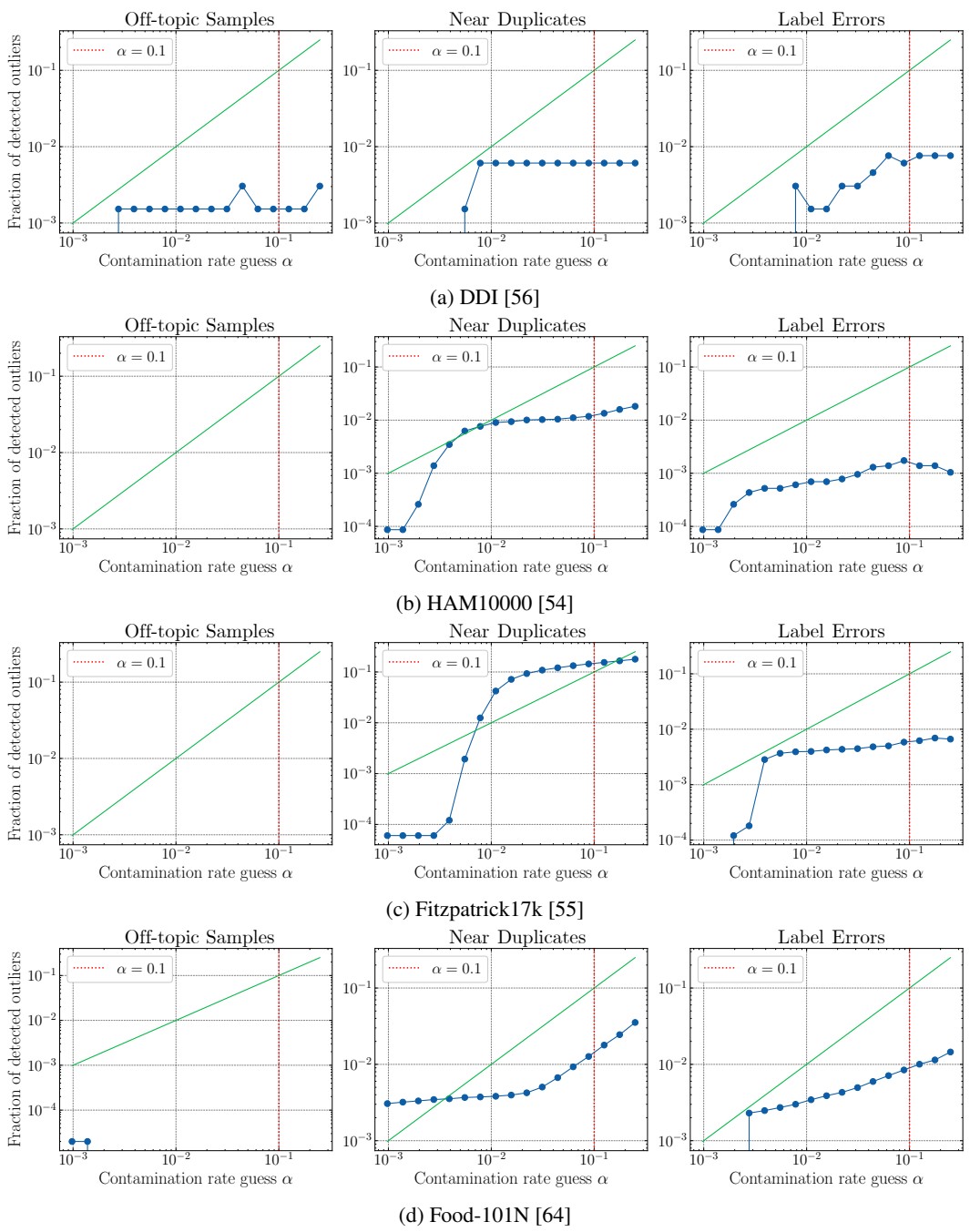

Figure 12: Dependence of the fraction of detected data quality issues on the contamination rate guess $\alpha$ for all considered issue types and representative datasets, at a fixed significance level $q = 0.05$. The observed behavior is reported in blue. It is outside of the lower margin of the plots when no problems are found. The green solid line represents a fraction of detected issues which is equal to the contamination rate guess for reference. The vertical dotted red line indicates the default value $\alpha = 0.10$ used in the rest of the paper.

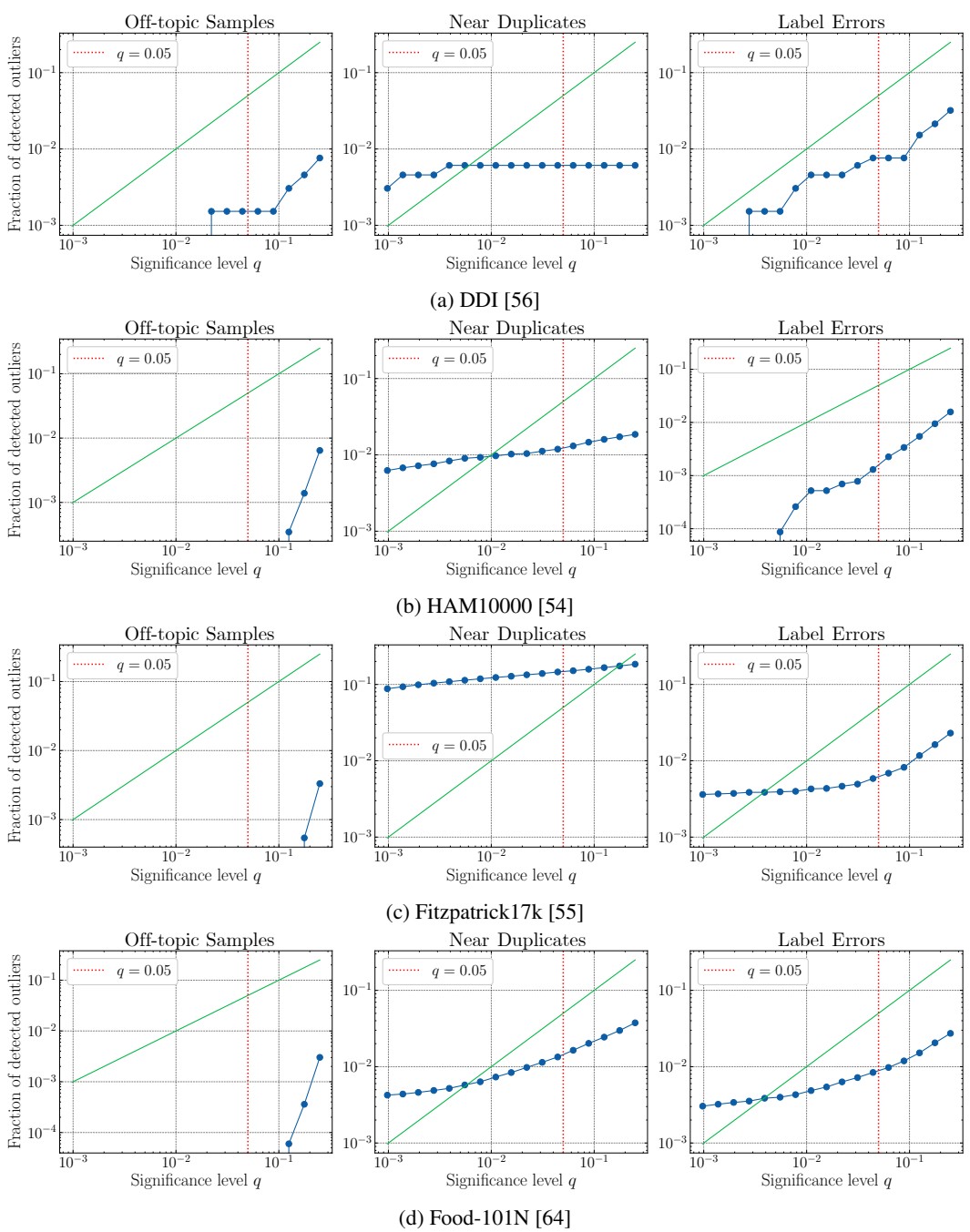

Figure 13: Impact of the choice of the significance level $q$ on the fraction of detected data quality issues, across issue types and for representative datasets, for a fixed contamination rate guess $\alpha = 0.10$. The observed dependency on $q$ is reported in blue, and it is below the lower margin of the plots when no problematic samples are found. The diagonal green solid line is just a reference to guide reading, and the dotted red line indicates the default choice $q = 0.05$.

# L  Inspection of benchmark datasets

This section reports the results of auditing multiple vision benchmarks using SELFCLEAN. In section L.1, we estimate the number of issues in fully automatic mode. Sections thereafter illustrate the rankings by visualizing the top 15 samples of each issue type, namely off-topic samples, near duplicates, and label errors. General conclusions are drawn in section 6, while here we report more specific observations.

Applying SELFCLEAN to multiple benchmark datasets across different domains has led to different insights on why some of these data quality issues may occur. Off-topic samples in the medical domain are often caused by device malfunctions, wrong configurations, tests, or other scanning errors (figure 18 Rank 2-8 and figure 21 Rank 1, 5, and 9). Near duplicates can often be traced to data acquisition problems, such as crawling both an image and its thumbnail (figure 23) or the metadata failing to correctly flag that two images have a common origin (figure 14 and 20). The most apparent label errors are often near-duplicate samples with different labels (figure 25 Rank 1&2, 4&6, 9&10, and 8&13), which indicate (understandable) difficulties in the annotation process, or off-topic samples with a label (figure 19 Rank 2-4), which easily arise in (semi-)automated annotation procedures.

## L.1  Estimation of issues in benchmark datasets

Table 16: Estimated percentage of data quality issues in vision benchmarks obtained using SELF-CLEAN's automatic mode with $\alpha = 0.10$ and $q = 0.05$. Images marked as originating from the same person, patient or lesion were excluded from the near-duplicate count whenever available.

| Dataset | Size | Estimated Issues | | | |
| --- | --- | --- | --- | --- | --- |
| | | Off-topic Samples | Near Duplicates | Label Errors | **Total** |
| *Medical Images* | | | | | |
| DDI | 656 | 1 (0.2%) | 4 (0.6%) | 5 (0.8%) | 10 (1.5%) |
| PAD-UFES-20 | 2,298 | 0 (0.0%) | 0 (0.0%) | 5 (0.4%) | 5 (0.4%) |
| HAM10000 | 11,526 | 0 (0.0%) | 1 ($<0.1\%$) | 17 (0.2%) | 18 (0.2%) |
| VinDr-BodyPartXR | 16,086 | 263 (1.6%) | 20 (0.1%) | 74 (0.5%) | 357 (2.2%) |
| Fitzpatrick17k | 16,574 | 18 (0.1%) | 2,446 (14.8%) | 103 (0.6%) | 2,567 (15.5%) |
| ISIC-2019 | 33,569 | 0 (0.0%) | 1,200 (3.6%) | 97 (0.3%) | 1,297 (3.9%) |
| CheXpert[6] | 223,414 | 6 ($<0.1\%$) | 0 (0.0%) | 303 (0.1%) | 309 (0.1%) |
| PatchCamelyon | 327,680 | 98 ($<0.1\%$) | 12,845 (3.9%) | 589 (0.2%) | 13,532 (4.1%) |
| *General Images* | | | | | |
| STL-10 | 5,000 | 0 (0.0%) | 7 (0.1%) | 21 (0.4%) | 28 (0.5%) |
| ImageNet-1k Validation | 50,000 | 0 (0.0%) | 36 (0.1%) | 262 (0.5%) | 298 (0.6%) |
| CelebA | 202,599 | 2 ($<0.1\%$) | 810 (0.4%) | 1,033 (0.5%) | 1,845 (0.9%) |
| Food-101N | 310,009 | 310 (0.1%) | 4,433 (1.4%) | 2,728 (0.9%) | 7,471 (2.4%) |

---

[6]Label errors refer to atelectasis detection only since the classification task admits multiple labels, and expert agreement is the highest for this condition.

## L.2 ImageNet-1k

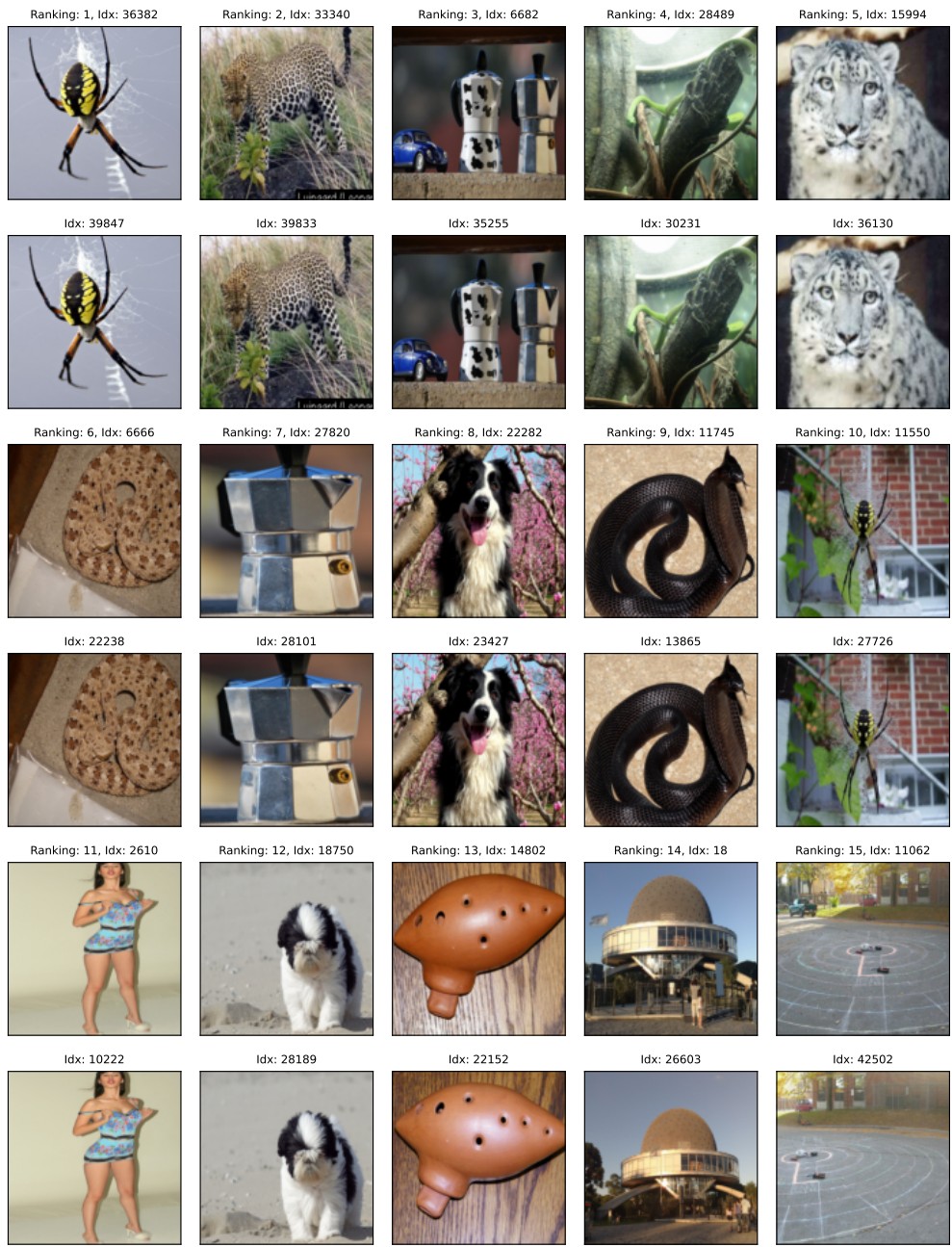

Figure 14: Ranking produced by SELFCLEAN for near duplicates in the ImageNet-1k validation set, of which the top 15 are shown along with the respective rank and index.

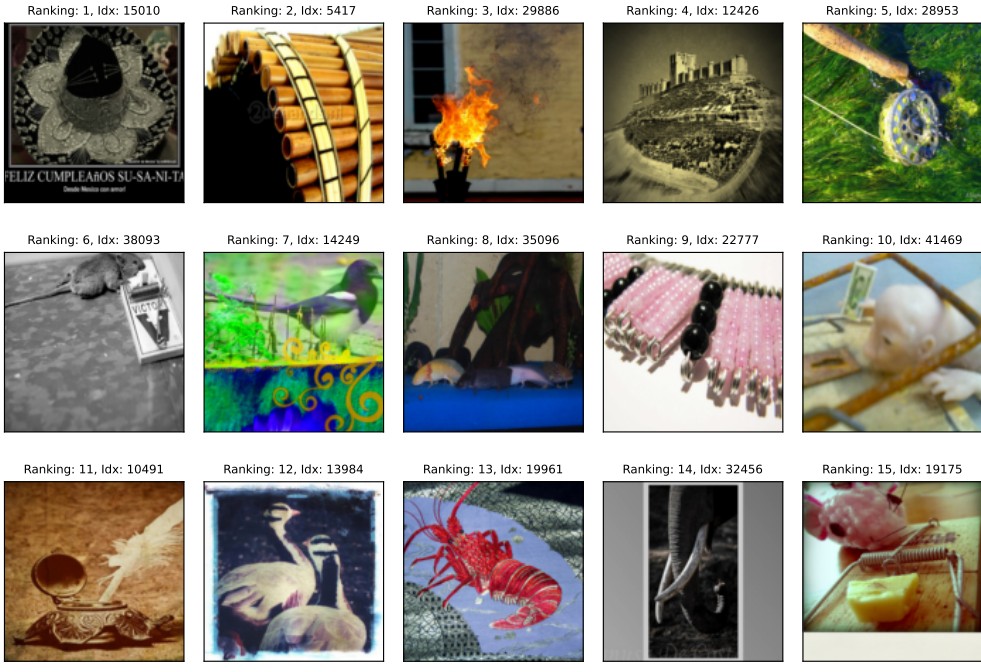

Figure 15: Ranking produced by SELFCLEAN for off-topic samples in the ImageNet-1k validation set, of which the top 15 are shown along with the respective rank and index.

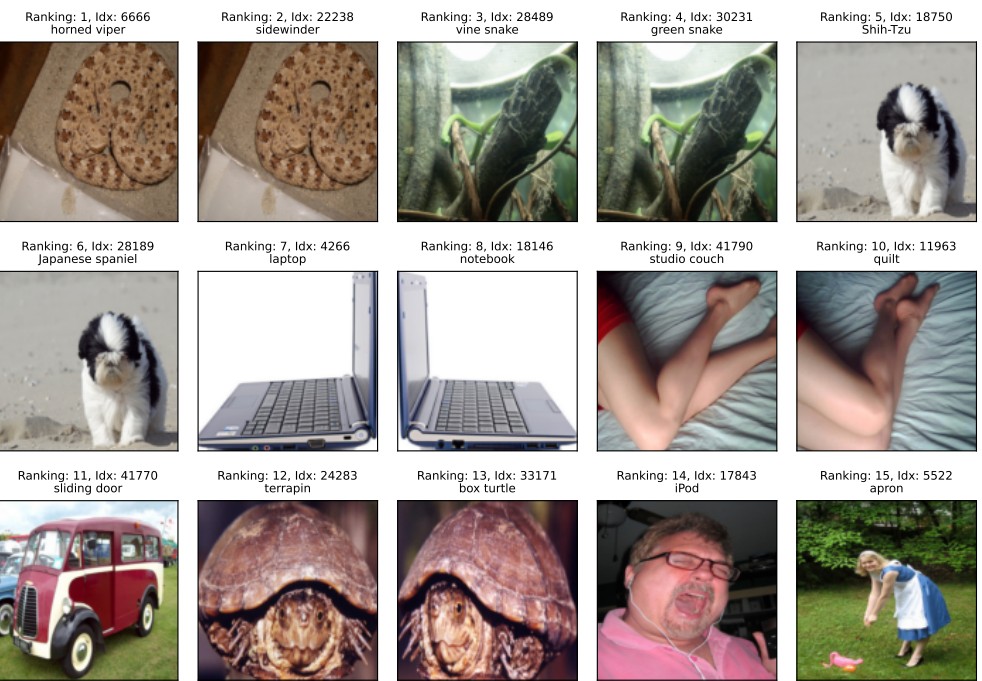

Figure 16: Ranking produced by SELFCLEAN for label errors in the ImageNet-1k validation set, of which the top 15 are shown along with the respective rank, index, and original label.

## L.3 CheXpert

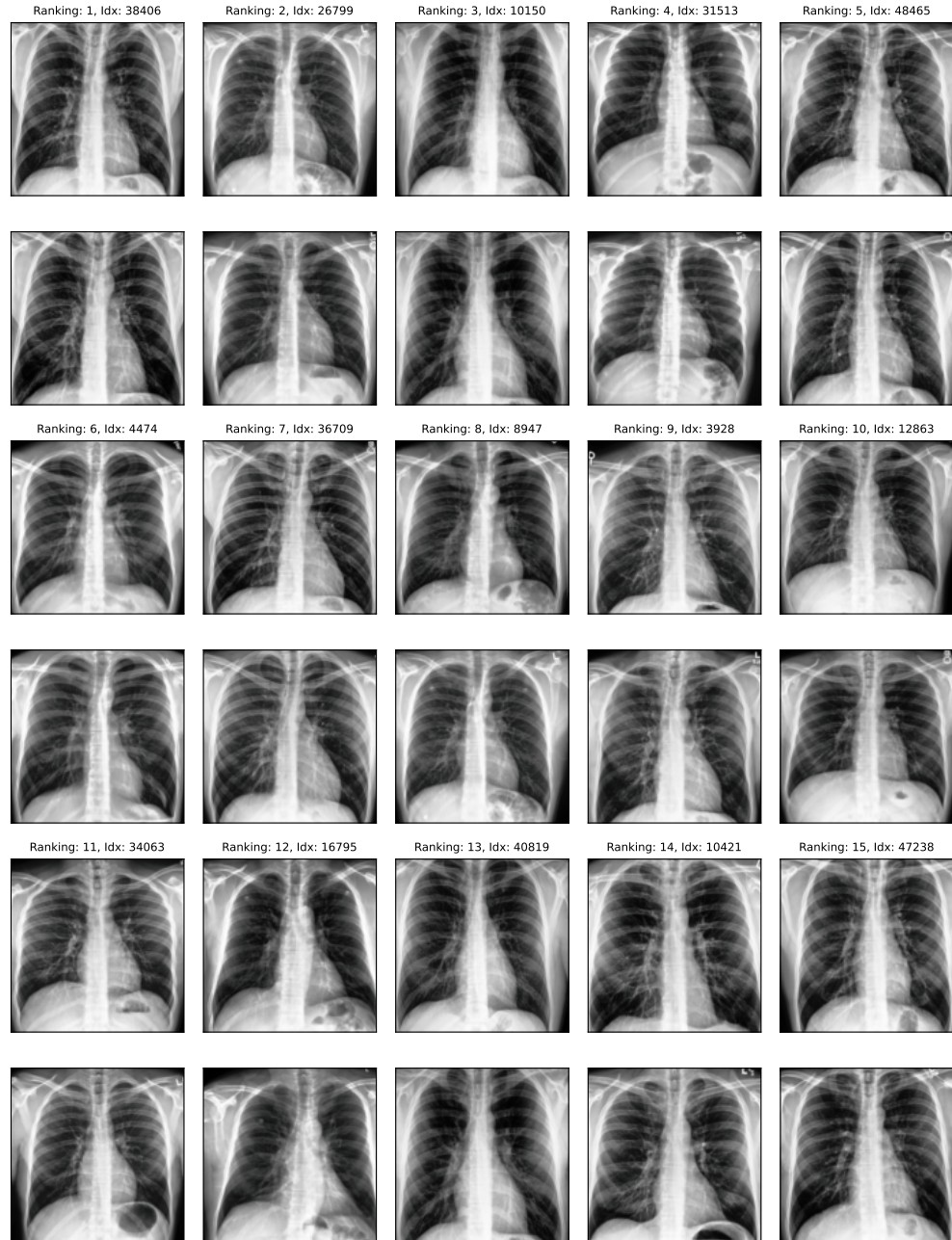

Figure 17: Ranking produced by SELFCLEAN for near duplicates in CheXpert, of which the top 15 are shown along with the respective rank and index.

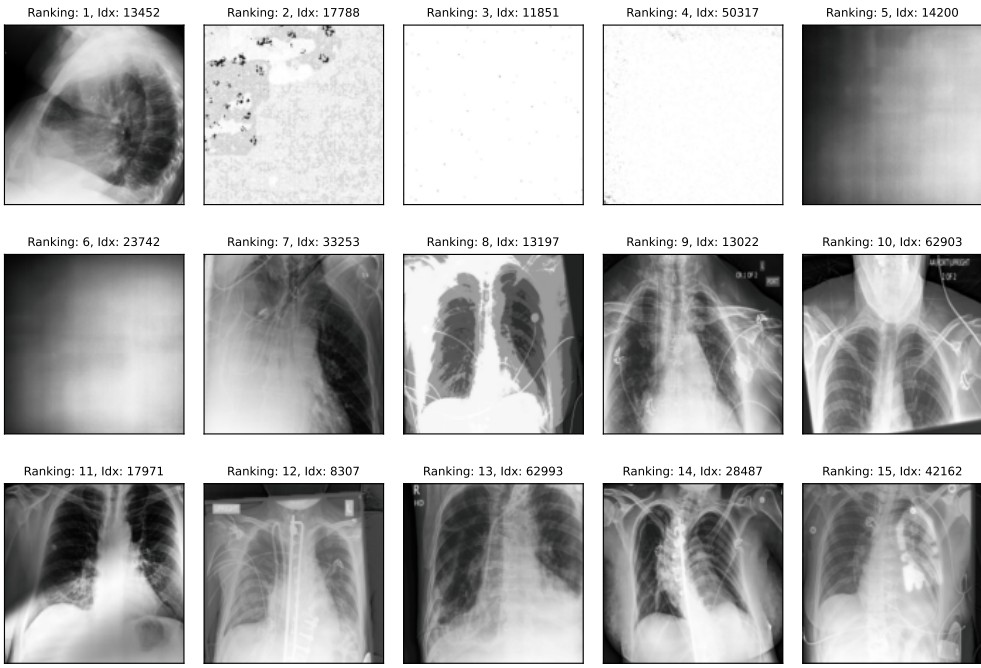

Figure 18: Ranking produced by SELFCLEAN for off-topic samples in CheXpert, of which the top 15 are shown along with the respective rank and index.

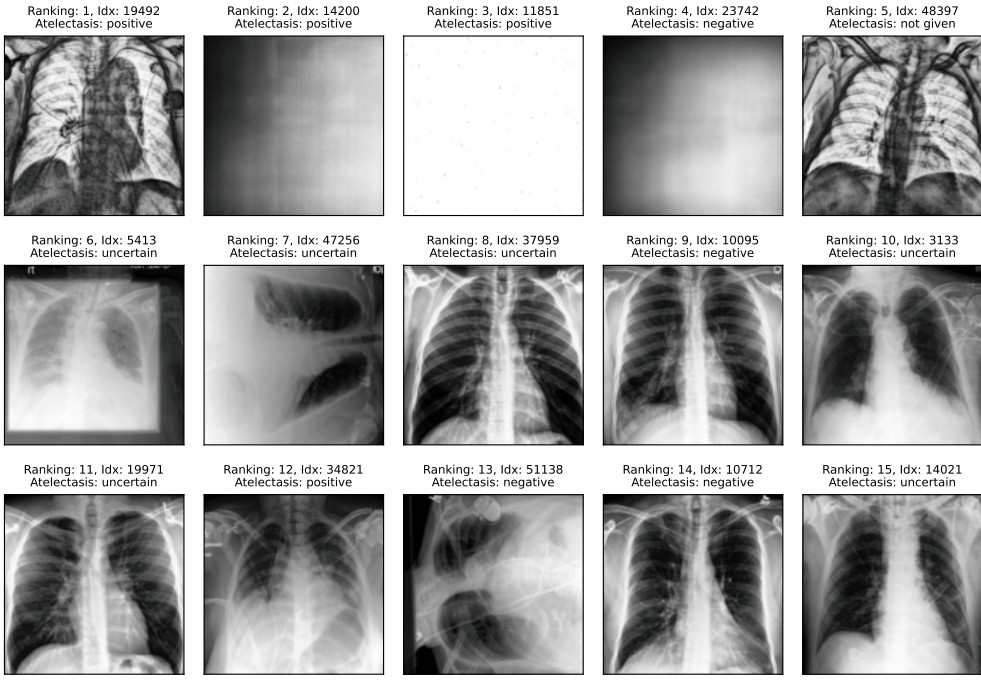

Figure 19: Ranking produced by SELFCLEAN for atelectasis label errors in CheXpert, of which the top 15 are shown along with the respective rank, index, and original label.

## L.4  PatchCamelyon

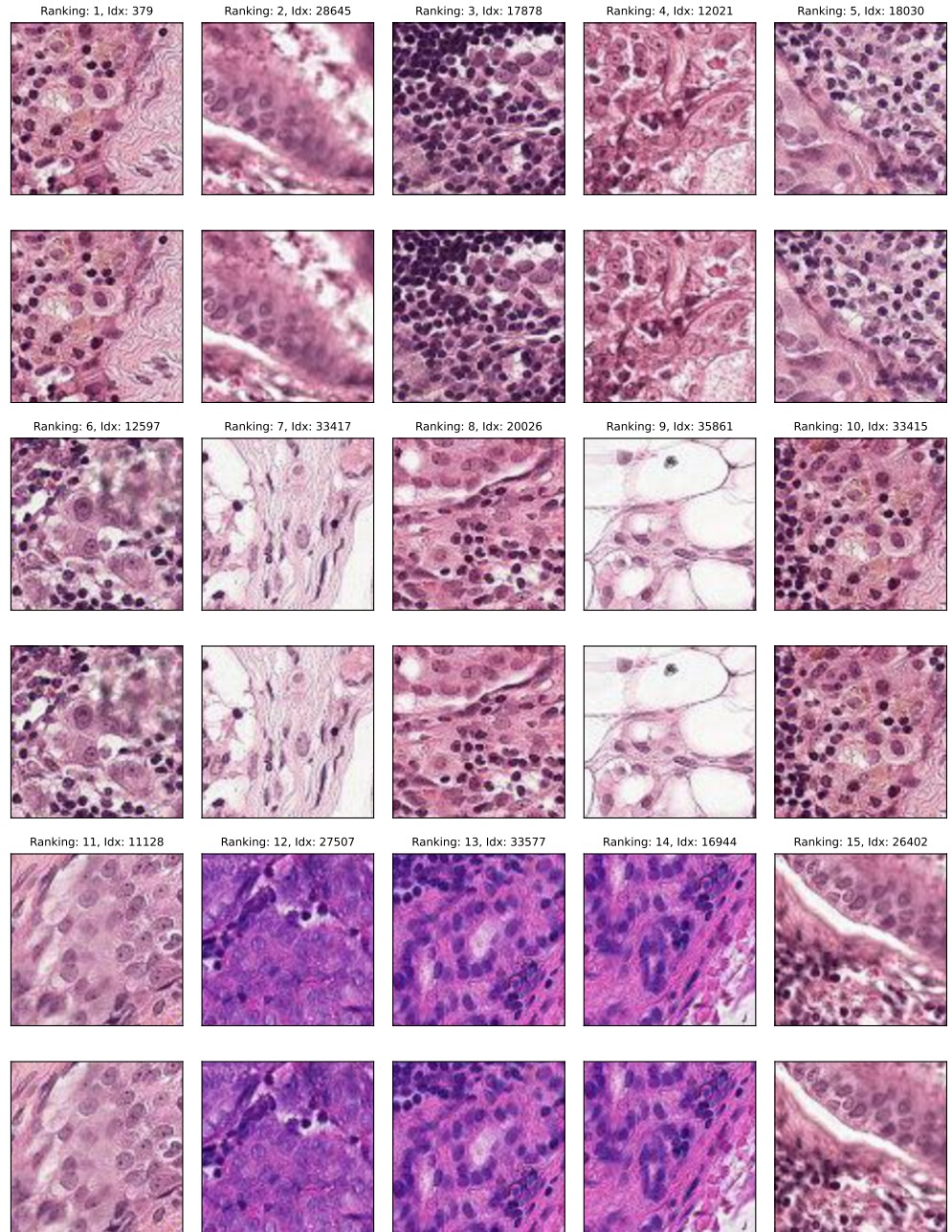

Figure 20: Ranking produced by SELFCLEAN for near duplicates in PatchCamelyon, of which the top 15 are shown along with the respective rank and index.

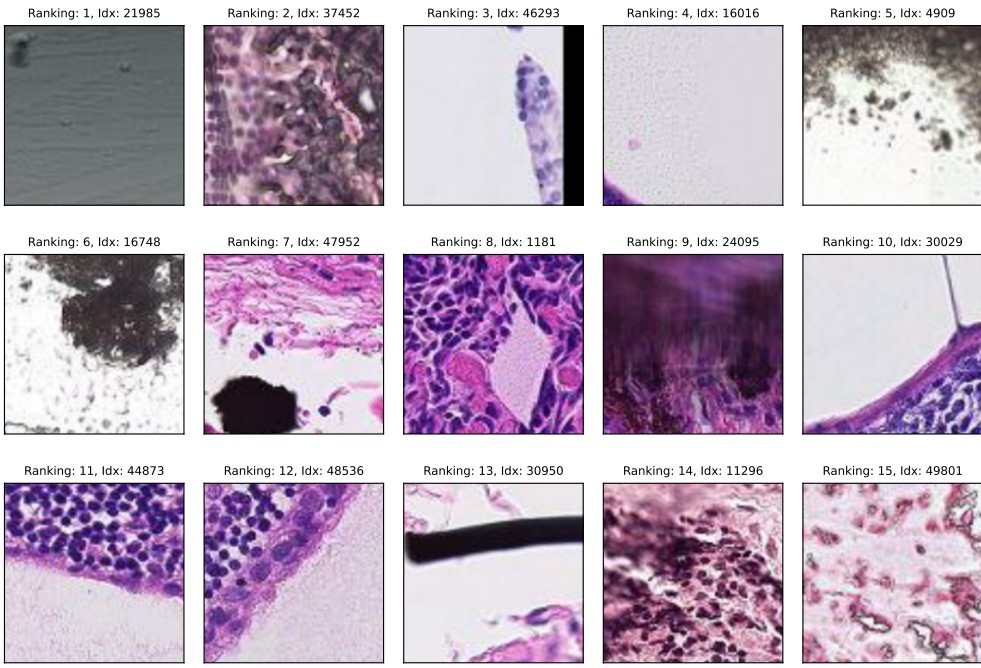

Figure 21: Ranking produced by SELFCLEAN for off-topic samples in PatchCamelyon, of which the top 15 are shown along with the respective rank and index.

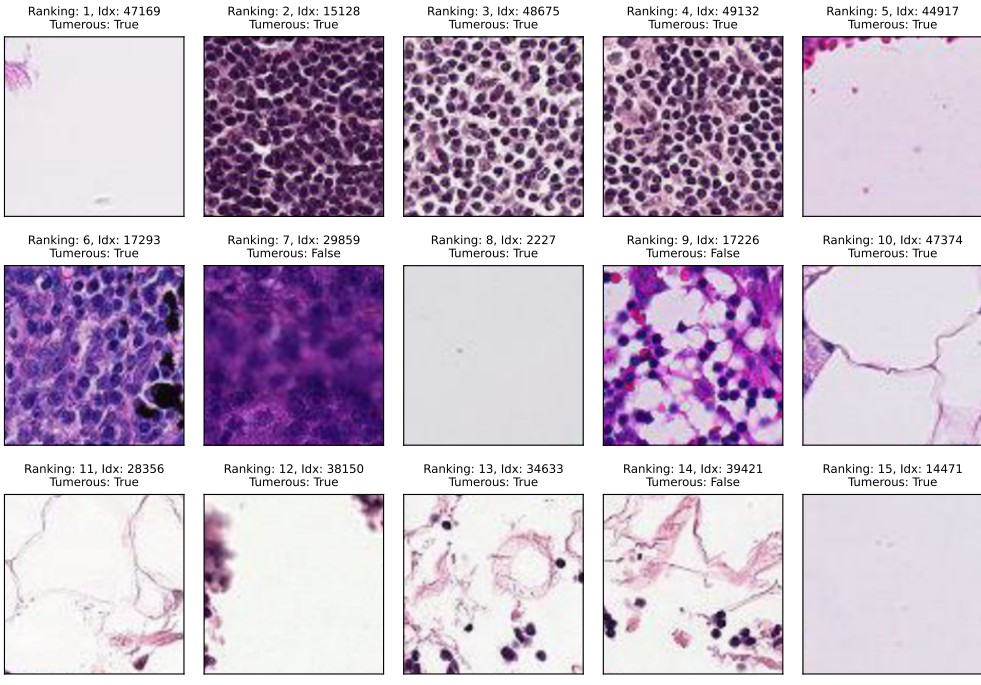

Figure 22: Ranking produced by SELFCLEAN for label errors in PatchCamelyon, of which the top 15 are shown along with the respective rank, index, and original label, i.e., if the patch is tumerous.

## L.5  Fitzpatrick17k

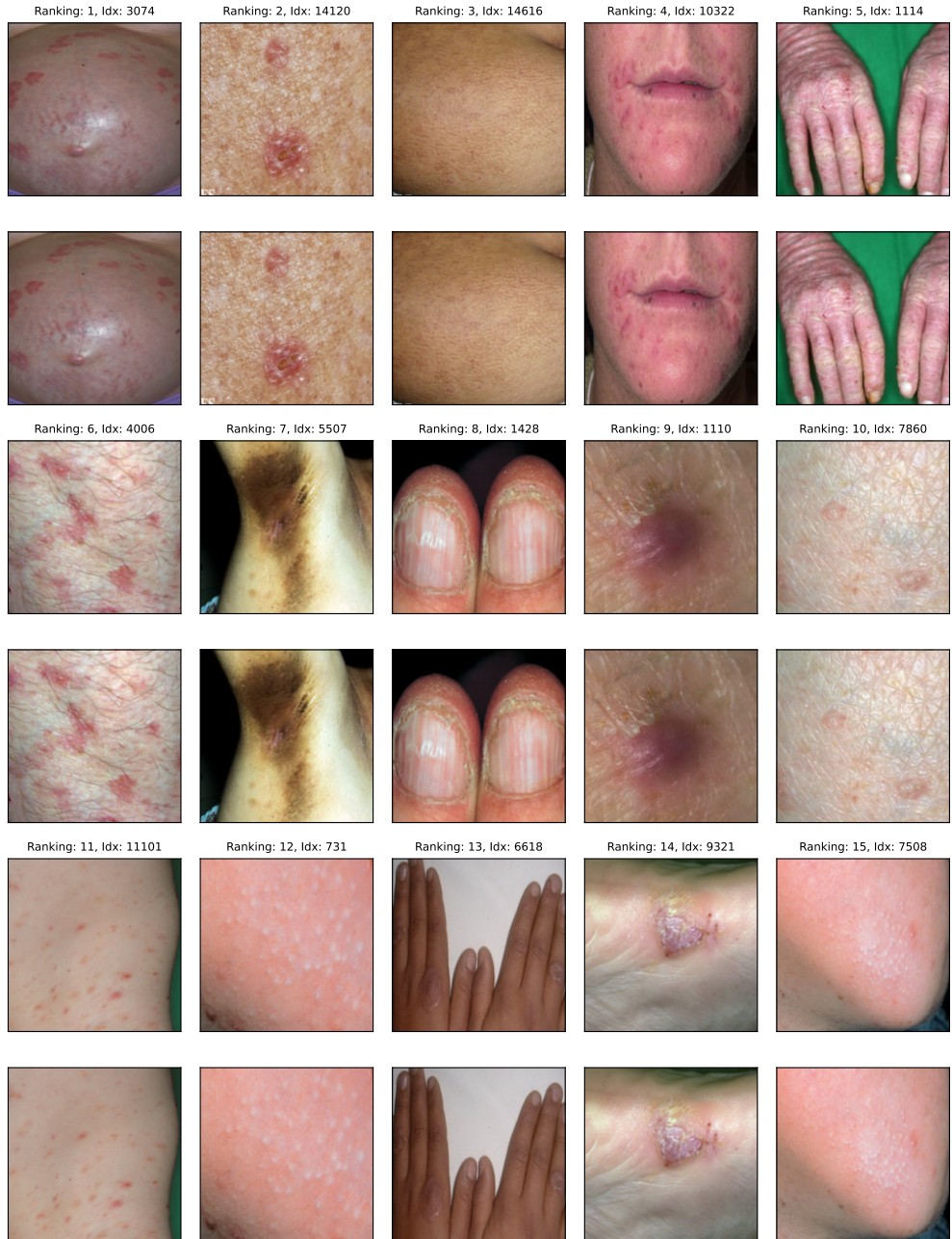

Figure 23: Ranking produced by SELFCLEAN for near duplicates in the Fitzpatrick17k, of which the top 15 are shown along with the respective rank and index.

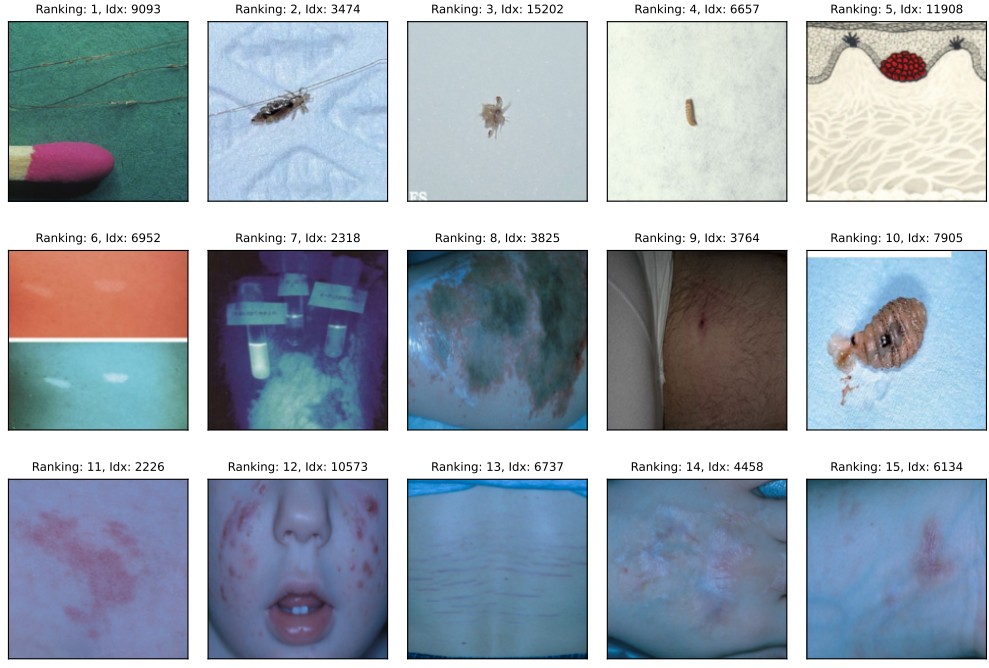

Figure 24: Ranking produced by SELFCLEAN for off-topic samples in the Fitzpatrick17k, of which the top 15 are shown along with the respective rank and index.

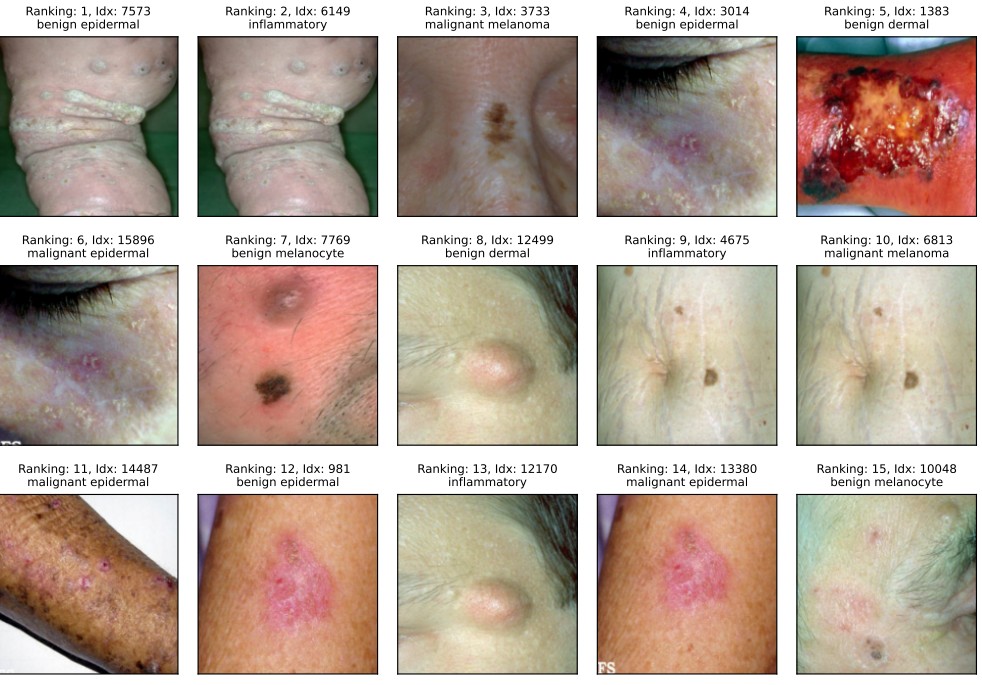

Figure 25: Ranking produced by SELFCLEAN for label errors in the Fitzpatrick17k, of which the top 15 are shown along with the respective rank, index, and original label.

