# OpenReview forum: "Intrinsic Self-Supervision for Data Quality Audits"
_NeurIPS.cc/2024/Datasets_and_Benchmarks_Track — NeurIPS 2024 Track Datasets and Benchmarks Poster_

### Official Review · Reviewer_fiv4 · 2024-07-05
**An interesting work, but it requires more in-depth analysis and discussion.**

**Rating:** 7
**Confidence:** 4
**Correctness:** Yes.
**Clarity:** Yes.

**Review:**

Please check the comments below.

**Strengths:**

- Data cleaning is a highly relevant research topic that aligns well with the scope of this track. The proposed SELFCLEAN tool has great potential to help researchers create and maintain image classification datasets.
- The experimental results are extensive and utilize a wide variety of datasets across general image classification, radiology, and dermatology tasks.
- Although SELFCLEAN is composed of existing techniques (SSL and Distance-based indicators) without introducing new components, its mechanism is reasonable for handling different types of data quality issues compared to confidence-based data cleaning methods. As a tool for data cleaning, this study has good originality and significance.

**Additional Feedback:**

The reviewer will consider raising the rating if the main concerns are addressed.

**Documentation:**

Yes.

**Ethics:**

It would be better to go through a separate ethics review for the usage of CelebA dataset in this submission by a specialized ethics reviewer.

**Limitations:**

Please check the Opportunities For Improvement.

**Opportunities For Improvement:**

## Major Weaknesses
- Analysis of the types of data quality is insufficient.
1. The authors define samples as off-topic when they are included in the dataset by mistake and consider them outliers in the feature space. However, the hidden stratification [1] phenomenon is not well-considered in this case. For example, when some subsets of the samples have low prevalence (e.g., 100 white dog images and one black dog image), the black dog image has a high potential to be determined as an outlier. Therefore, simply using ‘mistakes' to represent off-topic samples is not rigorous. It would be better to discuss more about this phenomenon in the main paper and Section H.3.
2. The effect of near duplicate samples on model performance is unclear. For example, if removing duplicate samples does not affect model evaluation, the importance of this issue will be lower than other types.
3. [2] defines four types of label errors. It would be better to show more analysis. The current version seems to only consider correctable label errors (label flip issue).
4. The severity of each data quality is not discussed. For example, which issue is the most serious?
5. The reviewer thinks that image quality should also be considered. For example, the image matches the ground truth label well and is not duplicated. However, it can be partially corrupted, have low resolution, be blurry, or have other issues.
- Missing some important baselines.
A recent work [3] proposes a benchmark for advanced label error detection methods. It is necessary to compare the SELFCLEAN with these methods.
- Synthetic settings may not reflect the data quality issue in real datasets well.
One possible solution is to use the CIFAR10H dataset [4] for experiments, which collects 51 judgments per image on average.
- Ethical concerns on the usage of CelebA dataset.
Please check whether this dataset can be used for experiments according to the work [5].

[1] Hidden stratification causes clinically meaningful failures in machine learning for medical imaging, 2020 ACM conference on health, inference, and learning.
[2] Pervasive Label Errors in Test Sets Destabilize Machine Learning Benchmarks, 2021 NeurIPS.
[3] AQuA: A Benchmarking Tool for Label Quality Assessment, 2023 NeurIPS.
[4] Human uncertainty makes classification more robust, 2019 ICCV.
[5] Mitigating Dataset Harms Requires Stewardship: Lessons from 1000 Papers, 2021 NeurIPS

## Minor Weaknesses
- The term 'computer vision datasets' is too broad. This paper seems to focus only on image classification tasks.
- The format in References is inconsistent. One possible option is the IEEE Reference Style Guide.
- Many preprint arXiv papers in References have been accepted and published. Please use their official proceedings.
- It would be better to move Section D. Evaluation metrics from the appendix to the main paper.
- It would be better to provide more instructions for running the code. README seems too brief. The reviewer fails to find the SSL training codes and pre-trained weights. Could the authors provide more details about them?
- Is the human verification tool open-sourced?

**Relation To Prior Work:**

Yes.

**Summary And Contributions:**

This study addresses three types of data quality issues, which include off-topic images, near duplicates, and label errors, with the aid of self-supervised representation learning and distance-based indicators. Extensive results on 12 image classification datasets show the effectiveness of the proposed SELFCLEAN methodology.

---

> ### Author Rebuttal · Authors · 2024-08-15
>
> We are grateful to reviewer fiv4 for their constructive criticism.
> Below are our replies to the major and minor weaknesses that were raised.
>
> **Major Weaknesses**:
>
> > *R.fiv4.1*: Hidden stratification phenomenon
>
> We completely agree with the reviewer that the *candidate* off-topic samples detected by SelfClean are not necessarily mistakes, as discussed in general for candidate data quality issues at the start of "Recommended use" in section 6.
> However, we rather see the off-topic ranking as a useful tool to expose hidden stratification, as it highlights which samples are considered to be global or local outliers by a model that is trained without labels.
> Moreover, the potential negative effects of off-topic sample cleaning in the presence of hidden stratification are indirectly explored in appendix H.3, where we use metadata to examine bias against minority groups which are not known during model training.
>
> We appreciate the reviewer's comment, as it highlights that the discussion of off-topic samples should be improved to immediately clarify that, although we are looking for mistakes, it may detect atypical but correct samples.
> The following clarification will be included in our revision at line 131 between "some examples." and "We achieve":
> "Atypical samples, due e.g. to the phenomenon of hidden stratification [1], may be included intentionally, and although they are not off-topic they may be revealed in the same search, but require different treatment."
>
> > *R.fiv4.2*: Effect of near duplicate samples
>
> The most prominent effect of near duplicates is when they cause a train-evaluation leakage and thus produce overly optimistic performance estimates.
> This can be seen in Table 3 for Fitzpatrick17k where cleaning the evaluation set and data leaks results in significantly lower performance (-4.1\%pt. in F1).
> As shown in Table 16, this dataset contains 15\% of near duplicates, but only a very small number of off-topic samples and label errors, so the effect can be attributed to lifting data leaks.
> In our experience, important data leaks are a common pattern since training-evaluation splits are often performed randomly, thus causing near-duplicate samples to be assigned to different splits and significantly changing performance scores through memorization.
>
> On the other hand, near duplicate pairs within the same data split may re-weight samples in an uncontrolled way, which may affect score estimates for evaluation splits and learning dynamics for the training split.
> Their influence in this setting is more limited for the datasets we considered so far, but we argue that information on near duplicates is anyways valuable and allows to handle all of these scenarios correctly.
> This point is now briefly included in a second paragraph of section 6 "Influence of dataset cleaning", which can be found in the answer to *R.fiv4.4*.
>
> > *R.fiv4.3*: Analysis label errors
>
> We thank the reviewer for pointing this out and agree that a more detailed analysis on the type of label error is interesting.
> Following the suggestion, we computed the alignment of the SelfClean label error ranking for each of the four types of label errors using a reannotated version of ImageNet-1k [2].
> Results show that label error rankings correspond best to correctable errors, and worst to samples with low annotator agreement.
> Across all four types SelfClean outperforms competitors by a clear margin.
>
> | Label Error Type | Positive Samples (%) | Method | Rep. | AUROC (%) | AP (%) |
> | :--- | :---: | :--- | :--- | ---: | ---: |
> | Correctable |  2.9 | CLearning | INet | 44.4 | 2.9 |
> |  | | FastDup | INet | 60.0 | 3.8 |
> |  |  | SelfClean | DINO | **66.2** | **5.4** |
> | Multi-label |  1.2 | CLearning | INet | 55.1 | 1.3 |
> |  | | FastDup | INet | 63.4 | 1.8 |
> |  |  | SelfClean | DINO | **67.7** | **2.6** |
> | Neither | 0.6 | CLearning | INet | 52.2 | 0.6 |
> |  |  | FastDup | INet | 65.8 | 0.8 |
> |  |  | SelfClean | DINO | **69.5** | **1.2** |
> | Non-agreement | 1.2 | CLearning | INet | 50.5 | 1.3 |
> |  |  | FastDup | INet | 62.0 | 1.7 |
> |  |  | SelfClean | DINO | **65.3** | **1.9** |
>
> Note, however, that the classification of label issues in the four categories correctable, multi-label, neither, and non-agreement follows the point of view of human re-annotation and confident learning correction, specialized to mutually exclusive classes.
> The category of label errors defined in our paper is different, as it implies a discrepancy with the actual generating distribution.
> For instance, samples where the correct label is neither the one given in the dataset nor the one proposed by confident learning may be actually off-topic in our categorization.
> Non-agreement samples may have a clear label and be hard to annotate or be intrinsically ambiguous [5].
> The analysis presented above, although very interesting, should be therefore interpreted with care.

---

> > ### Author Rebuttal · Authors · 2024-08-15
> >
> > > *R.fiv4.4*: Severity of data quality
> >
> > The severity of data quality issues was purposely not discussed as this highly depends on the dataset and task, and the identification of general trends depending on the domain and data type deserves separate investigation.
> > A first, very partial indication can be obtained by comparing Table 3 on the influence of cleaning with Table 16 on issues found in fully automatic mode, as in the answer to *R.fiv4.2*.
> > We see the largest effect for Fitzpatrick17k, which mainly contains significant data leakage, followed by DDI, which is very small and contains a few near-duplicates but also some label errors.
> > To clarify this point, we added the following as a second paragraph in section 6 "Influence of dataset cleaning":
> >
> > "The importance of each data quality issue type depends on the dataset and task, and identifying trends by domain and modality requires further investigation.
> > For the limited number of cases in Table 3, taking into account Table 16, data leaks caused by near duplicates across splits seem to have the highest impact, followed by label errors.
> > However, we argue that information on off-topic samples and near duplicates within the same data split is always valuable, even if it only serves the purpose of restoring trust."
> >
> > > *R.fiv4.5*: Image quality
> >
> > We agree with the reviewer that image quality is also an important element for data cleaning.
> > However, in this submission, we focused on data quality issues that we have found to be prevalent and difficult to detect with standard computer vision methods.
> > On the one hand, some image quality issues such as corruption or strong blurring may be identified as off-topic samples, as highlighted in the top section of Table 1 for STL+BLUR, VDR+BLUR, and DDI+BLUR.
> > On the other hand, public data cleaning packages such as CleanLab already cover many image quality aspects (darkness, brightness, odd aspect ratio, odd size, blur, low information) using ad-hoc rules.
> > We recognize that integrating these rules in the SelfClean package would make it a more complete tool.
> >
> > This comment prompted us to add the following sentence at the end of the third paragraph of the introduction:
> > "There are of course other types of data quality issues, including many that can be detected using ad-hoc rules, such as odd brightness, aspect ratio, resolution, sharpness, and entropy in the case of images."
> >
> > > *R.fiv4.6*: AQuA benchmark
> >
> > Thank you for bringing the AQuA benchmark to our attention.
> > Please note that in our submission, we formulate the detection problem as a ranking problem and thus also extensively rely on ranking metrics for evaluation.
> > This differs from AQuA, which evaluates label error detection as a classification problem.
> > To evaluate SelfClean on AQuA, we would have to rely on automatic cleaning, which conflicts with our recommendation on the methodologies usage (see section 6 "Recommended use").
> > Nevertheless, for completeness we are trying to include AUM, CIDER, and SimiFeat in our comparison to ensure all benchmarked label cleaning methods are also represented in our evaluation.
> > We will provide updates as soon as possible.
> > In the long term, we will also consider extending AQuA to include ranking comparisons.
> >
> > > *R.fiv4.7*: CIFAR10H for label errors
> >
> > We thank the reviewer for suggesting to include an evaluation of label errors on CIFAR10H, which provides a valuable example of natural contamination.
> > Results show that SelfClean outperforms its competitors by a clear margin for correctable and non-agreement label errors.
> >
> > | Label Error Type | Positive Samples (%) | Method | Rep. | AUROC (%) | AP (%) |
> > | :---: | :---: | :---: | :---: | :---: | :---: |
> > | Correctable | 0.5 | CLearning | INet | 57.2 | 0.8 |
> > |  |  | FastDup | INet | 68.9 | 0.9 |
> > |  |  | SelfClean | DINO | **76.3** | **1.7** |
> > | Non-agreement | 0.7 | CLearning | INet | 56.3 | 0.9 |
> > |  |  | FastDup | INet | 66.8 | 1.2 |
> > |  |  | SelfClean | DINO | **72.0** | **3.2** |
> >
> > Please note, however, that we already present similar experiments on real label errors alongside synthetic label noise.
> > These include ImageNet-1k [2] and Food-101N [3], where human reannotation was gathered for a subset of the data, see section 5.2 and appendix H.2.
> >
> > > *R.fiv4.8*: Ethical concerns CelebA
> >
> > We thank the reviewer for helping to prevent unethical usage of CelebA.
> > The outcome of the ethics review was that there is no problem with the usage of CelebA in our submission.
> > However, following the suggestion from the ethics we added a statement regarding CelebA, addressing potential ethical concerns:
> >
> > "The CelebA dataset contains images of public figures, and while it is widely used in research, it is important to consider privacy, consent, and potential biases [4].
> > We have ensured that our usage complies with the dataset's terms and conditions, and we advise caution to avoid perpetuating any biases inherent in the dataset.
> > Our work does not involve any manipulation or generation of images that could misrepresent individuals."

---

> > ### Author Rebuttal · Authors · 2024-08-15
> >
> > **Minor Weaknesses**:
> >
> > > *R.fiv4.9*: Term 'computer vision datasets' is too broad
> >
> > We agree with the reviewer that the term 'computer vision datasets' is too broad, and thank them for pointing this out.
> > On the other hand, off-topic samples and near duplicates are not limited to classification tasks.
> > We thus have changed it to "image datasets" accordingly.
> >
> > > *R.fiv4.10*: Reference format
> >
> > We thank the reviewer for pointing out the inconsistencies.
> > The formatting is fixed by the required style file.
> > We will check all bibtex entries to ensure appropriate reference format.
> >
> > > *R.fiv4.11*: Published ArXiv papers
> >
> > We thank the reviewer for this comment and will update bibtex entries accordingly.
> >
> > > *R.fiv4.12*: Section D Evaluation metrics
> >
> > We agree that moving Section D to the main paper would be better and thus changed the submission accordingly.
> >
> > > *R.fiv4.13*: More instructions code and missing repositories
> >
> > We agree with the reviewer that the README can be improved, and will make this a priority.
> > Tutorials for running the code have also been developed in the meantime and can be provided upon request.
> > Due to the anonymization platform's limitations, submodules, such as our pre-training weights, are not integrated into the anonymized repository.
> > However, they will be part of the open-sourced version of the repositories.
> > If the reviewer would like to see them during the rebuttal, we will find a way to make them accessible.
> >
> > > *R.fiv4.14*: Human verification tool
> >
> > At the time of the submission, the human verification tool was not yet open-sourced.
> > However, has been made available in the meantime.
> > You can access the anonymized repository at: https://anonymous.4open.science/r/selfclean_verification_tool-BDD2
> >
> > Please note that, while this is useful for reproducibility, it is not a core contribution of our submission and there are other public tools with similar functions.
> >
> > **References**:
> >
> > [1] Dunnmon, J., Carneiro, G., \& Ré, C. (2020). Hidden Stratification Causes Clinically Meaningful Failures in Machine Learning for Medical Imaging.
> > [2] Northcutt, C. G., Athalye, A., \& Mueller, J. (2021). Pervasive Label Errors in Test Sets Destabilize Machine Learning Benchmarks.
> > [3] Lee, K., He, X., Zhang, L., \& Yang, L. (2017). CleanNet: Transfer Learning for Scalable Image Classifier Training with Label Noise.
> > [4] Peng, K., Mathur, A., \& Narayanan, A. (2021). Mitigating Dataset Harms Requires Stewardship: Lessons from 1000 Papers.
> > [5] Vasudevan, V., Caine, B., Gontijo Lopes, R., Fridovich-Keil, S., \& Roelofs, R. (2022). When Does Dough Become a Bagel? Analyzing the Remaining Mistakes on ImageNet.

---

> > > ### Comment · Reviewer_fiv4 · 2024-08-19
> > > **Response to the rebuttal**
> > >
> > > The authors have addressed most of the concerns. The reviewer has raised the rating from 6 to 7. However, three remaining parts should be improved.
> > > - The explanation of the effect of near duplicate samples is not convincing. It would be better for the authors to provide some references (related work) to support the train-evaluation leakage issue.
> > > - It would be better to compare this work with the AQuA benchmark for completeness. Selecting one or two datasets for comparison would be enough during the rebuttal.
> > > - Since this is the dataset/benchmark track, it is necessary for the authors to provide a detailed tutorial/guidebook for reviewers to check reproducibility.

---

> > > > ### Author Rebuttal · Authors · 2024-08-21
> > > >
> > > > We thank the reviewer fiv4 for acknowledging our response and raising their score.
> > > >
> > > > > *R.fiv4.15*: The explanation of the effect of near duplicate samples is not convincing. It would be better for the authors to provide some references (related work) to support the train-evaluation leakage issue.
> > > >
> > > > The cause of data leakage [1, 2] investigated in this submission is the presence of near duplicates across splits [3, 4],
> > > > which results in high performance on the training and evaluation sets but poor generalization to truly unseen data [3].
> > > >
> > > > In our submission, we conservatively estimate the percentage of errors for 12 image datasets and find that on average 1.8\% of samples are near duplicates of other images.
> > > > This is significantly higher than the estimates for off-topic samples (0.1\%) and label errors (0.4\%).
> > > > Combined with the fact that some of these datasets, 5 out of 12 of the analyzed ones, do not contain a dedicated train-evaluation split, this results in leakage across splits.
> > > >
> > > > To assess the impact of the data quality issues, we cleaned specific dataset splits in Table 3.
> > > > For Fitzpatrick17k, cleaning the evaluation set results in significantly lower performance (-4.1\%pt. in F1).
> > > > As 95\% of the data quality issues of Fitzpatrick17k are near duplicates, this effect can be attributed to leakage.
> > > > This is a concrete example where the generalization performance is significantly overestimated because of train-evaluation leakage due to near duplicates.
> > > >
> > > > > *R.fiv4.16*: It would be better to compare this work with the AQuA benchmark for completeness. Selecting one or two datasets for comparison would be enough during the rebuttal.
> > > >
> > > > We acknowledge the comment from the reviewer and have accordingly raised the priority of adapting the AQuA benchmark to include an evaluation in terms of ranking performance.
> > > > We will post progress updates.
> > > >
> > > > > *R.fiv4.17*: Since this is the dataset/benchmark track, it is necessary for the authors to provide a detailed tutorial/guidebook for reviewers to check reproducibility.
> > > >
> > > > We agree on the importance of reproducibility.
> > > > We therefore extended the README of the evaluation repository with the commands to reproduce all experiments.
> > > >
> > > > https://anonymous.4open.science/r/SelfClean-Evaluation-0E0F
> > > >
> > > > The self-supervised training code is accessible at: https://anonymous.4open.science/r/ssl_library-84F8
> > > >
> > > > **References:**
> > > >
> > > > [1] Lones, M. A. (2021). How to avoid machine learning pitfalls: A guide for academic researchers.
> > > > [2] López, J. A., Chen, B., Saaz, M., Sharma, T., \& Varró, D. (2024). On Inter-dataset Code Duplication and Data Leakage in Large Language Models.
> > > > [3] Kapoor, S., \& Narayanan, A. (2023). Leakage and the reproducibility crisis in machine-learning-based science.
> > > > [4] Abhishek, K., Jain, A., \& Hamarneh, G. (2024). Investigating the Quality of DermaMNIST and Fitzpatrick17k Dermatological Image Datasets.

---

> > > > > ### Author Rebuttal · Authors · 2024-08-27
> > > > >
> > > > > We thank the reviewer *fiv4* for their patience while we ran the requested experiment.
> > > > >
> > > > > > *R.fiv4.16*: It would be better to compare this work with the AQuA benchmark for completeness. Selecting one or two datasets for comparison would be enough during the rebuttal.
> > > > >
> > > > > We compared SelfClean to other methods in terms of binary classification performance for label error detection on CIFAR-10 using the $F_1$ score, as per Table 8 of the AQuA paper [1].
> > > > > To this end, we apply SelfClean in fully automatic mode, even if this only provides an estimate and should not be used without human supervision, as discussed in Section 6 "Recommended use".
> > > > > The results show that SelfClean's performance is competitive with the other methodologies and achieves the second-best performance for 3 out of 4 noise types and the third-best for one of them.
> > > > >
> > > > > | Label Error Type | AUM | CINCER | CLearning | SimiFeat | SelfClean |
> > > > > | :--- | :--- | :---: | :---: | :---: | :---: |
> > > > > | Uniform | 84.9 | 81.7 | 18.1 | $\mathbf{8 9 . 8}$ | 86.1 |
> > > > > | Asymmetric | 85.0 | 85.0 | 18.1 | $\mathbf{8 9 . 0}$ | 85.9 |
> > > > > | Class-dependent | 37.0 | $\mathbf{6 4 . 9}$ | 37.9 | 58.6 | 38.5 |
> > > > > | Instance-dependent | 78.5 | 79.3 | 19.0 | $\mathbf{8 3 . 2}$ | 79.5 |
> > > > >
> > > > > A comparison of the ranking for candidate label errors in terms of AUROC and AP would be more informative and fit better in the paper.
> > > > > Unfortunately extracting scores from the methods included in the AQuA benchmark is not straightforward and would require careful validation, which goes beyond what we can achieve during rebuttal.
> > > > >
> > > > > **References:**
> > > > >
> > > > > [1] Goswami, M., Sanil, V., Choudhry, A., Srinivasan, A., Udompanyawit, C., \& Dubrawski, A. (2023). AQuA: A Benchmarking Tool for Label Quality Assessment.

---

> > > > > > ### Comment · Reviewer_fiv4 · 2024-08-30
> > > > > >
> > > > > > Thanks for the detailed response. Based on the author's rebuttal and other reviewers' comments, the reviewer will maintain the rating (accept).

---

### Official Review · Reviewer_tscY · 2024-07-12
**SSL-based method for detecting data errors**

**Rating:** 7
**Confidence:** 3
**Correctness:** No reason to believe that something i…
**Clarity:** Paper is clear and easy to follow.

**Review:**

I do not have any major concerns regarding the paper. It is easy to follow, the proposed method is plausible and outperforms the SOTA. The experiments and ablation study are also plausible and support the claims.

**Strengths:**

The method is easy to follow and achieves SOTA performance.

**Additional Feedback:**

No additional feedback.

**Documentation:**

Not applicable.

**Ethics:**

No ethical concerns.

**Limitations:**

I do not see a negative societal impact. Limitations are dressed in the ablation study.

**Opportunities For Improvement:**

In section 5, the identified label errors with SelfClean are compared with human annotators. I would be interesting to report the same for the other benchmarked methods.

**Relation To Prior Work:**

The method is compared to prior work. However, the other works are introduced in the appendix not in the main paper.

**Summary And Contributions:**

The paper introduces SelfClean, a method for detecting data errors based on representations learned from SSL. The method is benchmarked against SOTA methods for detecting near duplicates, label errors and off-topic samples and outperforms the SOTA in several cases.

---

> ### Author Rebuttal · Authors · 2024-08-15
>
> We sincerely thank reviewer tscY for their sharp feedback.
> Below we briefly comment on the opportunity for improvement and on the relation to prior work.
>
> > *R.tscY.1*: In section 5, the identified label errors with SelfClean are compared with human annotators. I would be interesting to report the same for the other benchmarked methods.
>
> We agree that evaluating the rankings produced by other methodologies with human verification, as we did for SelfClean, would provide a more complete picture.
> We are setting up this experiment and will post updates on whether it is feasible during the rebuttal period.
>
> > *R.tscY.2*: The method is compared to prior work. However, the other works are introduced in the appendix not in the main paper.
>
> Due to space constraints and the number of prior works we compare to, we had to introduce them in appendix F rather than the main paper.
> This appendix is referenced both at line 182 and in the caption of table 1.

---

> > ### Author Rebuttal · Authors · 2024-08-21
> >
> > We thank the reviewer for their patience and would like to discuss the requested extension of the human verification experiment.
> >
> > > *R.tscY.1*: In section 5, the identified label errors with SelfClean are compared with human annotators. I would be interesting to report the same for the other benchmarked methods.
> >
> > We extended the validation of the algorithmic rankings with humans to include the two best-performing competing methods for each task.
> > The validation has been performed for ImageNet, a general image dataset, as the rebuttal timeline is insufficient to collect medical expert annotations.
> > Furthermore, due to ImageNet's size and the rebuttal's timeline, comparing near duplicates is not feasible, as running pHashing and SSIM each take 28 days to compute more than one billion comparisons.
> >
> > The table compares the percentage of issues found by humans in the 50 lowest-ranked samples with 50 random samples and in samples 1 through 25 with samples 26 through 50.
> > We report the percentage of issues in each sample and the corresponding $p$-value of a Mann–Whitney $U$ test, which represents the probability for the ranking to be unrelated to the position of positive samples.
> >
> > The crowdsourced evaluation produced significantly different sample distributions for SelfClean and competing methods.
> > To cross-check our results against unintended changes, given the large time gap between the initial evaluation and the current one, we repeated the procedure for SelfClean for the very same samples.
> >
> > The results show a significantly better correspondence of the SelfClean rankings to crowdsourced annotations compared to competitors.
> > However, there are relatively high fluctuations in significance scores when the experiment is repeated, hinting at a possible low consistency of crowd workers.
> >
> > | Data Quality Issue | Method | Lowest 1-50 (%) | Random Sample (%) | $p$-value | Lowest 1-25 (%) | Lowest 26-50(%) | $p$-value |
> > | :---: | :---: | :---: | :---: | :---: | :---: | :---: | :---: |
> > | Off-topic Samples | HBOS | 2 | 4 | 0.72 | 0 | 4 | 0.84 |
> > |  | ECOD | 2 | 6 | 0.85 | 4 | 0 | 0.16 |
> > |  | SelfClean | 62 | 48 | 0.08 | 56 | 68 | 0.80 |
> > |  | SelfClean (relabeled) | 32 | 6 | $\mathbf{4.8 \times 10^{-4}}$ | 44 | 20 | $\mathbf{0.04}$ |
> > | Label Errors | FastDup | 0 | 6 | 0.96 | 0 | 0 | undef |
> > |  | CLearning | 4 | 10 | 0.88 | 4 | 4 | 0.5 |
> > |  | SelfClean | 36 | 0 | $\mathbf{1.6 \times 10^{-6}}$ | 48 | 24 | $\mathbf{0.04}$ |
> > |  | SelfClean (relabeled) | 34 | 14 | $\mathbf{0.01}$ | 44 | 24 | 0.07 |

---

> > > ### Comment · Reviewer_tscY · 2024-08-26
> > >
> > > I appreciate that the authors performed these (and the other) experiments during the rebuttal phase. As my rating is already high, I will leave it unchanged.

---

### Official Review · Reviewer_qYrV · 2024-07-22

**Rating:** 6
**Confidence:** 3
**Correctness:** yes
**Clarity:** yes

**Review:**

**Quality:**
- The paper presents a rigorous approach to a significant problem in machine learning: data quality. The methodology is well-researched and grounded in contemporary machine learning techniques.
- The use of self-supervised learning for representation is a strong choice, as it avoids the biases inherent in supervised learning while still capturing the nuances of the data.
- The paper includes comprehensive experiments across multiple datasets, demonstrating the robustness and generalizability of the SELFCLEAN method.

**Clarity:**
- The paper is well-structured, with clear sections that logically flow from the introduction of the problem to the methodology, experiments, and conclusions.

**Originality:**
- SELFCLEAN introduces a novel approach to data cleaning that relies on self-supervised learning and distance-based indicators, which is a unique contribution to the field.
- The formulation of data cleaning as ranking or scoring problems is innovative and significantly reduces the need for manual inspection.

**Significance:**
- Addressing data quality issues is crucial for the advancement of AI, especially in high-stakes domains such as healthcare. The SELFCLEAN method has the potential to improve the reliability of AI systems.
- The work contributes to the growing field of data-centric AI, emphasizing the importance of clean data over the quantity of data.

**Pros:**
1. **Innovative Approach**: The use of self-supervised learning for data cleaning is a novel and promising approach.
2. **Comprehensive Evaluation**: The paper includes a wide range of datasets and a detailed analysis of different data quality issues.
3. **Practical Implications**: The SELFCLEAN method can be applied to real-world datasets, improving the reliability of AI systems.
4. **Human-in-the-loop**: The method offers a mode that includes human oversight, which can be crucial for certain applications where automated decisions may not be sufficient.
5. **Openness to Improvement**: The authors acknowledge areas for future work and improvements, showing an understanding of the evolving nature of the field.

**Cons:**

1. **Scalability Concerns**: While the method is effective, the current formulation may not scale well with extremely large datasets without approximation methods.
2. **Dependence on Hyperparameters**: The automatic cleaning procedure relies on hyperparameters like the contamination rate guess and significance level, which may require careful tuning.
3. **Comparisons to SOTA**: Authors may be encouraged to compare their method to more methods, e.g., Jia et al 2023, Pleiss et al 2020 etc for off-topic samples.
4. **Tables Presentation**: Figures are well illustrated that readers can know the core methodology very quickly. However, the tables are less easy to understand. For example,

Overall, the paper presents a significant contribution to the field of machine learning with a well-thought-out and innovative approach to data quality audits. The SELFCLEAN method shows promise in improving the reliability of AI systems, particularly in critical domains where data quality is paramount.

Reference:
- Jia, Qingrui, et al. "Learning from training dynamics: Identifying mislabeled data beyond manually designed features." Proceedings of the AAAI Conference on Artificial Intelligence. Vol. 37. No. 7. 2023.
- Pleiss, Geoff, et al. "Identifying mislabeled data using the area under the margin ranking." Advances in Neural Information Processing Systems 33 (2020): 17044-17056.

**Strengths:**

1. **Significance of the Contribution**:
   - The paper introduces SELFCLEAN, a novel and potentially transformative approach to data cleaning that is particularly relevant in the era of big data and deep learning.
   - It addresses a critical issue in machine learning – the presence of noisy, off-topic, and incorrectly labeled data – which can significantly impact model performance and reliability.

2. **Relevance to the Broader Research Community**:
   - The SELFCLEAN methodology is applicable across various domains, including computer vision and medical imaging, making it of broad interest to researchers and practitioners in AI.
   - By focusing on self-supervised learning, the paper contributes to an area of active research and could inspire further work in unsupervised and self-supervised paradigms.

3. **Quality of the Research**:
   - The paper is methodologically sound, with a clear problem statement, a well-defined approach, and a comprehensive experimental evaluation.
   - The use of multiple datasets and the comparison with existing methods demonstrate the robustness and generalizability of SELFCLEAN.
   - The research is thorough, with ablation studies and analysis of different components of the SELFCLEAN framework, contributing to a deeper understanding of its strengths and potential limitations.

4. **Ethical and Social Implications**:
   - By improving data quality, the SELFCLEAN method can lead to more accurate and fair AI systems, which is essential for ethical AI practices.
   - The paper acknowledges potential biases in data and the importance of human-in-the-loop validation, showing a responsible approach to AI development.
   - The authors discuss the potential for SELFCLEAN to be used in high-stakes domains such as healthcare, where the implications of data quality are particularly significant.

5. **Innovation**:
   - The SELFCLEAN approach is innovative in its use of self-supervised learning for data cleaning, offering a new perspective on an old problem.
   - The formulation of data cleaning as ranking or scoring problems is a creative solution that simplifies the complex task of data auditing.

6. **Practical Impact**:
   - The paper's findings have practical implications for dataset curation and can lead to improved benchmarks, which are crucial for the advancement of AI technologies.
   - The proposed method could be integrated into existing data pipelines, providing a significant boost to the data quality of various AI applications.

7. **Transparency and Reproducibility**:
   - The paper provides a detailed description of the methodology, experimental setup, and hyperparameters, which facilitates transparency and reproducibility of the research.

8. **Interdisciplinary Potential**:
   - The work has interdisciplinary potential, as it combines insights from machine learning, data management, and domain-specific knowledge (e.g., medical imaging), which could attract a diverse range of researchers.

**Additional Feedback:**

no.

**Documentation:**

n/a

**Limitations:**

yes

**Opportunities For Improvement:**

1. **Computational Resources**:
   - The requirement for self-supervised pre-training on the entire dataset may demand substantial computational resources, which could be a limitation for researchers or practitioners with limited access to high-performance computing.

2. **Scalability**:
   - The paper acknowledges that the current formulation of near duplicate detection does not scale well with very large datasets, indicating a need for more scalable solutions.

3. **Dataset Bias**:
   - SELFCLEAN may inherit biases present in the dataset, particularly if the dataset is not diverse or representative of the broader population it aims to serve.

4. **Dependence on Hyperparameters**:
   - The automatic cleaning procedure is sensitive to the choice of hyperparameters such as the contamination rate guess and significance level, which may require careful tuning and could introduce subjectivity.

5. **Lack of Remediation Strategies**:
   - The paper focuses on identifying data quality issues but does not provide extensive strategies for how to remediate these issues once identified.

6. **Potential Over-Cleaning**:
   - There is a risk that the cleaning process might remove valuable outliers or rare cases that, while not fitting the general distribution, are still informative for certain tasks.

7. **Human-in-the-loop Trade-offs**:
   - Relying on human validation introduces additional complexity and may not be feasible at scale, and there is a risk of introducing human bias into the cleaning process.

8. **Evaluation Methodology**:
   - The evaluation of data cleaning frameworks lacks a standard protocol, which may affect the comparability and generalizability of the results.

9. **Assumption of Data Distribution**:
    - The methodology assumes a certain distribution of data quality issues, which may not hold true for all datasets, particularly those with unique characteristics or non-standard contamination.

10. **Generalization to Other Types of Data**:
    - The SELFCLEAN method has been demonstrated on image datasets; its effectiveness on other types of data, such as text or time-series, is not discussed.

11. **Label Granularity**:
    - The paper notes that label error detection becomes more challenging as label granularity increases, suggesting that the method may not be as effective for fine-grained classification tasks.

**Relation To Prior Work:**

yes

**Summary And Contributions:**

This paper discusses a novel methodology called SELFCLEAN for improving the quality of datasets in computer vision and medical imaging. The authors highlight the importance of clean data for training robust machine learning models and the challenges associated with cleaning large-scale datasets, with contributions as following:

1. **Data Quality Issues**: The paper begins by discussing the common issues found in benchmark datasets, such as off-topic images, near duplicates, and label errors. These issues can lead to inaccurate estimates of model performance.

2. **SELFCLEAN Methodology**: The authors propose SELFCLEAN, a data cleaning procedure that uses context-aware self-supervised representation learning and distance-based indicators to identify data quality issues. This method does not require manual annotation and is designed to work with the dataset itself.

3. **Ranking and Scoring Problems**: SELFCLEAN formulates dataset cleaning as a set of ranking or scoring problems. This approach reduces the effort needed for manual inspection and allows for automated decisions based on score distributions.

4. **Representation Learning**: The method involves training a deep feature extractor using self-supervised learning (SSL) to obtain latent representations of the dataset samples. The authors use vision transformer (ViT) encoders and compare different SSL methods like SimCLR and DINO.

5. **Distance-Based Indicators**: SELFCLEAN uses distance-based indicators to detect off-topic samples, near duplicates, and label errors. It employs agglomerative clustering, pairwise distances, and intra-/extra-class distance ratios for these tasks.

6. **Experiments**: SELFCLEAN outperforms competing methods in detecting data quality issues across various benchmarks. It shows superior performance in aligning with metadata and expert verification in natural settings.

7. **Influence of Dataset Cleaning**: The paper discusses the impact of cleaning data quality issues on model performance. Cleaning the evaluation set significantly alters scores, and cleaning the training set has a positive impact for many benchmarks.

---

> ### Author Rebuttal · Authors · 2024-08-15
>
> We are grateful to reviewer qYrV for their extensive and detailed review. The main concerns and opportunities for improvement are addressed below.
>
> **Cons**:
>
> > *R.qYrV.1*: Scalability Concerns
>
> We agree that the current formulation does not scale well for near duplicates, as mentioned in appendix B "Limitations".
> However, we used multiple engineering tricks, such as memory mapping, multi-threading, and custom implementations, to ensure the current implementation is sufficiently fast for many real use cases.
> Please also consult our answer to *R.qYrV.5* for specific details on the computational cost of running SelfClean.
> We are also testing ideas to improve scaling for very large datasets, such as approximation methods and iterative analysis of nearest-neighbor distances, as mentioned in appendix B "Limitations".
>
> > *R.qYrV.2*: Dependence on Hyperparameters
>
> We agree that the automatic cleaning option features some weak dependence on these two hyperparameters as investigated in appendices L.3 and L.4.
> Specifically, in L.3 we show that the fraction of found problems does not depend much on the contamination rate guess over several orders of magnitude, and in L.4 that the significance level has a relation that is less than linear with the found problems.
>
> Please note that competing approaches also use similar hyperparameters, such as a contamination rate to set a hard threshold [1].
> Furthermore, as described in section 6 "Recommended use" we encourage using SelfClean with a human-in-the-loop that does not feature any additional hyperparameters.
>
> > *R.qYrV.3*: Comparisons to SOTA
>
> Thank you for suggesting additional methods for comparison.
> We will explore the feasibility of incorporating these during the rebuttal period and will provide updates accordingly.
> However, we would like to clarify that the methods referenced, specifically Jia et al. (2023) and Pleiss et al. (2020), focus on detecting label errors rather than off-topic samples.
>
> > *R.qYrV.4*: Tables Presentation
>
> We are aware that some of our tables expose the complexity of evaluating multiple methods on several tasks and datasets.
> Unfortunately, the reviewer's comment stops after "[...] For example,".
> It would be very useful to have concrete suggestions on how to improve the readability of our tables.

---

> > ### Author Rebuttal · Authors · 2024-08-15
> >
> > **Opportunities For Improvement**:
> >
> > > *R.qYrV.5*: Computational Resources
> >
> > The computational requirements of the self-supervised pre-training indeed change with the size of the dataset.
> > For example, the examples provided in the anonymized SelfClean repository can all be run reasonably quickly on Google Colab with GPU acceleration.
> > Specifically, for a dataset of 13,000 samples (such as ImageNette) applying SelfClean takes under 1.5 hours with the current implementation on Colab.
> > For larger datasets pre-training will require more time and computational power.
> > However, it only has to be carried out once, and model weights can be reused as a starting point for downstream tasks.
> >
> > > *R.qYrV.6*: Dataset Bias
> >
> > SelfClean may indeed inherit biases present in the dataset, as discussed in appendix B "Limitations".
> > This was further investigated for off-topic sample detection in the experiment of section H.3, where the results show that, at least in these situations, no systematic bias against underrepresented groups is present.
> >
> > It is important to note that inherent dataset bias may similarly affect other methodologies requiring the training of a supervised classifier, such as the popular approach of confident learning [3].
> >
> > > *R.qYrV.7*: Lack of Remediation Strategies
> >
> > The paper indeed only focuses on identifying data quality issues as discussed in appendix B "Limitations".
> > It is our opinion that the tasks of identification and remedy are best discussed separately, as their core can be used independently of each other.
> > Furthermore, remedial strategies are often specific to the case at hand.
> >
> > > *R.qYrV.8*: Potential Over-Cleaning
> >
> > The problem of over-cleaning is indeed a reason why the best treatment of data quality issues depends on the specific case.
> > The focus of the paper is detecting data quality issues and not the improvement of classifier scores.
> > Furthermore, by resorting to human confirmation (preferably carried out by experts of the respective data subject), we argue that valuable outliers should get treated accordingly.
> > Please also consult our answer to *R.fiv4.1* for more details and a clarification added to the paper.
> >
> > > *R.qYrV.9*: Human-in-the-loop Trade-offs
> >
> > We agree that relying on human confirmation requires additional resources.
> > On the other hand, [2] has already proposed a solution to reduce the effort required for confirmation.
> >
> > Introducing human bias is indeed a possible result of manual data quality verification.
> > The process should therefore be performed with due care.
> > Multiple annotations, cleaning protocols and transparent documentations are some of the measures which can help to mitigate human bias.
> >
> > > *R.qYrV.10*: Evaluation Methodology
> >
> > We agree that the lack of evaluation standards for data cleaning hinders comparisons.
> > However, we make our evaluation workflow fully available and we encourage practitioners to reuse it.
> > Furthermore, we release SelfClean's implementation to facilitate its inclusion in future standard benchmarks.
> >
> > > *R.qYrV.11*: Assumption of Data Distribution
> >
> > We agree that there are assumptions about the distribution of data quality issues that influence the effectiveness of the automatic cleaning strategy.
> > This is one of the reasons why the human-in-the-loop setting is recommended, as violations of the assumptions can be effectively detected.
> > The fully automatic mode is intended as a means of quickly checking the number of quality issues and estimating the impact of cleaning.
> >
> > > *R.qYrV.12*: Generalization to Other Types of Data
> >
> > This paper intentionally focuses on identifying data quality issues in image datasets, as discussed in appendix B "Limitations".
> >
> > > *R.qYrV.13*: Label Granularity
> >
> > Label error detection may indeed become more difficult as granularity increases as stated in the paper and observed by the reviewer.
> > However, as summarized in section 5.3 and evaluated in appendix G.3, SelfClean performance stays on par with other benchmarked methods in these settings.
> >
> > **References**:
> >
> > [1] Zhao, Y., Nasrullah, Z., \& Li, Z. (2019). PyOD: A Python Toolbox for Scalable Outlier Detection.
> > [2] Gröger, F., Lionetti, S., Gottfrois, P., Groh, M., Daneshjou, R., Consortium, L., Navarini, A. A., \& Pouly, M. (2023). Towards Reliable Dermatology Evaluation Benchmarks.
> > [3] Northcutt, C. G., Jiang, L., \& Chuang, I. L. (2021). Confident Learning: Estimating Uncertainty in Dataset Labels.

---

> > > ### Author Rebuttal · Authors · 2024-08-27
> > >
> > > We thank reviewer *qYrV* for their patience while we ran the requested experiments.
> > >
> > > > *R.qYrV.3*: Comparisons to SOTA: Authors may be encouraged to compare their method to more methods, e.g., Jia et al 2023, Pleiss et al 2020 etc for off-topic samples.
> > >
> > > Reviewer *fiv4* suggested incorporating SelfClean into the AQuA benchmark [1] for label error detection.
> > > By doing so, we also compared its performance against three additional label error detection methods, including AUM.
> > > We refer to our answer *R.fiv4.16* for results and discussion.
> > >
> > > **References:**
> > >
> > > [1] Goswami, M., Sanil, V., Choudhry, A., Srinivasan, A., Udompanyawit, C., \& Dubrawski, A. (2023). AQuA: A Benchmarking Tool for Label Quality Assessment.

---

### Author Rebuttal · Authors · 2024-08-29

We are grateful to all reviewers for their attentive analysis and for their comments, which greatly improved the manuscript.

We are encouraged that they recognize data cleaning to be a crucial (qYrV) and **highly relevant research topic** (fiv4), which **aligns well with the track's scope** (fiv4).
It gives us confidence that our analysis is deemed rigorous and innovative (qYrV), plausible (tscY), **original and significant** (fiv4), and of broad interest to researchers and practitioners (qYrV).
Reviewers also acknowledge that the submission contains a **comprehensive amount of experiments** and ablation studies (qYrV, fiv4) across a wide variety of datasets (fiv4) and consider them **to support the claims** (tscY).
It is great to see our effort towards transparency and reproducibility recognized (qYrV).
Last but not least, we are very happy that our submission was found to be **well-structured** (qYrV) and easy to follow (tscY).

We also appreciate the valuable and constructive critique of our work.
We **carefully considered all points** raised by the reviewers and addressed them during rebuttal by providing details or running additional experiments.
Specifically, the discussion led to the following **improvements**:

- The human validation of rankings was extended to competing methods.
- CIFAR10H was added to the comparison with metadata.
- SelfClean was integrated into the AQuA benchmark and compared to additional label error detection methods based on training dynamics.
- Label error detection was separately evaluated for four different types of errors.
- Clarifications have been provided for off-topic samples, severity and diversity of data quality issues, and CelebA's usage.
- We extended the repositories' READMEs with commands to run all experiments.

We thank all reviewers and the AC for their consideration and remain at their disposal for additional clarifications.

---

### Decision · Program_Chairs · 2024-09-26

**Decision:**

Accept (Poster)

**Comment:**

All three reviewers are positive with two clear accept. The authors have addressed most of the reviewers' questions satisfactorily. The problem addressed in this paper is important and timely, and it will benefit the wider research community.